# Climate variability and grain production in Scania, c. 1702-1911

Martin Skoglund[1]

[1] Division of Agrarian History, Swedish University of Agricultural Sciences, Uppsala, 756 51, Sweden

*Correspondence to*: Martin Skoglund (martin.skoglund@slu.se)

**Abstract.** Scania (sw. *Skåne*), southern Sweden, offers a particularly interesting case for studying the historical relationship between climate variability and grain production, given the favourable natural conditions in terms of climate and soils for grain production, as well as the low share of temperature-sensitive wheat varieties in its production composition. In this article, a contextual understanding of historical grain production in Scania, including historical, phenological and natural geographic aspects, is combined with a quantitative analysis of available empirical sources to estimate the relationship between climate variability and grain production between the years c. 1702-1911. The main result of this study is that grain production in Scania was primarily sensitive to climate variability during the high summer months of June and July, preferring cool and humid conditions, and to some extent precipitation during the winter months, preferring dry conditions. Diversity within and between historical grain varieties contributed to making this risk manageable.

Furthermore, no evidence is found for grain production being particularly sensitive to climate variability during the spring, autumn and harvest seasons. At the end of the study period, these relationships were shifting as the so-called early improved cultivars were being imported from other parts of Europe. Finally, I also shed new light on the climate history of the region, especially for the late 18[th] century, previously argued to be a particularly cold period, through homogenization of the early instrumental temperature series from Lund (1753-1870).

## 1 Introduction

In recent years, numerous studies have explored the relationship between grain yields, prices and climatic change in medieval and early modern Europe. The fundamental assumption underlying these studies is that grain production to a substantial degree was affected by variability in temperature and precipitation ( Edvinsson *et al.*, 2009; Holopainen *et al.* 2012; Camenisch, 2015; Esper *et al.*, 2017; Pribyl, 2017; Ljungqvist *et al.*, 2021a; Ljungqvist *et al.*, 2021b). Most of these studies have either focused on particularly temperature-sensitive grain types like wheat, or temperature-sensitive agricultural regions, like Finland or the Scottish Highlands (Parry & Carter, 1985; Brunt, 2015; Huhtamaa & Helama, 2017a). In these historical contexts, cold conditions becomes the 'grim reaper' (Holopainen & Helama, 2009). However, in the long-term, grain farming even in the northern border regions of European agriculture has shown considerable adaptability and resilience (Solantie, 1992; Huhtamaa & Helama, 2017b; Degroot, 2021). A diversified grain production has been identified as an important aspect of this resilience (Michaelowa, 2001). In this article, I argue that an understanding of the impact of climatic variability and change on agriculture

as well as explanations of resilience in terms of grain diversity, need to be grounded in an understanding of the phenology of historical grain varieties.

Attempts to account for the resilience, or the ability of early modern farmers and farming systems to cope with climate variability in intensive grain farming areas of Europe north of the Alps like northern France and England, have remained mainly hypothetical (Michaelowa, 2001; Tello *et al.*, 2017). Early modern Scania offers an especially interesting case in this regard. The climate of Scania is mild, hosting a continental climate stabilized by the proximity to the Baltic Sea. From an agronomic viewpoint, it is often stressed that Scania has the longest vegetative period of present-day borders Sweden (Osvald, 1959; Persson, 2015). Moreover, the southwestern half of Scania, roughly the extent of the historical county of Malmöhus, contains large areas of soils of exceptionally high quality (Lantbruksstyrelsen, 1971). For most of the historical period, Scania was an important surplus producer of grains in the Kingdom of Denmark and from 1658 in the Kingdom of Sweden (Åmark, 1915; Bohman, 2010). At the same time, since at least the 17th century up until the end of the 19th century, Scanian farmers relied on Scandinavian grain varieties adapted to cooler and humid climates with short growing seasons, i.e. conditions often prevalent at the northern limits of arable agriculture (Lundström *et al.*, 2018; Larsson *et al.*, 2019).

The aim of this article is to study the relationship between climate variability and grain production in Scania during the period c. 1702-1911. The study period is divided into the early study period (1702-1911) and the late study period (1865-1911). Given that the role of climate cannot be conceptualized neither in a simplistic or deterministic manner, it has to be contextualized in the specific agrarian and ecological context (Haldon *et al.*, 2018; van Bavel *et al.*, 2019; Degroot, 2021). Accordingly, this article starts out by contextualizing the study and setting the historical background, particularly detailing historical grain production in Scania during the study period. Following this is a conceptual and theoretical discussion of the relationship between crop production and climate and the concept of resilience. Subsequently, I present and discuss the climate- and agricultural production data and the employed methods. Finally, results are discussed in relation to the historical context as well as to previous research.

## 1.1 Background

Scania is situated in the southern-most tip of the Scandinavian Peninsula in the borderlands between Sweden and Denmark. The farming districts on the plains of Scania have, and continue to be, some of the most productive arable farming regions in Scandinavia, owing mostly to its mild climate and rich soils. In the Danish and Swedish historiography, Scania is commonly referred to as a *kornbod* (roughly translated as 'breadbasket'). Adam of Bremen in his Gesta Hammaburgensis ecclesiae pontificum from c. 1075 AD describes Scania as the most prosperous of the provinces in the Danish kingdom (Bremensis, 2002). However, the natural geography of Scania is and was not uniform (Svensson, 2016). Besides the arable plains, Scania was constituted by a diverse landscape of forests, disparate but mostly hospitable coastal areas, lakes and hills with different

soils and natural conditions (Lidmar-Bergström *et al.*, 1991).[1] Farming was to some extent adapted to this variability in natural conditions, especially in the period prior to the late 19th century (Dahl, 1989; Gadd, 2000; Bohman, 2010). During the years c. 1750-1850, Scania underwent what has been called the agrarian revolution, implicating a general transformation of agriculture as well as dramatic and sustained increases in production (Olsson & Svensson, 2010). Subsequently, Scanian agriculture has continued to sustain its growth trajectory, intermittently interrupted by various agrarian and economic crises

(Myrdal & Morell, 2013). Scanian farmers did also face challenges. Situated between two rivalling Kingdoms, Denmark and Sweden, the fertile plains of Scania have been fought over and acted as a battleground in numerous wars. After 1711, there were fewer conflicts compared to the preceding centuries (Frost, 2000).

Like in other parts of Europe, colder climatic conditions prevailed in Sweden and Scania for most of the second half of the

16th century and throughout most of the 17th century. The period c. 1560-1630 was particularly cold and experienced overall increased climatic variability (see Fig. 1). In the 1690s, there was also recurrent span of cold years with late springs in the Baltic area, culminating in the disastrous years of 1695-1697 leading to mass mortality throughout the region and especially in northern Sweden, Estonia and Finland (Dribe *et al.*, 2015; Lilja, 2008). Reconstructions of ice-winter severity from the western Baltic indicate that the period experienced greater volumes and persistence of winter ice compared to preceding and

subsequent periods, and the Sound between Scania and Zealand was covered with ice for most of the years 1694-1698 (Speerschneider, 1915; Koslowski & Glaser, 1999).

During portions of the 18th century, there was a 'return' to milder temperatures, albeit with some notable exceptions with especially cold periods in the early 1740s and 1780s. The most notably challenge in terms of natural conditions pointed out in

previous research is the increasing degree of sand drift and soil erosion in Scania during the later parts of the 18th century and early 19th century (Mattsson, 1987). As Bohman (2017a & 2017b) has shown, these agro-ecological crises were mostly local and temporary, counteracted by land management policies at the local and regional level. The main causes behind the increasing soil erosion and sand drift has been framed as anthropogenic, through deforestation and intensified land use practices. Mattsson (1987), relying on instrumental and observational meteorological records from Lund, argued that another

underlying factor behind these agro-ecological issues was climatic variation in the form of the generally colder conditions during the Little Ice Age and increased heavy winds and storms, particularly easterlies, during the latter half of the 18th century.

In the following century, the 1810s and the 1840s stand out for being cold (Tidblom, 1876; Cappelen *et al.*, 2019). In general, the 19th century was one of great transformation and expansion for agriculture in Scania making it difficult to identify any

prolonged climatic periods that were beneficial or detrimental for agriculture. Nonetheless, there certainly were years that

---

[1] The Scanian landscape has also undergone change over time, particularly during periods of land reclamation when forests and wetlands have been converted to arable lands.

experienced particularly bad agro-meteorological conditions like summer droughts, for example in 1811, 1822, 1826, 1837, 1868, 1870 and 1899 (Tidblom, 1876; SMHI, 2021a). The 1868 summer drought was particularly bad since it followed a year of severe crop failures in northern Sweden that had already depleted much of the grain stocks available for aid (Dribe *et al.*, 2015; Västerbro, 2018). According to Utterström (1957) and Edvinsson *et al.* (2009), lack of precipitation and drought during the summer were the main agro-meteorological risks in southern Sweden in the 18[th] and 19[th] centuries.

[Figure 1 is somewhere around here]

## 1.2 Farming in Scania

Descriptions of agriculture in Scania during and subsequent to the study period have relied on the ethnographic and geographical categorizations made by Campbell (1928), who outlined three different types of farming districts: the plain, the intermediate (or "brushwood", sw. *risbygd*), and the forest districts (Dahl, 1989; Svensson, 2013). Villages in the plain districts generally practiced a three-field farming system and were characterized by their specialization in grain production (Campbell, 1928). In the intermediate districts, villages had a higher share of livestock production and often practiced a one-field farming system (Bohman, 2010). Finally, the forest districts had the most diversified economy, where handicrafts and forest-related industries complemented grain and livestock production (Svensson, 2016). Bohman (2010) estimated that during the 18[th] century and the first half of the 19[th] century, crop production constituted roughly 90 % of the total production value in the plains district and somewhere around 80 % in the intermediate and forest districts. Controlling for price changes, crop production increased its share of overall production value at least until the 1860s.

Despite that the types of farming districts varied in their respective specializations, practically all farming in Scania was performed in a mixed farming system, where livestock husbandry and grain production were integrated and mutually dependent (Bohman, 2010; Myrdal & Morell, 2013). Until the 19[th] century, most farms in Scania belonged to a village where farming operated under an open-field system (sw. *tegskifte*) with a mixture of private and communal management. Limited enclosure reforms, *storskifte*, were introduced starting in 1757, followed by radical enclosure reforms in 1803 (*enskifte*) and 1827 (*laga skifte*). These latter reforms involved the breakout of the individual farms from the communal management, effectively privatizing land ownership and management. Implementation of these reforms was gradual and intermittent (Gadd, 2011; Gadd, 2018). Hence, for large parts of the study period, decision-making regarding grain production was largely mediated through the institutions of the village.

### 1.3 Grain crops

Rye, barley and oats dominated the composition of grain production during the study period and had done so since the Viking Age, albeit with much internal variation over time. For example, oats production saw a large increase in its share of overall grain production during an export boom in the 19th century (Welinder, 1998; Bohman, 2010).[2] In the late 19th century and early 20th century, the new so-called improved cultivars (mainly in the form of autumn-rye and autumn-wheat) increasingly took the place as the most dominant grain crops (Leino, 2017). In previous research, the type of farming district and soil types has been

seen as the primary factors determining differences in crop composition (Dahl, 1942; Dahl, 1989).[3] Given their historical importance, this study will mainly be limited to analyzing the production of barley, rye and oat varieties. Wheat varieties will also be included in the later study period (1865-1911).

*Rye varieties*

Leino (2017) has studied some of the historical grain varieties in Sweden. Examples of rye varieties prevalent in Scania were

late-rye (sw. *senråg*), autumn-rye (sw. *larsmässoråg*), sand-rye (sw. *sandråg*) and spring-rye (sw. *vårråg*). Swidden-rye (*svedjeråg*) was most likely also grown, especially in the forest districts. Scanian farmers preferred to grow their rye on sandy soils or other well-drained soils (Dahl, 1989; Gustafsson, 2006). Leino (2017) notes that in historical sources, late-rye is often characterized as allowing very late sowing, all through December in Scania (in some extreme cases this nominally autumn crop was apparently sown in early spring). According to Leino (2017), this type of late sowing of late-rye offered the possibility

to incorporate autumn-rye into a two- or three-field system without the need for a full year of fallow after the preceding harvest. This somewhat blurred line between spring- and autumn-rye is consistent with genomic studies of Scandinavian rye landraces (Hagenblad *et al.*, 2012). More detailed sources on sowing dates are difficult to find. By all accounts sowing dates varied locally. In parish descriptions from Malmöhus county in the early 19th century, sowing dates for autumn-rye vary from the middle of August until early October, although the most commonly noted sowing period was the latter part of September

(Bringéus, 2013). Spring-rye appears to have been sown after barley, sometime in May, depending on the village. Autumn-rye is noted to have been harvested earlier than other crops, although all harvesting of rye is reported in either late July or August.

In a broader context of European rye landraces in the pre-1900 period, Fennoscandian landraces have been found in genomic studies to belong to a particular and separate meta-population of rye landraces, distinct from landraces in continental Europe.

---

[2] While the sources seldom allow for further details, there were many varieties of each grain crop. These different varieties could sometimes vary quite starkly from each other in terms of their characteristics (Leino, 2017).

[3] Wheat and barley were more dominant in the arable plain districts. The share of oats was lowest in the forest districts and the share of rye was roughly the same in the different types of districts. Regarding soil types, Bohman (2010) found that barley and oats (and wheat) were more dominant on high quality soils and that rye was more common on poorer soils. The relationship between vegetable production and animal production shifted over time, with an increasing share of vegetable production throughout the 18th and the first half of the 19th century, varying from about 64 % to 97 % of the overall production value (Bohman, 2010).

Furthermore, even southern Scandinavian rye landraces have been found to have more in common genetically with landraces

from northeastern Europe rather than those from maritime western Europe (Larsson *et al.*, 2019).

*Barley varieties*

*Southern six-row barley* (sw. *sydsvenskt sexradskorn*) was common in Scania, especially in the forest districts, even in the late 19th century. It was sown late, often well into June, due to its sensitivity to frost and its rapid growth, allowing ripening despite late sowing. Two-row varieties like *Scanian two-row barley* (sw. *skånskt tvåradskorn*), was also grown, at least during the 19th

century but probably earlier as well. Two-row varieties required more intensive agricultural practices, longer growth periods and richer soils, but offered better resistance to frost and often gave larger yields, compared to six-row varieties. In the previously mentioned parish descriptions from the early 19th century, sowing dates for barley range from late April to early June, while harvesting is mostly described as taking place sometime in August (Bringéus, 2013).

Similar to rye, genomic evidence on barley landraces from Scandinavia and southern Scandinavia in particular, indicate spatial

and temporal consistency from the 17th century up until the late 19th century. (Lundström *et al.*, 2018). A distinctive feature of these Scandinavian barley landraces in terms of genetic markers is the prevalence of the non-responsive ppd-h1 allele, which prolongs the flowering during periods of increasing daylight, prolonging the vegetative state and potentially increasing yields in cooler and wetter conditions (Jones *et al.*, 2012; Aslan *et al.*, 2015). It has been suggested that selection and maintenance of barley seed with this particular allele was part of a long-term adaptation process by early farmers (Cockram *et al.*, 2007).[4]

*Oat varieties*

Historical oat varieties can be grouped into two broad categories: white oats and black oats. Generally, white oat varieties were grown on poorer, and especially wet, soils. According to Campbell (1950), they were better suited for making bread compared to the fodder-oriented varieties that became increasingly more common during the course of the 19th century. Black oat varieties were more resistant to droughts and were preferably grown on richer, manured soils. Campbell (1950) argued that *Nordic*

*White oats* (sw. *nordisk vithavre*) was the most common variant in Scania. It is more uncertain whether black oat varieties were grown, although they were grown in all neighboring provinces (Halland, Blekinge and Småland), which suggests, together with the fact that there was a widespread trade in seed-grains, that black oats were at least grown locally and intermittently (Campbell, 1950; Leino, 2017). Oats appear to have been sown about 1-3 weeks prior to barley and spring-rye, and harvested at about the same time as, or shortly before, barley (Bringéus, 2013).

According to Dahl (1942), oat farming in Scania was *not* an adaptation to local climate like in parts of northwestern Europe. Rather it was other natural conditions, primarily the type of moraine soil common in some areas around the Baltic like Denmark, Scania and northern Germany (sw. *baltisk morän*), as well as local hydrological conditions, namely on soils that

---

[4] An allele is one of several possible expressions of a given gene. The ppd-h1 allele is the non-response expression of the ppd-H1 (Photoperiod-H1) gene (Turner *et al.*, 2005).

were poorly drained that was decisive for oat cultivation. It is important to note that Dahl (1942) conceptualized natural conditions and climate as something static, and that the only secular changes that occurred in natural conditions were due to human intervention, for example by not investing in drainage or through over-cropping.[5] However, given that the climate actually varied over time, one would expect climate effects interacting with factors like soil and the type of cultivated crop. For example, periods of a wetter climate should have had more negative impacts on crops grown on poorly drained soils, whereas crops cultivated on well-drained soils should have been more exposed to drought periods (Osvald, 1959; Weil & Brady, 2017).

## 1.4 Crop diversity, resilience and adaptation

This brief overview of the diversity of grain varieties to be found in early modern Scania suggests a flexible farming system in terms of sowing and harvest dates as well as the ability to produce grains under differing agrometeorological conditions. It is important to note in this context the inherent capacity of crop varieties to adapt to local environmental conditions (including local farming practices), that over time should have led to much greater variety, and indeed resilience, than this brief overview suggests (Leino, 2017, Aslan *et al.*, 2015). In the context of historical grain production, I define resilience as the ability of a production system to maintain itself over a longer period through a combination of biological and institutional flexibility and durability in the face of a variable environment. When discussing adaptation I refer to how a given crop or farming practice performs in a given set of environmental circumstances. I subsume the concept of exaptation (passive or accidental adaptation) under adaptation, given the difficulty in disentangling the two. For example, a particular crop may perform better during colder periods, increasing the production of the crop, which could be due to farmers *actively* adapting to changeable circumstances or the crop being more adapted (passively) relative to the other crops being cultivated. Moreover, even if farmers are actively increasing the production of a given crop, it can still be very difficult to establish whether it is due to adaptation to environmental change, a response to shifting market demands, technological innovation or cultural trends.

Previous research has stressed that, at least in relation to climate 'extremes', a diversified crop production including both spring and autumn crops of different varieties was more resilient in areas of Europe north of the Alps (Michaelowa, 2001; Ljungqvist *et al.*, 2021a). Michaelowa (2001) partly blamed the excess specialization towards autumn-wheat for the relatively poor performance of French agriculture compared to English agriculture during the 18th century, where the latter was more diversified, cultivating autumn-wheat, autumn-rye as well as spring-barley and oats.[6] Utterström (1961) and Michaelowa (2001) argued that colder periods in the early modern period, specifically in the late 17th and 18th centuries, led to reductions

---

[5] Dahl (1942) explained changes in the prevalence of oats by trends in the economy, where oats as a less prioritized crop suffered in times of war or scarcity and increased in 'good times' when there was more labor power and manure available. Periods of scarcity or plenty seemed to be something external to the agricultural economy in Dahl's perspective, caused by either war or some other unknown external factor. In general, Dahl (1942) had a static perspective of agriculture in the times before enclosure, were there very little change across the centuries and farmers and villagers adhered to more or less set farming and cropping practices.

[6] Allen (1992) notes that wheat and pulses expanded in the 18th century at the expanse of rye, barley and oats.

in livestock production in France, England and Sweden, and in response grain production usually increased with the intention to fill in the nutritive gap. [7] If such adaptations were took place, they must have been difficult to implement in the short term and probably also insufficient given that grain production was also vulnerable to spats of cold weather. Pfister (2005) showed how cold and wet conditions during the different seasons of the year were detrimental to livestock production in the Swiss Alps as well as the difficulties of the local communities to adapt given that the cultivated grains and vines were also vulnerable to cold and wet conditions. Grain shortages, sometimes resulting in famines, were common in many parts of Europe up until at least the 19[th] century (Appleby, 1980; Dribe *et al.*, 2015; Esper *et al.*, 2017).

There have been attempts to detail the relationship between grain production and climate variability in northern Europe during the early modern period in more detail. Brunt (2004) found that English wheat yields during the 1770s were mainly sensitive to temperature and to a lesser extent precipitation, depending on local soil conditions. Especially important were summer temperatures. Cooler summer temperatures through the month of July benefited wheat yields, supposedly by prolonging the grain-filling period. A warm and dry August was then beneficial by allow the crops to dry for the coming harvest. Ideally, rainfall would be spread out over many days during the early summer months. Concentrated rainfall during a short time-span risked ruining the crop, the harvest month of August being especially vulnerable (Brunt, 2004). In a later study, Brunt (2015) found that wheat yields were significantly affected by weather shocks throughout the ca 1690-1850 period, with the 19[th] century largely conforming to the 1770s as to the effects from temperature and precipitation during summers.[8]

Pei *et al.* (2016) studied the relationship between yield ratios and temperature at a continental scale and proposed that European farmers during the period c. 1500-1800 used crop management as a mechanism for climate adaptation. Specifically, farming systems drifted towards increased rye production during colder periods, which the authors argue was a more cold-resistant crop. In an earlier study Pei *et al.* (2015) asserted that extensification of land use was the most prominent strategy in mitigate climatic stress during the same period. However, given differences in soil, climate, available grain varieties and other factors, it seems more reasonable to expect more heterogeneous and contextually dependent adaptation practices at the local and regional level (van Bavel *et al.*, 2019; Ljungqvist *et al.*, 2021a). While rye almost certainly was more cold resistant than wheat, in relation to oats the same seems to be true only when we exclude wetter climatic areas (e.g. western Sweden or parts of Scotland). In relation to barley there is limited evidence indicating that rye *overall* was the more cold-resistant grain. For example, in northernmost Sweden and Finland grain production was limited almost exclusively to barley. An important caveat in making these type of comparisons is the fact that rye was mostly grown as an autumn-crop whereas barley and oats were exclusively grown as spring-crops and that in many agricultural areas of northern Europe they were more often supplementary than rival crops (Huhtamaa & Helama, 2017b).

---

[7] According to Bohman (2010), the increasing share of grain production in 18[th] century Scania was driven by price trends.
[8] Brunt (2015) further found that the effects from weather shocks were large enough to obfuscate long-term productivity trends in subsequent yield estimations, especially in the 1690s and late 1850s.

Considering Sweden, and southern Sweden in particular, one finds a composition of grain production that was diversified, comparable to that of England in the 18[th] century described by Michaelowa (2001), with the important exception of wheat that in Sweden was only a marginal crop. Utterström (1957) argued that for grain production northern Sweden, temperature was the most important climatic variable, whereas it was precipitation for southern Sweden. Using more up-to date climate and grain harvest data, Edvinsson *et al.* (2009) largely confirmed the stipulations made by Utterström (1957), at least from 1724 up until the late 19[th] century, finding a negative association between subjective harvest assessments and June and July temperatures and a positive association with precipitation in the same months and November and December temperatures. After c. 1870, Edvinsson *et al.* (2009) found a shift in the relationship. Precipitation in the summer, including the month of May, was still positively correlated with the harvest assessments. However, summer temperatures were no longer statistically significant, whereas all the four first months of the year (JFMA) showed positive associations with harvests assessments. A short digression is in order here. Compared to summer and spring temperatures, relationships with winter temperatures are more difficult to explain, given that they are indirect, occurring before the growing season for spring crops. Temperature and precipitation during the winter months do affect the overwintering autumn-crops by facilitating or inhibiting the survival of the grains themselves, as well as fungi, soil bacteria and other various grain pests (Holopainen & Helama, 2009; Osvald, 1959). In addition, the nutritive balance of the soil is affected (Aðalsteinsson & Jensén, 1990). Again, it is quite difficult to establish, both empirically and theoretically, the mechanisms and links between these relationships and the subsequent grain harvest. It should be noted that these 'indirect' effects are also at play during the other seasons. For example, de Vries *et al.* (2018) found that summer droughts have different effects on soil bacteria and fungus, and that these effects have long-term consequences for vegetation growing on the soil.

Returning to the discussion of the results from Edvinsson *et al.* (2009), they argued that the overall relationship between climate variability and grain harvests was weak, partly explained by the lack of detail in climate data. Furthermore, they found that the magnitude of the relationships increased in the period 1871-1955 compared to the previous roughly 150 years, which they primarily explained in terms of the increasing shift towards higher yielding and more temperature sensitive wheat production. More controversially, they also hypothesized that climate variability itself was less important to harvest in pre-industrial agriculture, due to chronic seed shortages and a more risk-averse behavior on the part of farmers.

With respect to the differences between different grains, Edvinsson *et al.* (2009) employed aggregate official statistics at a national level for the period 1803-1955 (with a gap between the years 1821 and 1859). They found that wheat and rye harvests were positively correlated with October through April temperatures, that barley harvests was positively correlated with temperatures in April, May and August-September and finally that oat harvests were negatively correlated with June-July temperatures. Harvests of all the mentioned grains were positively associated with increased precipitation in May through July. Wheat and rye harvests were negatively associated with increased precipitation in March, whereas the same was true for barley and oat harvests in relation to precipitation in September. These results are possibly skewed towards the late 19[th] century and especially the first half of the 20[th] century considering the gap between 1821 and 1859 as well as the dramatic shifts in the

types of cultivated grain varieties that Sweden underwent in the late 19[th] century (Leino, 2017). Beside the study from Edvinsson *et al.* (2009), Palm (1997) tried to estimate the relationship between the yields of various grains at a farm in Halland between c. 1750-1870, with limited results.

The division of Sweden into a southern and northern half in regards to the main agro-meteorological constraints for agriculture and grain production made by Utterström (1957) and later affirmed by Edvinsson *et al.* (2009) arguably needs to be complemented. As discussed by Ljungqvist & Huhtamaa (2021), the relationships between climate and agriculture in Scandinavia vary from region to region. From an agronomic perspective, southern Sweden is a diverse place in terms of natural geography. Northwestern Scania and the provinces further north on the west coast (Bohuslän, Halland, Västra Götaland) are

wetter and colder than most of Scania, whereas most of the east coast of southern Sweden is drier (especially during spring and autumn) and experiences on average a few hundred extra hours of sun each year (Persson *et al.*, 2012). As mentioned previously, Scania stands out relative to the rest of southern Sweden in terms of the duration of the growing season (Osvald, 1959). Ljungqvist & Huhtamaa (2021) propose that in neighboring Denmark hosting similar conditions for grain production, the shortening of the growing season during wetter and/or colder years might not have led to major issues related to frost.

Important additions to these considerations of natural geography is the question of what type of grain varieties were cultivated, and in what type of farming system.

In the following sections, I attempt to estimate the relationship between climate variability and grain production in Scania during the period 1702-1911, divided into an early study period (1702-1865) and a later study period (1865-1911). First, I describe and discuss the sources and methods employed, followed by a presentation and a discussion of the results. Finally, I

conclude the article by interpreting and contextualizing the obtained results.

## 2 Sources and methods

### 2.1 Sources on agriculture and grain production

Scania stands out in a Swedish context regarding the availability and extent of a specific historical source material, namely the priestly tithes (*sw. prästetionde)*, which in many parts of Scania remained flexible and proportional to output throughout the

285 18[th] and 19[th] centuries (Olsson & Svensson, 2010). Using surviving tithe records from 36 parishes in Scania, Olsson and Svensson have produced a database, the *Historical Database of Scanian Agriculture* (HDSA), with roughly 85,000 unique farm level observations covering the period 1702-1881, where one observation is one farm's production in one year (Olsson & Svensson, 2017b). The structure of the HDSA is that of an unbalanced panel and includes besides production data on crops and animals, data on farm size and household characteristics, land tenure and other institutional factors, soil quality and land

size, geographical factors as well as relative crop and animal prices.

The soil data in the HDSA is based on modern soil grading. Previous historical studies have relied on modern soil grading, arguing that it better captures the 'natural' fertility of agricultural landscapes compared to those found in historical sources. For instance, Bekar (2004) used national survey data from the 1950s and 1960s in his study of historical grain production in England.[9] Bohman (2010), used data from Göransson (1972) who performed a local study of soils in Scania based on the gradient system established by the Swedish national soil survey published in 1971 (Lantbruksstyrelsen, 1971). The data from Göransson (1972) that was subsequently been incorporated into the HDSA. The soil grading system was based on 10 levels, where 1 denotes the lowest and 10 the highest quality soils. According to the national survey, Scania was the only region in Sweden with grade 10 soils (Lantbruksstyrelsen, 1971).

Constructing grain production series for the earlier period 1702-1865 from the HDSA involves attempting to solve some issues. Firstly, there is an issue related to how the tithe was collected, i.e. that it was collected before threshing, and the amount of seed that was obtained by threshing the same type of grain differed across parishes and farming districts. Therefore, all crop production series are adjusted to local threshing coefficients, based on actual threshing accounts in different parishes, in line with Olsson & Svensson (2017a).

Secondly, there are issues of non-stationarity in grain production time-series, particularly in the 18th century and beyond, requiring de-trending methods in order to obtain reliable and linear estimations of relationships (Jörberg, 1972; Huhtamaa, 2015; Shumway & Stoffer, 2017). At the same time, detrending risks removing information related to the long-term effects of climate variability on grain production (see Esper *et al.*, 2017 and Ljungqvist *et al.*, 2021 for a discussion of this in the context of historical grain prices). I estimate *normalized production anomalies* (NPA) in line with Beillouin *et al.* (2020), employing a locally weighted scatterplot smoothing (loess) for each grain as well as total grain production in the HDSA.[10] A common smoothing span of 0.25 is used for all series. The formula for the NPA is:

$$\tilde{a}_t = \frac{(Y_t - \mu_t)}{\mu_t}, \tag{1}$$

where $\tilde{a}$ is the normalized production anomaly for a given grain in a cluster or aggregate region at each *t* year. $Y_t$ is the average of the observed annual production outcome for the specific grain. $\mu_t$ is the expected production outcome according to the loess fit.

Thirdly, there is the issue that the HDSA panel is *unbalanced*. Similar to, for example, tree ring-based temperature reconstructions, where the number of tree rings available for the reconstruction usually decline further back in time (Esper *et*

---

[9] While the national soil surveys in Sweden and England had quite similar aims, to evaluate and map the 'natural' fertility of soils, they differ in terms of their criteria. The English national soil survey was predominantly based on geological and climatological indicators, while the Swedish survey was based on a mix of geological, yield and price data as well as local expertise (Lantbruksstyrelsen, 1971; Gilg, 1975). Most European countries carried out similar national surveys in the decades following the 1950s (Jones *et al.*, 2005).

[10] Beillouin *et al.* (2020) use the term *normalized yield anomalies* (NYA), but here I am mainly relying on production or harvest data I substitute the term yield with the term production to avoid confusion, given the importance of distinguishing the harvest from yield in general when discussing agricultural production.

*al.*, 2016), the number of farms in the HDSA is lower in the early decades of the 18[th] century (the number of farms also goes down in the final decades of the database coverage). This introduces an increased risk of sampling bias. This problem is partly counteracted by the loess detrending and partly by clustering the data into most-similar clusters using Hierarchical Cluster Analysis, described in Section 2.2.

Grain production data is only available up until 1865 in the HDSA. After 1865 there is instead official statistics on grain production on the county and parish levels, based on reports from the local rural societies (sw. *Hushållningssällskapen*) up until 1911 (BiSOS, 1865-1911). I rely on county-level data only. The 19[th]-century Swedish official statistics has been subject to some important criticisms. The manner in which the data was collected varied to some extent locally as it was up to the local representatives in the rural societies to establish data collection procedures (Svensson, 1965). This is less of an issue considering that I do not compare different parishes with the BiSOS data. Moreover, it is commonly argued that total crop production and the amount of arable area is systematically underreported and underestimated in the official statistics. Again, this is not an issue to the extent that I am principally interested in the variations in output over time that is associated with climate variability. There are no obvious reasons to suspect that this part of the variation in output is related to the general underestimation in official statistics. BiSOS data is de-trended applying the same technique as for the HDSA, namely with estimations of NPA (see Eq. 1).

## 2.2 Clustering

Considering the institutional and geographical diversity of Scania, aggregating the data risks masking location or type specific relationships between agriculture and climate, or conversely that some localized trends distorts the overall picture. To homogenize data from the HDSA, obtain clearer signals and to reduce the risk of introducing a geographical or institutional bias by grouping the data by parish, type of farming district or cadastral status, all villages in the sample are divided into three different clusters using an hierarchical cluster analysis (HCA).

HCA is an algorithmic-based method that cluster the data into 'most-similar' groups based on a chosen parameter in the data, in this case threshing-adjusted rye production over time on the village level (see Section 2.1). Rye was one of the two most important grains during the study period, and since it was mainly grown as an autumn-crop it required some specific management practices at the village level, and therefore serves as a more appropriate distinguisher than barley or oats. I use an agglomerative HCA, where each village initially forms a cluster by itself, pairing up with other villages as the hierarchy 'moves up' (Day & Edelsbrunner, 1984). Distances between groups are estimated using the Ward's D method and Euclidean distances.[11] Since in HCA the true or optimal number of clusters is not known and the number of clusters is therefore determined by some metric or criteria determined by the researcher. Multiple such criteria have been suggested in the literature,

---

[11] Euclidean distance is the straight line between two points in classical metric space (Howard, 1994).

often relating to the largest visible or measurable distances between the branches in the cluster dendrogram. One of the most common such metrics is the *gap statistic* (Tibshirani *et al.* 2001). Specifically, I use the clusGap function in the factoextra package in *r*, setting the maximum number of potential clusters at 10 and running 100 bootstrap samples (Kassambra & Mundt, 2020).

Some descriptive and interpretative issues come with this approach. If a resulting cluster consist of several different types of farming districts and administrative units, e.g. parishes, are separated into different clusters, describing, interpreting and contextualizing results become difficult. In order to reduce the descriptive and interpretative issues related to the HCA, the clustering results thus obtained are contextualized using the most common historical categorizations found the historical literature when discussing regional specialization in agricultural production, namely the type of farming district (see Section 355 1.3). Clusters are also described in term of cadastral status and soil characteristics using information available in the HDSA.

## 2.3 Grain production clusters

Fig. 2 shows the cluster dendrogram obtained using the method described in Section 2.2, cut at three clusters. Describing these clusters in terms of historical and geographical categorizations, some distinguishing patterns emerge.

[Figure 2 is somewhere around here]

Fig. 3 reveals that all clusters are represented by villages in the northernmost parishes of Hjärnarp and Tostarp, the forest and mixed farming districts centered around Billinge and Kågeröd parishes as well as the parishes located in the forest and mixed farming districts around lake Vomb in southern Scania. Cluster 1 is most heavily represented by the parishes in the proximity of Billinge and Kågeröd as well as around lake Vomb. Cluster 2 is the most geographically spread, covering all of the areas of 365 the total sample, except the plain district parishes around Malmö and Lund. Finally, Cluster 3 is mostly concentrated on the parishes around Malmö and Lund and with some villages in parishes around Röstånga and Kågeröd as well as around the southern edges of lake Vomb.

[Figure 3 is somewhere around here]

Tab. 1 shows the outcome from the clustering in terms of proportion of arable in each grade. Cluster 1 has the largest share of the low quality soils (grade 1 to 4), roughly 39 %, as well as the least amount of high and moderately high quality soils (grade 7-8 and 9-10). Cluster 2 has the largest variance in terms of shares in different types of soils as well the largest share of

moderately high quality soils, ca 54 %. Finally, cluster 3 has the largest share of the highest quality soils, 16 %, as well as the largest amount of moderate soils, 44 %.

[Table 1 is somewhere around here]

Fig. 4 below shows the amount of villages from each type of farming district in the three clusters as well as the institutional make-up in terms of property rights regimes of each cluster (i.e. freehold land owned and managed by peasant-farmers, crown land owned by the state but managed by tenants, and manorial land owned by the nobility but managed by their tenants). Cluster 1, is more mixed, with farms in all three different types of farming districts, albeit with most farms in the intermediate and forest districts, with a moderate share of peasant-owned and managed farms. Cluster 2 has the largest amount of manorial farms and almost all farms are located in the intermediate and forest districts. The largest amount of plain districts farms can be found in Cluster 3, which also has the largest amount of crown and peasant-owned farms. Furthermore, Cluster 3 contains almost no intermediate districts farms, and a moderate amount of farms in the forest districts.

[Figure 4]

In terms of grain production, Cluster 3 has the largest average production of all grains over time as well as the largest share of rye and barley in its production, which is not surprising given that it has the largest share of plain district villages as well as the largest share of highest quality soils, as shown in Fig. 5. Cluster 2 has similar average production levels as Cluster 3 in the first decades of the 18th century, followed by a slight stagnation for the rest of the century, followed by large increases in production of all grains and in particular oats during the first half of the 19th century. The cluster with the lowest quality soils, Cluster 1, also has the lowest average production levels, although it shows continual increases throughout the period 1702-1865.

[Figure 5 is somewhere around here]

To summarize, Cluster 1 is institutionally mixed and has the lowest quality soils, Cluster 2 is more manorial, has the largest share of soil grades 6-10 of all the clusters and is the most geographically spread cluster. Finally, Cluster 3 is mostly peasant-

owned or managed, has the largest share of the highest quality soils (grade 8-10) and lands in the plain districts, notably in the plains around Lund. Average production levels increase in an ascending order from Cluster 1 (the lowest) and Cluster 3 (the highest), although there is some variation over time.

## 2.4 Sources on the climate

Temperature is one of the most important agro-meteorological indicators, especially during the growing season. Instrumental temperature measurement data are available from the city of Lund from year 1753. The series contains gaps and has several noted inhomogeneities in the form of instrument relocations, instrument replacements and changes in observers (Tidblom, 1876). For the purposes of this study, the temperature series was homogenized and gaps where filled using an Adapted Caussinus–Mestre Algorithm (ACMANT) software relying on the target unhomogenized temperature series as well as a network of homogenized temperature series (Domonokos & Coll, 2017). The homogenization procedure is further detailed in Appendix A. Results of the homogenization procedure are also presented and discussed in Section 3.2 and Section 3.4. I use a homogenized monthly temperature time series centered on Lund from 1702-1865, including infilling of gaps between 1702-1752 and 1820-1833.

I also use precipitation data from Lund, available from 1748, as well as the number of rainy days per month (Tidblom, 1875). Hydroclimate is much less spatially coherent compared to temperature; hence, a similar homogenization approach in line with the temperature series is not suitable. Therefore, I supplement the instrumental precipitation data with regional hydroclimate reconstructions. I use three hydroclimate reconstructions. The first, from Cook *et al.* (2015), is a Palmer Drought Severity Index (henceforth, PDSI) reconstruction from the Old World Drought Atlas, covering the entire study period, where I use the grid cell centered at 55.75°N, 13.75°E, roughly corresponding to east-central Scania. The second is a reconstructed Standardized Precipitation-Evapotranspiration Index (henceforth, SPEI) for southern Scandinavia compiled by Seftigen *et al.* (2017), which also covers the whole study period (see Fig. 6). These two reconstruction are independent, being based on mutually exclusive data. The third and final hydroclimate reconstruction is a May through July precipitation reconstruction (henceforth, MJJ*pr*) by Seftigen *et al.* (2020) based on the wood densitometric indicator referred to as blue intensity (BI), covering the period after 1798. The second and third reconstructions are not strictly independent given that they are based on same tree ring data, although they are extracted using different methods and the MJJ*pr* is more oriented towards capturing high-frequency variability (Seftigen *et al.*, 2020).

[Figure 6 is somewhere around here]

For the period after 1865, I use monthly average, minimum and maximum temperatures and monthly accumulated precipitation instrumental data, also from Lund, available at the Swedish Meteorology and Hydrology Institute (SMHI, 2021), as well as the hydroclimate reconstructions mentioned above. I also use daily air temperature data from Lund during the period 1863-1911 to calculate the average occurrence of the first autumn-frost and last spring-frost. Given that ground temperature

can vary from air temperature, I use a slightly conservative estimate where days with 1 degrees C or less in average temperature between January and May and considered as days with spring-frosts, and between June and December as autumn-frosts. These estimates are then compared to estimations made by SMHI for the period 1960-1990.

Some studies employ climate variables based on annual change, month-to-month changes or anomalies from some long-term or moving trend when estimating the relationship between historical grain production and climate (Brunt, 2004; Edvinsson *et al.;*Bekar, 2019). However, the evidence on any potential information added by increasing the complexity of the climate variable involved has been limited (Vogel *et al.*, 2019a; Vogel *et al.*, 2019b). Thus, I follow the example of Beillouin *et al.* (2020) and use 'simple' climate variables.

## 2.5 Estimating the relationship between grain production and climate

The main analysis is based on cluster-wise Pearson correlation analysis of pairs of variables. I estimate correlation coefficients between annual normalized yield anomalies of rye, barley, oats, total grain production and climatic variables on a monthly and seasonal basis, using HDSA for the early study period 1702-1865 and the BiSOS data for the later period 1865-1911. Given that harvesting was usually completed in late August or during September, I have used lagged (i.e. the values from the previous year) climatic variables for the autumn and early winter months (OND). In addition, I estimate the same relationships during drier and wetter years, respectively. Wet and dry years for the HDSA are defined according to the 33th (dry) and 67$^{th}$ (wet) percentiles of the SPEI during the period 1651-1951. For the HDSA period, this translates to 58 dry years and 55 wet years (see Tab. B1 and Tab. B2 in Appendix B). Due to the low *n* in the later period 1865-1911 ($n = 47$), I split the data into two halves, each representing the lower (drier, $n = 23$) and higher (wetter, $n = 24$) halves of the SPEI during those years (see Tab. B3 and Tab. B4 in Appendix B).

## 3 Results

### 3.1 Estimations of past temperatures and frequencies of growing season frosts in Scania

In this section, I present the results from the estimations of two agro-meteorological indicators, both specifically related to temperature. First, I present the main results from the monthly temperature series homogenization for the early study period are outlined. Second, I describe the results regarding the average occurrence of the first autumn-frosts and last spring-frosts during the late study period.

### 3.1.1 Monthly temperature series homogenization

After homogenization of the Lund monthly temperature series, the largest corrections occur during the summer (JJA) and autumn (SON) months, where temperatures are adjusted upwards, especially during the late 18th century (see Fig. 7 and Fig. 8). There is a slight tendency for upwards adjustments for the spring (MAM) and winter (DJF) temperatures as well, although it is comparably small. Several breaks were detected in the temperature series, almost exclusively at points in time when there was a change in observer and change in observation location (see Appendix A). The largest breaks occurred in the late 18th century, and smaller inhomogeneities were detected throughout the early instrumental period, except for the two earliest decades, 1753-1774, when no significant breaks were detected. The homogenization process is discussed in more detail in Appendix A.

[Figure 7 is somewhere around here]

[Figure 8 is somewhere around here]

### 3.1.2 Average occurrence of the first autumn-frost and last spring-frost, 1863-1911

The average date for the last spring frost during the 1863-1911 period was the 15th of April. In almost half (44%) of these years, there were no occurrences of temperature measurements below 1 degrees C. Only two years experienced late spring frosts in May, namely the years 1864 and 1867, of which the latest estimated spring frost was the 24th of May in 1867, a year which is known for its exceptionally cold spring (Västerbro, 2018). The average date for the first autumn frost was the 8th of November. The earliest estimated autumn frost occurred on the 16th of October 1879. These dates are similar to those estimated by SMHI using data from the 1960-1990 period, where the average date for the first spring frost was between the 1st and 15th of April in the western and south-western edges of Scania, between the 15th of April and 1st of May for the rest of western and southern Scania, and finally between the 1st and 15th of May in northern and north-eastern Scania (SMHI, 2017a). The average date for the first occurrence of autumn frosts follow a similar geographical pattern, e.g. between the 15th of November and the 1st of December for the western edges of Scania to between the 1st and 15th of October for the northernmost forested areas bordering Småland (SMHI, 2017b).

### 3.2 Relationship between climate variability and grain production

In this section, I present the correlation results between climate and grain production indicators in the early and late study periods, respectively. Furthermore, I present correlation results of restricted samples where years with dry or wet summers are analyzed separately.

### 3.2.1 Grain production and climate variability 1702-1865

During the bulk of the period, 1702-1865, there is a negative association between summer temperatures and grain production in all three clusters, with July and June producing the strongest signal, shown in Fig. 9. May yields low negative correlation coefficients for rye in Cluster 1 and Cluster 3, and August yields a low negative coefficient for barley in Cluster 1. Overall, the signal obtained from the oats series is weak and mostly divergent from the other grains, except for its strong positive association with a higher SPEI, i.e. wetter summer conditions, which it has in common with the other grains. The seasonal temperature indicators largely correspond to monthly indicators, with JJA consistently showing strong negative associations with total grain production in all clusters. Except a low positive association between oat production and spring temperatures in Cluster 3, monthly and seasonal temperature indicators for the spring and autumn gives almost no statistically significant results.

The results revealed by Fig. 10 for monthly summer precipitation show, inverse with those of temperature and hydroclimate, strong positive correlations with grain production. In addition, there are more differences between the various clusters and grains. There are no significant correlations between rye production and the instrumental summer precipitation variables, although in Cluster 2 and 3 there is a positive correlation with reconstructed MJJ. On the other hand, there is a negative association between January (and February, in Cluster 1) precipitation and rye production in all clusters. Barley, oats and total grain production all show large positive correlations with June and July precipitation and with reconstructed MJJ. The number of rainy days in June and July produce a strong positive signal in relation to almost all grain production, and especially barley production in Cluster 2 and 3. There is also a positive, albeit weak, association between the number of rainy days in May and for oat production in Cluster 2 and rye production in Cluster 3.

Overall, the magnitude of the correlations are similar across grains and climate indicators, except correlations involving barley production that provides the strongest effects. Cluster 3 stands out in relation to the other clusters in producing a stronger signal between most grain production and precipitation.

[Figure 9 is somewhere around here]

[Figure 10 is somewhere around here]

### 3.2.2 Grain production and climate variability 1865-1911

The results from the later study period, depicted by Fig. 11, are for the most part consistent with those of the earlier period using the HDSA data, although only for the spring grains (excluding spring-rye and spring-wheat). Notably, the coefficients are much higher for the latter period compared to the earlier period, roughly double in magnitude. May and to a larger extent June and July temperatures are negatively associated with the series for oats, barley and mixed-grain. Furthermore, maximum and minimum June temperatures also yield negative coefficients. Similar to the 1702-1865 period, the SPEI index is positively

correlated with the spring grains. The related MJJ$pr$ yield positive coefficients which are strong (between $r = 0.46$ for barley and $r = 0.57$ for oats). June precipitation is positive for the three spring grains, whereas no statistically significant results are obtained for July, which is surprising considering the importance of July precipitation in the period 1702-1865. Practically no statistically significant results were found for autumn-rye (except a negative association with precipitation in November), autumn-wheat (except a positive association with MJJ$pr$, $r = 0.46$), spring-rye and spring-wheat.

[Figure 11 is somewhere around here]

### 3.2.3 Grain production and climate variability during years with dry and wet summers

Repeating the analysis on restricted samples where only the years with the driest and wettest summers, respectively, are included, the direction of the relationships remain consistent. Fig. 12 shows the results of the correlation analysis in the early

period, considering only dry years (as defined by the SPEI). The magnitude of the negative association between summer temperatures and all grain production except for oats increases, yielding correlation coefficients between -0.3 and -0.52.

[Figure 12 is somewhere around here]

There is no statistically significant effect from monthly summer precipitation, except for the MJJ$pr$ which shows a positive association with most grain production in all clusters, except oats in Cluster 2 and Cluster 3. Considering only wet years in the

early period (Fig. 13), summer temperatures are still negatively associated with grain production in all clusters, especially Cluster 1. Additionally, there is a positive association between rye production and September temperatures in Cluster 2 and 3 in wet years. Notably, the highest correlations coefficients are obtained by the correlation between grain production and precipitation during June, and especially July, during wet years in the early 1702-1865 period ($r$ between 0.31 and 0.78).

[Figure 13 is somewhere around here]

**4 Discussion**

The main result obtained in this study was the negative association between all grain production and summer temperatures and a corresponding positive association with summer precipitation, especially during the high summer months of June and July, in the early study period (1702-1865), as well as for spring-crops in the late study period (1865-1911). Another important finding was the lack of signal for autumn grains in the late study period as well as the weak relationship between autumn and

spring temperatures and grain production during the whole study period. Finally, during homogenization of the monthly temperature series from Lund a cold bias was identified in the late 18[th] century and as well as multiple statistically identified breaks, almost all of which could be associated in time with changes in observers and/or instrument locations. In the following sections, all these results are discussed in more detail.

## 4.1 The relationship between temperature, precipitation and grain production across the seasons

For roughly two centuries, between early 18[th] and early 20[th] centuries, the results of this study show that Scanian grain production had a reversed relationship to temperature compared to other parts of Scandinavia, as well as other parts of Europe (Esper *et al.*, 2017; Pribyl, 2017; Brunt, 2015; Holopainen *et al.*, 2012; Waldinger, 2012; Holopainen & Helama, 2009). This merits some further discussion.

Precipitation and drought during the growing season were pointed to by Utterström (1957) and later by Edvinsson *et al.* (2009) as the primary agro-meteorological constraints for pre-industrial agriculture in southern Sweden. Much of the arable land in Scania, as elsewhere in southern Sweden, was situated on well-drained and elevated soils whereas meadows were often on more low-lying and wet soils (Dahl, 1942; Gadd, 2001). These circumstantial factors combined with the consistent findings of negative associations with June and July temperatures, and positive associations with precipitation in June and July indicates that summer drought was the greatest agro-meteorological risk to grain production. Relative to the benefits of intensive grain production in the region, this was by all accounts a risk worth taking. Indeed, most of the land improvements that occupied farmers in the 19[th] century after enclosures were about transforming wetter lands to well-drained arable lands, primarily through ditching, rather than efforts to preserve soil moisture (Bohman, 2010).

Nonetheless, it is worth considering what farmers could do to mitigate the risk of drought. In regards to the grain cultivation on well-drained soils, probably very little. It was after all the very same characteristics in the soil that increased the risk of drought that also made a large production of grains possible. Grain production in the cluster with the best soils in this study, Cluster 3, showed a relationship with climate variability that was of similar or greater magnitude than the other clusters. The extensive land reclamation efforts that took place during the 18[th] and 19[th] century could have helped mitigate against drought as an unintentional and temporary side-effect by making new lands of variable qualities, not least in terms of drainage, available (Håkansson, 1997). A diversified composition of grain production also helped to some extent to make grain farming more resilient in Scania. The slight but important variation *between* the grains in terms of their relationship to summer temperatures and precipitation, where oats and rye were more sensitive to variation in May and in particular June, and barley was more sensitive to variation in July, accordingly spread out risks. Diversity *within* each grain variety would also have been helpful in mitigating the risk to drought or other climate anomalies (Hagenblad *et al.*, 2012; Hagenblad *et al.*, 2016; Leino, 2017; Lundström *et al.*, 2018).

As noted in Section 1.4, Edvinsson *et al.*(2009) suggested that before the agrarian transformations in the 18th and 19th centuries, yields were in general so low as to lead to a chronic shortage of seeds, which they suggested overrode the effects from climate variability and hence the weak relationship between temperatures and grain yields. Theoretically, low yields leading to low seed quantities could obfuscate the effect of temperature, necessitating some kind of control for the previous year's weather. For example, Bekar (2019) found that English manorial harvests in the 13th and 14th centuries were persistent, i.e. subpar

harvests, partly induced by 'weather shocks', persistent into the subsequent year for both wheat and other grain crops like barley and oats. Notwithstanding these results, the relevance of 14th century England for 18th century Scandinavia is arguably limited. In instances were one might assume persistent harvests would be more apparent (although it is arguably an understudied phenomenon), like northern Finland, one still finds strong current-year temperature effects on grain yields and production during the early-modern period (Huhtamaa & Helama, 2017b; Huhtamaa, 2015; Solantie, 1988). Having limited

amounts of seed did not obfuscate or exclude the effects from weather. Rather the evidence seems to suggest it made farmers more vulnerable and the effects more apparent. There are few reasons to suspect that chronic seed shortages was a major issue in 18th century Scania, given that it was mostly an exporter of grains and experienced more or less ongoing increases in production during the period (Olsson & Svensson, 2010).

In relation to this argument, I would highlight another important result of this study, namely the absence of a climate signal in

the spring and autumn months, as well as the last summer month of August to some extent. In the neighboring lands of Denmark, Ljungqvist & Huhtamaa (2021) suggests that it is possible that frosts were not a major problem, even in the wetter and colder periods of the LIA. The results from Section 3.1.2 showed that the average date for the first occurrence of autumn-frost in Lund was the 8th of November and not earlier than October in the northernmost areas of Scania, well after the harvest month of August as well as the sowing of autumn-rye. In many years there were no spring-frosts later than March, whereas in

those years when they occurred after March the average date was the 15th of April in Lund. In the highlands in the north, spring-frosts on average occurred later. Nonetheless, spring-frosts, when they occurred, generally did so just before or at the start of the growing season. Thus, the results in this study points to spring and autumn frosts not being a systematic threats to grain production, except in localized conditions. An implication of this is that the combination of the climate in Scania with the farming systems Scanian farmers adhered to offered good margins for the spring and autumn agricultural work seasons,

for example by allowing for delays in sowing and harvesting.

Based on the findings in this study, I would revise the notion forwarded by Utterström (1957) and Edvinsson *et al.* (2009). It was not only precipitation but rather that the combination of precipitation and precipitation during the summer were the main constraints for the production of spring grains during the whole study period and for all grains in the early study period. Furthermore, I would not only frame the relationship in the form of constraints and risks. Grain farming in Scania was adapted

to, and benefitted by, cool and humid summer conditions. Even in years with wet summers there was a positive association between grain production and summer precipitation and a negative association between the former and summer temperatures. This would likely also be the case in other parts of southern Sweden were similar grain varieties were cultivated in the same

type of farming systems on well-drained soils. In the later study period, both maximum and minimum June and July temperatures were negatively correlated with the production of spring-grains, suggesting an optimal temperature range for these crops during these months and that the occurrence of extreme cold and heat had some detrimental effect.

## 4.2 The role of grain varieties

An account of the relationship between grain production and climate variability has to account for the type and varieties of grains being cultivated as well as the farming system more broadly. In Scania in the late 19[th] century, new autumn grain varieties, similar to those on the European continent, gradually replaced the old varieties that were more similar to those on other parts of Fennoscandia. I argue that this, rather than changes in the climate. was the most likely cause behind the diminished signal in the relationship between climate variability and autumn grains in the latter study period, given that the relationship with climate variability remained intact for most of the spring-crops. Edvinsson *et al.* (2009) also argued that the shift towards new grain varieties changed the underlying relationship between grain production and climate variability, from a negative to a positive association with temperatures, especially in the spring and early summer. The farming systems of Scania likewise underwent changes where arable lands were expanded and intensified with new crop rotations, land improvements and increased drainage, external sources of fertilizer and burgeoning mechanization. All these changes created conditions that were more favorable for the new grain varieties.

Nonetheless, it is possible that climate changes over a longer time-scale was an active driver in the relationship between climate variability and grain production, considering that the historical grain varieties were changeable and adapted to changing circumstances, not least climate variability at different time scales (Leino, 2017). It can be argued that farming in Scania during the 17[th] up until the late 19[th] century had over time adapted to a more cool and humid summer climate, having experienced multiple cold years and extended periods with reduced average temperatures during the LIA in the 16[th] and 17[th] centuries, and possibly earlier as well. Current evidence suggest that the grain varieties cultivated during these centuries were similar or of the same group of varieties grown in more northerly and cold latitudes. For example in regards to rye, Larsson *et al.*(2019) found through genetic analysis of preserved Fennoscandian rye seeds that they all belonged to the same meta-population of rye landraces that had been stable for at least the last 350 years. Similarly, Aslan *et al.* (2015) found that barley landraces from Fennoscandia form a homogenous group of barley landraces, distinct from other parts of Europe. This particular group of northern European barley varieties carry the nonresponsive ppd-H1 allele that prolongs flowering when exposed to periods with increasing daylight hours. Presumably, this would be beneficial during cooler and wetter periods by taking full advantage of the extended growing season. Studies of modern Finnish barley cultivars have shown that yields for most varieties are negatively correlated with excess rain or drought around the sowing season and positive in the subsequent stages of crop development, whereas they are negatively correlated with temperature at most stages of crop development, especially before heading (Hakala *et al.*, 2012). The homogeneity of barley landraces over time in southern Sweden were confirmed by Lundström *et al.*(2018) who traced it back to at least the late 17[th] century. While Lundström *et al.*(2018) argued that such

homogeneity was maintained *despite* repeated crop failures in southern Sweden between the 1700s and the 1900s, I would argue, at least when considering Scania, that such homogeneity was probably maintained because of the *lack* of repeated crop failures.

The discussion of the results on the relationship between specific grains to climate variability should also be put in a broader perspective. In terms of crop composition and the type of field-system (a Swedish variant of the open-field systems called
*tegskifte*), the farming systems of Scania remained more or less the same until the 19th century, when enclosure and new crop rotation systems started to be introduced, starting in the plains districts. All the same, even after the introduction of new crop rotations, which normally meant increasing shares of fodder crops, in Scania grain production continued to retain its primacy, at least in the plains districts (Bohman, 2010). It is thus motivated to argue that the farming systems of Scania overall were resilient towards colder conditions, at least until the late 19th century, given the importance of grain production. However, the
relationship between livestock production and total agricultural production and climate variability would require a study of its own. Taking an even broader perspective, the larger agrarian economy of Scania was increasingly integrated northwards rather than westwards after Scania was annexed by Sweden in 1658. While it is likely that Scanian farmers cultivated northerly grain varieties before 1658, after this date the geographic, economic and political conditions were set for such grain varieties to consolidate their position.

**4.3 Implications of the late 18th century cold bias and ACMANT detected breaks**

Previous research that identified increasing soil erosion and sand drift in Scania during the 18th century partly blamed the coldness of the last three decades of the 18th century as indicated by the instrumental temperature measurements taken in Lund (Mattsson, 1987). After homogenization of the Lund temperature series 1753-1870, the largest corrections due were for upwards adjustments for summer temperatures in the same period, i.e. the late 18th century. In other words, the results suggest
the c. 1770-1800 period was not as cold as suggested by Mattson (1987) or the unhomogenized Lund temperature series. While these findings speak against a regional climate-driven ecological crisis, they do align with the results of Bohman (2017a, 2017b) who downplayed the spatial scale of the ecological crisis, emphasizing its local and conditional character, as well as the counter-acting efforts by local communities and authorities.

The breaks detected through the ACMANT procedure could almost all be associated with changes in observers and/or
665 instrument location. While there is uncertainty on the number of times the instruments were replaced, those known could not be associated in time with the detected breaks. This suggests that the human factor, i.e. the degree of consistency in training, skills and interest in observers, was the primary determinant in measurement quality and homogeneity. Faulty instruments or station location biases could in the end only be perceived and understood and subsequently corrected or adjusted for by the human observer. The first decades, 1753-1774, can be considered as a period of more competent (meteorological) observers,
followed by the period after ca 1850 when training and methodologies in meteorological observations had improved. Nonetheless, for most of the homogenization period there were issues of inhomogeneity requiring corrections.

## 4.4 Hydroclimate and historical grain production

Three different hydroclimate reconstruction were employed for this study. In the early period (1702-1865) very few and mostly inconsistent results were obtained using the scPDSI from the OWDA, and no statistically significant results were found for the late period (1865-1911). The SPEI from Seftigen *et al.* (2017) was found to be positively associated with most grain production except barley, consistent over different samples and periods as well as with results from instrumental precipitation. The results of the MJJpr and SPEI could be interpreted as them being more important for estimating hydroclimatic conditions relevant for grain production in the early summer (May and June, in particular). This is supported by the lack of statistically significant effects found between May and June climate variables and barley production and that the most important month for barley seems to have been July, at least in the early period. This also offers an explanation as to why sorting dry and wet periods with the SPEI indicator led to much larger associations between precipitation and temperature in June and July with grain production. If conditions were wet or dry in the early summer, the effects from subsequent precipitation and temperatures later in June and especially July would theoretically have been amplified. A similar argument was made by Brunt (2004) who showed that it was more beneficial to have precipitation spread out during the growing season. An important caveat to these interpretations is that there remains a large degree of uncertainty as to what specific hydroclimate effects are captured by or represented in these reconstructions, beyond MJJ or JJA averages. Nonetheless, there do seem to be a relationship between the conditions for tree growth in southern Sweden as represented in these reconstructions and grain production in Scania during the study period. Seftigen *et al.* (2015) asserted that even though most high-resolution climate proxies in northern latitude regions are temperature-based, there is also a need for precipitation-based proxies due to the importance of precipitation patterns for economic sectors such as agriculture. The results obtained here confirms both the importance of precipitation patterns for agriculture as well as the relevance of the proxy reconstructions in studying that relationship.

## 5 Conclusions

This article demonstrates the possibilities in estimating the relationship between climate variability and grain production in Scania during the pre-industrial period using available grain production data, climate reconstructions and the network of early instrumental records. Grain production in Scania did not show any systematic relationship or vulnerability to climate variability in the spring and autumn seasons, whereas a more clear signal could be detected between grain production and climate variability during the summer season, especially in the months of June and July. Until the introduction of new varieties of autumn-crops in the late 19[th] century, grain production was benefitted by cool and wet conditions throughout the summer, although there was a slight but important differentiation between rye and oats, which were more sensitive to conditions in May and June, and barley, which was mostly sensitive to conditions in July. The most apparent agrometeorological risk was summer drought. However, severe droughts like the one in the summer of 2018 were rare in Scania and the diversification within and between historical grain varieties cultivated meant that, by and large, this risk was manageable, especially when compared to the benefits of intensive grain production in the region. Scania largely conforms to the previous, albeit sparse, picture in the

Swedish historiography of the relationship between historical grain production and climate in southern Sweden. At the same time it stands out compared to studies of other parts of Scandinavia and continental Europe where positive associations between grain production and summer temperatures have been identified.

The results obtained here should be further developed on by integrating them into a broader model of the impacts of climate variability on agriculture where other factors, e.g. market prices and access, institutional and other geographical factors like soil conditions, are formally accounted for. This need is not least implied by the fact that even in the confined geographical area of Scania there was differentiation among sets of villages as regards to the relationship between their grain production and climate.

**Appendix A**

**Homogenization of the Lund temperature series**

Daily meteorological observations began in Lund in 1740, spearheaded by the Professor of Mathematics D. Menlös. Systematic instrumental meteorological observations began in 1747 for precipitation and late 1752 for temperature and air pressure under the responsibility of a formally appointed observer. Naturally, these observers were changed over time. Instruments were also replaced or upgraded on a few occasions. More problematic from a perspective of consistency and reliability, the location of the instruments also changed multiple times. There is also a gap in (quality) temperature measurements between the years 1821-1833, and some other minor gaps over the period 1753-1870 (see Fig. A1). Issues relating to the non-homogeneity of these meteorological series, not least the temperature series, were partly identified and discussed already in the 19[th] century by Tidblom (1875). Schalén *et al.* (1968) and Bärring *et al.* (1999) also discussed inhomogeneities in the series relating to the station history.

[Figure A1 somewhere around here]

Tidblom (1875) published the Lund temperature series in the form of pentad averages. He removed the daytime measurements, using only the morning and evening measurements arguing that these were less affected by the location of the thermometer. He also did some minor manual corrections for September through December in 1834 and individual days in June in 1842 and 1843. Overall, the adjustments made by Tidblom (1875) should be considered minor and it seems probably that there remains

inhomogeneities in the series. Nonetheless, the temperature series have subsequently been employed in at least a few historical studies. For example Palm (1997), used the series to estimate the impact of temperature variations on grain yields on a farm in Halland, southwestern Sweden during the years 1758-1865). To fill in the gaps Palm bridged the Lund-series with data from Copenhagen, using average differences. Mattson (1986) observed the trends in the Lund-series and argued that a reduction in temperatures in the last three decades of the 18th century, combined with changes in wind patterns, contributed to widespread

soil erosion in Scania during the 18th and early 19th centuries. Using inhomogeneous data sets carries many important drawbacks, not least the risks of spurious and unreliable results (Aguilar *et al.*, 2003). Hence, it is necessary for the purposes of this study to homogenize the temperature series.

Given that the exact location and relocations of the measurements for the period up until 1780 is unknown, it is difficult to

745 identify periods of measurement error and estimate correction coefficients manually. The most tested approaches for homogenization of temperature series involves using interpolation techniques relying on homogenized data from nearby stations or networks of stations (Venema *et al*., 2012). It is generally advisable to use more than one station for interpolation since it reduces the probability of a single station bias as well as the general reliability of the interpolation (Conrad & Pollak, 1950). Again, applying interpolation manually is problematic since it risks introducing new biases for the less detailed parts

of the Lund measurement station history. There are also computational difficulties in manually interpolating from a network of stations. Therefore, I employ the Adapted Caussinus-Mestre Algorithm for homogenizing Networks of Temperature Series (ACMANT) software. ACMANT relies on a homogenized network of stations and a computationally efficient algorithm for homogenizing climate data and data infilling.

I use a network of homogenized monthly temperature series located in the northwestern part of Europe for the homogenization

process. Namely, Berlin-Dahlem (from 1719 with a gap between 1722-1727), Central England (from 1659), Copenhagen (from 1768 with gaps between 1777-1781 and 1789-1797), De Bilt (from 1706), Stockholm (from 1756) and Uppsala (from 1722) (DWD, 2018; Bergström & Moberg, 2002; Cappellen, 2017; Moberg *et al*, 2002; van Engelen *et al.*, 2001; Labrijn, 1945). Given that spatial correlations in temperatures on a daily or weekly basis are lower across the network region compared with monthly or seasonal averages, and that daily temperature series are not available at all network stations, I employ monthly

averages for homogenization. The ACMANT homogenization procedure requires spatial correlation coefficients of at least 0.4 with network stations and a minimum of 4 network time series. Spatial correlation coefficients are calculated from the increments in the time series after monthly climatic means have been removed (Domonokos & Coll, 2017). Table A1 shows the descriptive statistics of the network stations, including monthly correlations and spatial correlations.

[Table A1 somewhere around here]

## Detected breaks and station history

The ACMANT homogenization procedure detects eight breaks in the Lund temperature series. These breaks can be interpreted in light of what is known about the station history. Fig. A2 shows known or suspected relocations and changes in observers between 1753 and 1870 as well as the detected breaks (Nenzelius, 1775; Tidblom 1875; Schalén, 1968; Bärring *et al.,* 1999).

[Figure A2 somewhere around here]

For the first 22 years, no breaks are detected, even though there is one change in observer (1763) and several replacements of instruments. The instrument location do appear to have been constant. Furthermore, in 1770 there was a Royal Ordinance that the results from all monthly meteorological observations were to be sent in to the Swedish Royal Academy of Sciences. The first two decades of measurements were published in the Swedish Royal Academy of Sciences (Nenzelius, 1775). It is possible that the attention and interest given to meteorological observations as a preeminent scientific venture at this particular moment in time led the first two observers, N. Schenmark and O. Nenzelius, to make serious efforts to make sure the series was consistent or as correct as possible. Tidblom (1875) argued that were was no reason to suspect that the instruments or observers were lacking in quality or skill, at least during Schenmark's time (1753-1763). In the 1770s, there are some intermittent notes from this period of corrections to faulty instruments, something that is much sparser in the subsequent period (Tidblom, 1875).

The first detected break occurs in 1775, the same year which A. Lidtgren and his assistant P. Tegman overtook observation responsibilities. It is possible that the thermometer also changed location at this time. Another break occurs in 1780, approximately coinciding with the change of location for the instruments in late 1779 to the upper story of Kungshuset. In 1798, 1804 and 1813 there after further breaks detected in the series. Nothing formally appears in the station history that could explain the 1798 break. However, the responsible observers, A. Lidtgren and P. Tegman, were both increasingly occupied with other duties, suggesting that actual meteorological observations were undertaken by some other unknown assistant. P. Tegman became Professor of Mathematics in 1787, awarded membership in the Royal Swedish Academy of Sciences in 1795 and in the board of Mathematics in 1798. Furthermore, he was appointed dean of Lund University in 1795 and became responsible for a church deanery in 1797. A. Lidtgren was also awarded membership in the board of Mathematics in the Royal Swedish Academy of Sciences in 1798 (Ståhl, 1834; Dahlgren, 1915). A. Lidtgren was recognized for his work in astronomy and astronomical observations, whereas much less is known about his work with meteorological observations (Dahlgren, 1948). Unlike their predecessors, neither P. Tegman nor A. Lidtgren made any publications regarding meteorological observations.

Both the 1804 and the 1813 breaks occur during times of more apparent and known changes in the station history. In 1804 the instruments changed location several times and in 1813 there was a change in observer to A. F. Knieberg who supposedly also relocated the instruments to his living residence (Tidblom, 1875). Finally, there are detected breaks in 1845, 1846 and 1859. Between the 1820s and mid 1830s the instruments changed location on a number of occasions (leading to the gap in the temperature series between 1821 and 1833 due to the low quality of the temperature data in that period). The location of the instruments is then not mentioned in the records until 1846 when they again appear to have changed location. In 1858, there was a change in observers, but there are no changes in location close in time to the 1859 break.

Thus, almost all detected breaks occur in the same year or the subsequent year to a change in observer or station location. Furthermore, almost all known changes in station location and observers are detected as breaks, at least in the 18[th] century. The largest exception is the break in 1845 that occurs more than a year after the latest known change in the station history, which is 1843 when P. G. Ahlander overtook observation duties. However, in 1846 there is both a station relocation and a detected break. The latest detected break occur in 1859, shortly after a change in observers to Ljunggren *et al.* in 1858.

Overall, the change of instruments appear to have been a less important factor in causing inhomogeneities in the series compared with changes in observer and instrument location. Not one of the known changes in thermometers occurred at a point in time approximate to a detected break. Presumably, if an instrument was faulty the more skilled observers, as is noted on several occasions in the station records, could correct for this. Similarly, an unskilled or careless observer would have been less likely to identify faulty instruments, accurately read and note down observations and to appreciate the consequences of moving the instruments to a particular location (see Pfister *et al.* 2019 who discusses issues related to the maintenance of reliable and consistent observers).

### The late 18[th] century cold-bias

As shown in Fig. 8 and Fig. 9, the largest corrections in the homogenization procedure occurred during the summer months and to a lesser extent spring months of the last decades of the 18[th] century, signified by the two detected breaks in 1775 and 1780 (Fig. A2). Other studies attempting to homogenize early instrumental temperature series have also found summers to be a larger source of measurement bias, compared with the other seasons. For example, when performing manual testing during homogenization of the Stockholm instrumental temperature series (beginning in 1756) Moberg (2002) noted, in line with Modén (1963), that the largest discrepancies due to station location were to be observed during the summer months. Böhm *et al.* (2010) also found a similar seasonal pattern in their study of early instrumental temperature series in central Europe. They argued that thermometers, which during the late 18[th] and early 18[th] century were mostly placed in a north-facing direction and without proper sheltering, were subject to a systematic summer season warm-bias. They also conceded that this bias differed for different stations, depending on latitude, altitude and other station-specific conditions.

In the Lund early instrumental period, particularly in the late 18[th] century, it appears conditions were reversed from those in central Europe, namely there is a *cold*-bias during the summer season as well as the during the growing season (AMJJAS) as a whole. Across the entire homogenization period, there is a slight correction upwards for spring and summer temperatures, which can be explained by the use of morning and evening observations, excluding daytime observations. To explain the much larger corrections made from 1775 until 1804 one has to consider the specifics of the station history during that period. During the years 1775-1806, the average time of the morning observations occurred up to two hours earlier during the months of MJJA, compared to the previous period 1753-1774. For the other months of the year the differences were much smaller, see Tab. A2. Furthermore, in late 1779 the thermometer was moved to the upper room of the Kungshuset, located at an altitude of 61m. Bärring *et al.* (1999) argues that these facilities probably were unheated until the 1830s. Given these conditions, it is feasible that the thermometer location between 1779 and 1804 was colder than the preceding and succeeding locations.

[Table A2 somewhere around here]

Thus, a consideration of the specifics of the station history, notably the change in observers, observation practices and station relocation, in combination with the homogenization results, suggests that the Lund temperature series exhibits a cold bias in the last decades of the 18[th] century. This result has bearing on historical climate reconstructions generally, but also for the agrarian and climate history of Scania and southwestern Sweden, specifically, as discussed in Section 4.3 (Mattson, 1986; Palm, 1997; Bohman, 2017a; Bohman, 2017b).

**Appendix B**

[Table B1 here]
[Table B2 here]
[Table B3 here]
[Table B4 here]

**Data availability.** All original data used for this article can be found available online through URLs in the reference list.

**Competing interest.** The author declare that he has no conflict of interest.

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

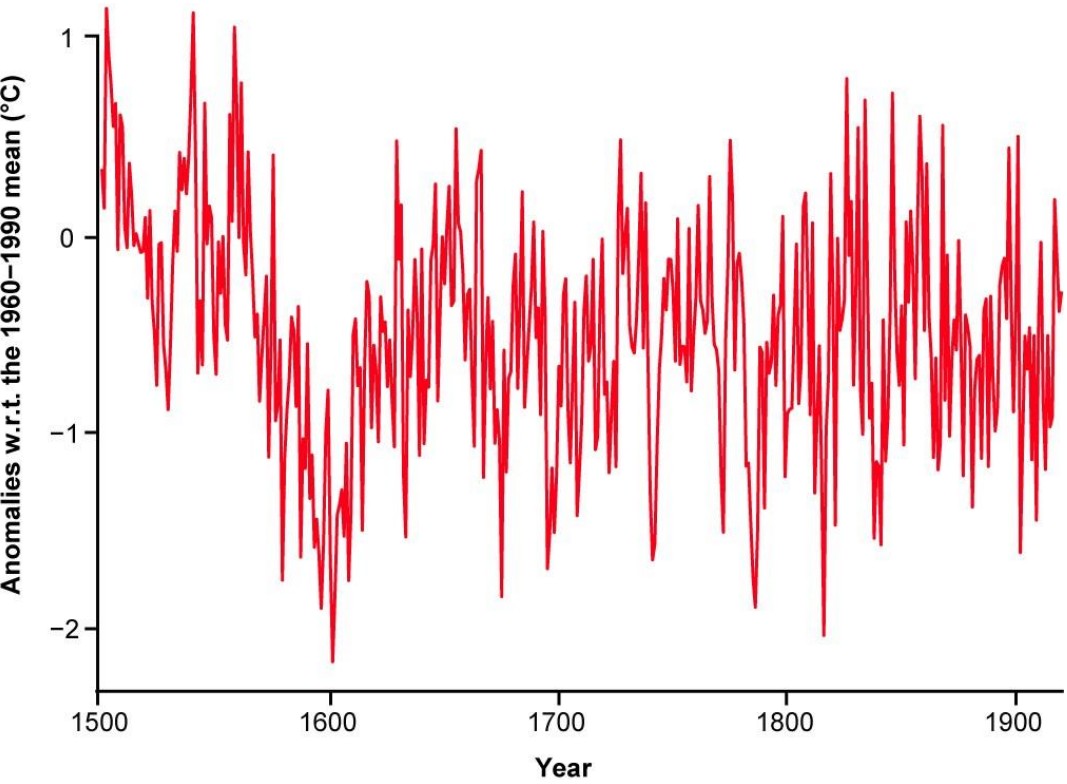

**Figure 1:** 1500-1920 summer (JJA) temperature reconstruction, from Ljungqvist *et al.* (2019). Based on grid cell at 12.5° E and 57.5° N, corresponding roughly to Mark Municipality in southern Västra Götaland. Source: Ljungqvist *et al.* (2019).


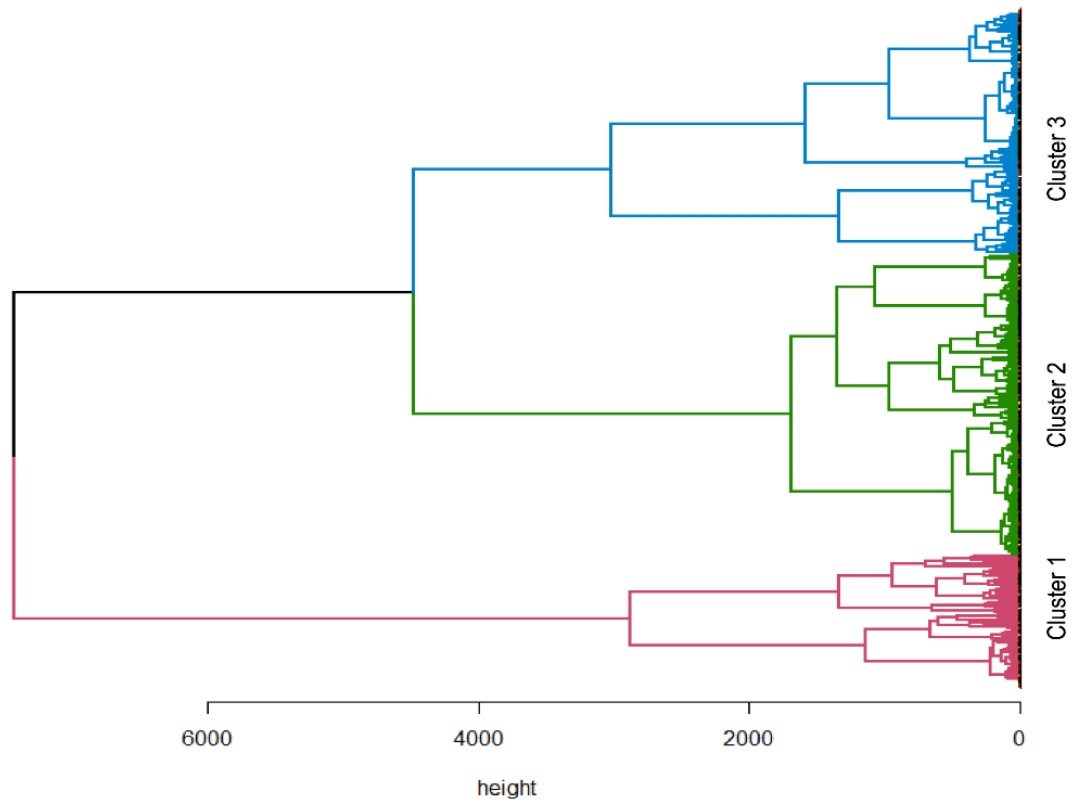

**Figure 2**: Cluster dendrogram illustrating the sorting process leading to three most-similar clusters. Source: HDSA.

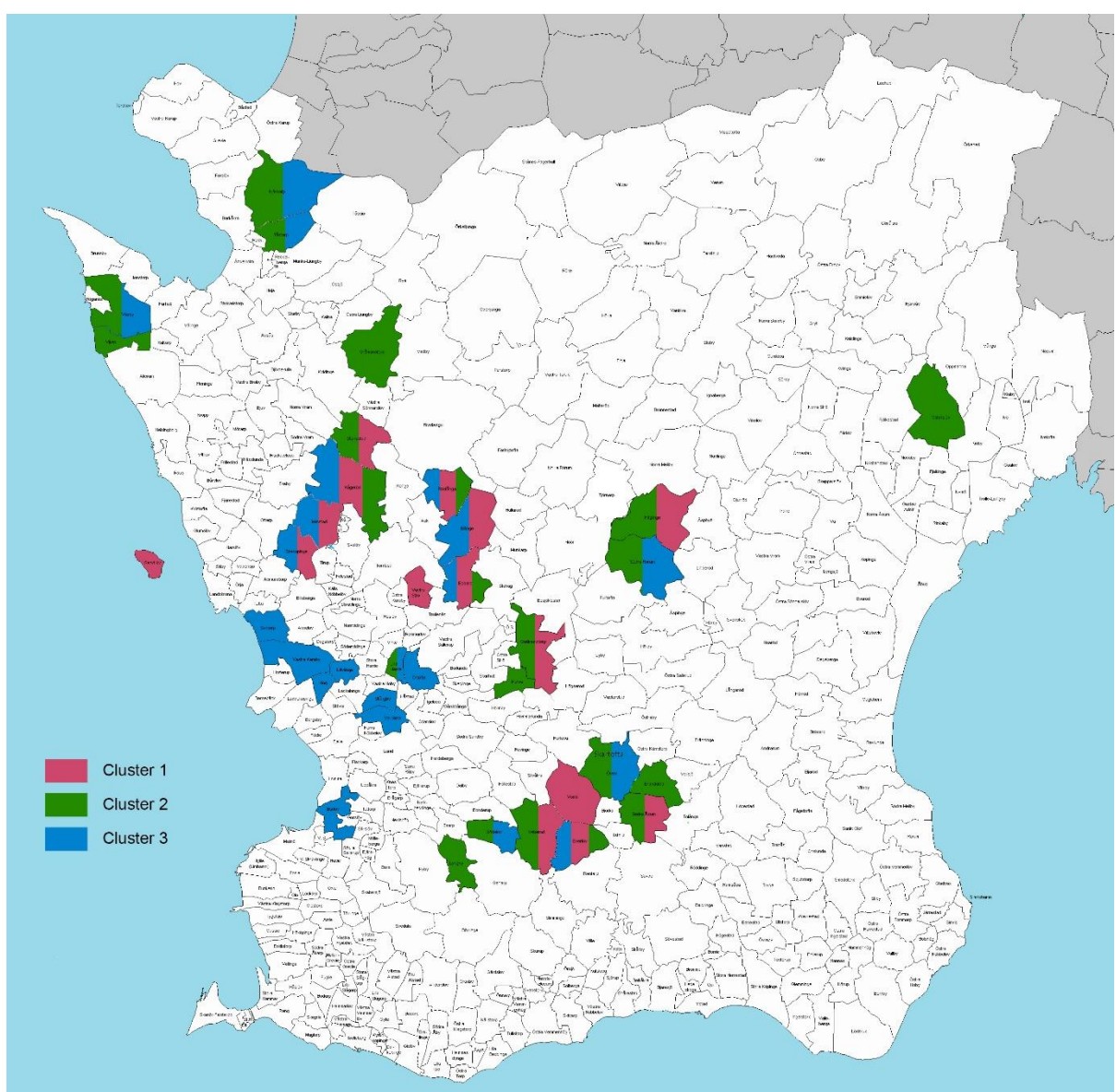

**Figure 3**: Geographical and administrative (parish) representation of each cluster. Source: author's own edit of the Parish map
of Scania from Wikimedia Commons (2010).

**Table 1**: Descriptive statistics for each cluster, including proportions of soils of different qualities.

| Clusters 1-3 | 1 | 2 | 3 |
|---|---|---|---|
| Years covered | 1711-1864 | 1702-1861 | 1702-1860 |
| Villages | 173 | 137 | 71 |
| Village-level observations | 8511 | 4311 | 5551 |
| Farms | 481 | 514 | 389 |
| Farm-level observations | 32420 | 22054 | 31432 |
| Soil grades by proportion in each cluster | | | |
| 1 | 0 | 0.001 | 0 |
| 2 | 0 | 0.001 | 0 |
| 3 | 0.16 | 0.04 | 0.01 |
| 4 | 0.23 | 0.11 | 0.08 |
| 5 | 0.15 | 0.12 | 0.25 |
| 6 | 0.16 | 0.16 | 0.19 |
| 7 | 0.15 | 0.38 | 0.10 |
| 8 | 0.07 | 0.15 | 0.21 |
| 9 | 0.02 | 0.04 | 0.01 |
| 10 | 0.06 | 0 | 0.15 |

Notes: note that there are no grade 1-2 soils in the sample, whereas the amount of the highest-grade (8-10) soils is quite large.

Source: HDSA.

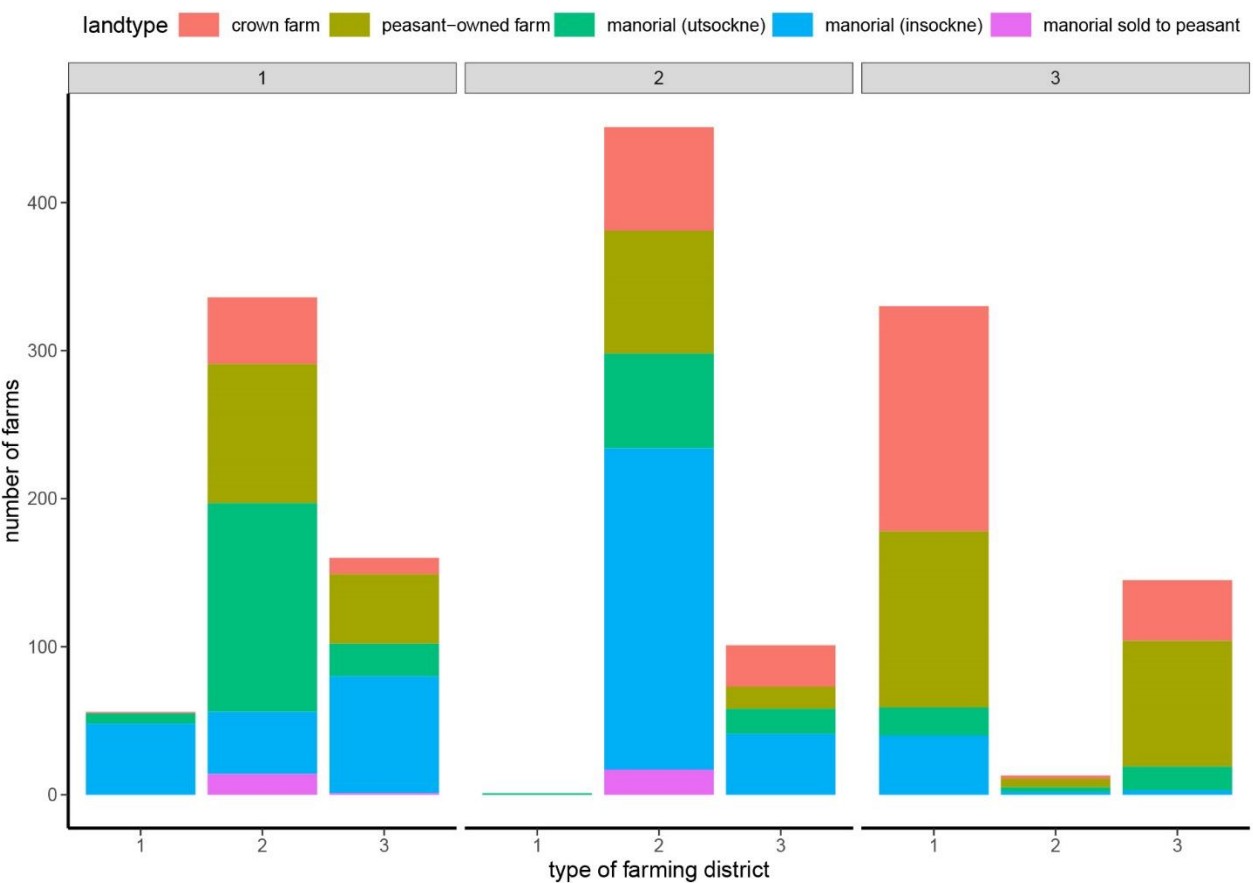

**Figure 4:** Institutional status of farms, including type of farming district. Each bar-plot represents a cluster with each cluster denoted in the grey-marked area. 1 denotes plain districts, 2 mixed districts and 3 signifies forest districts. Source: HDSA.


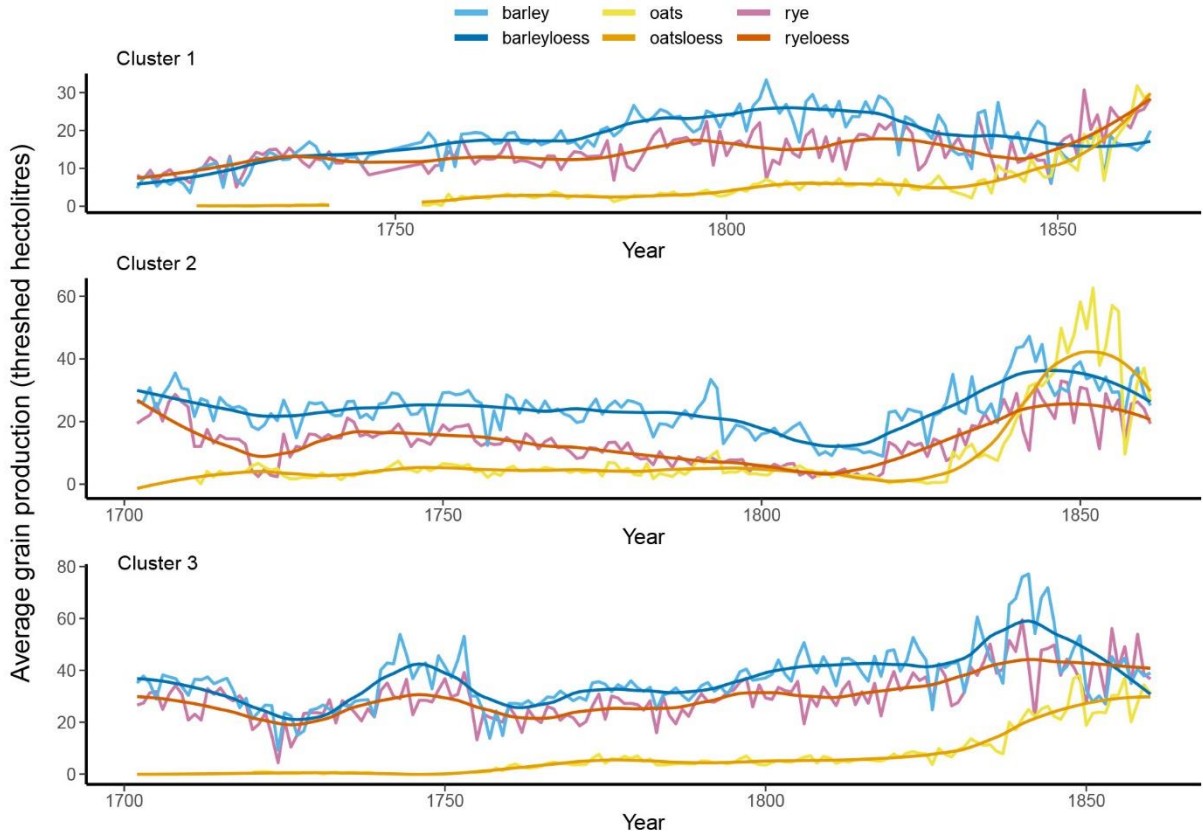

**Figure 5**: Average grain production (threshed hectoliters) in each cluster over time, including estimated loess. For Cluster 1 the years 1743-1746 are covered by only one farm, heavily skewing the average for those years. Therefore, I have substituted the values for rye and barley for the years 1743-1746 with values obtained from a linear estimation of the relationship between the production of that farm and the average production in the Cluster in the years 1727-1742. Source: HDSA.

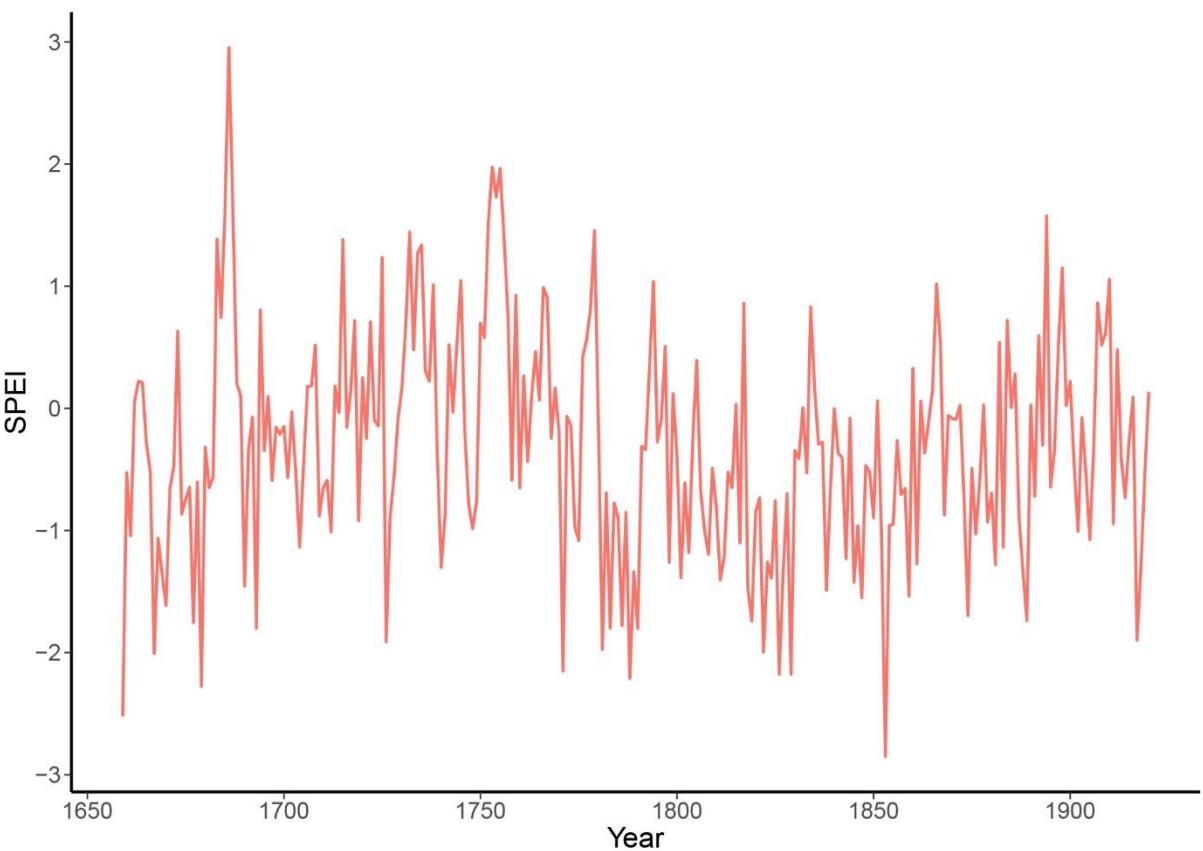

**Figure 6**: Reconstruction of southern Scandinavian SPEI, 1659-1920. Source: Seftigen *et al*. (2017).



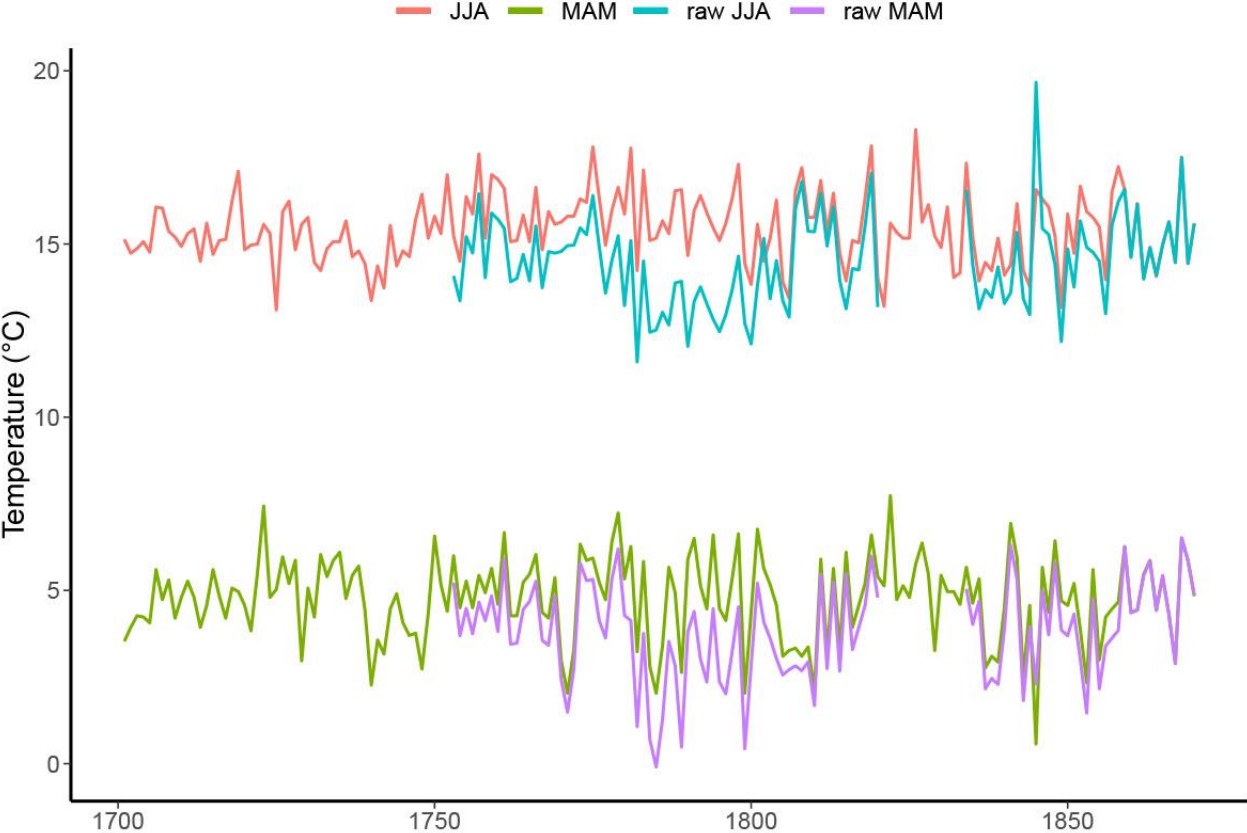

**Figure 7**: Raw and homogenized seasonal JJA and MAM mean temperatures at Lund, 1701-1870. Sources: Copenhagen (Cappelen *et al.*, 2019), Berlin-Dahlem (DWD, 2018), De Bilt (Durre *et al.*, 2008; Lawrimore *et al.*, 2011), Lund (Tidblom, 1876), Uppsala (Bergström & Moberg, 2002) and Stockholm (Moberg *et al.*, 2002, Moberg, 2021). Notes: See Appendix A for further discussion of the homogenization process.

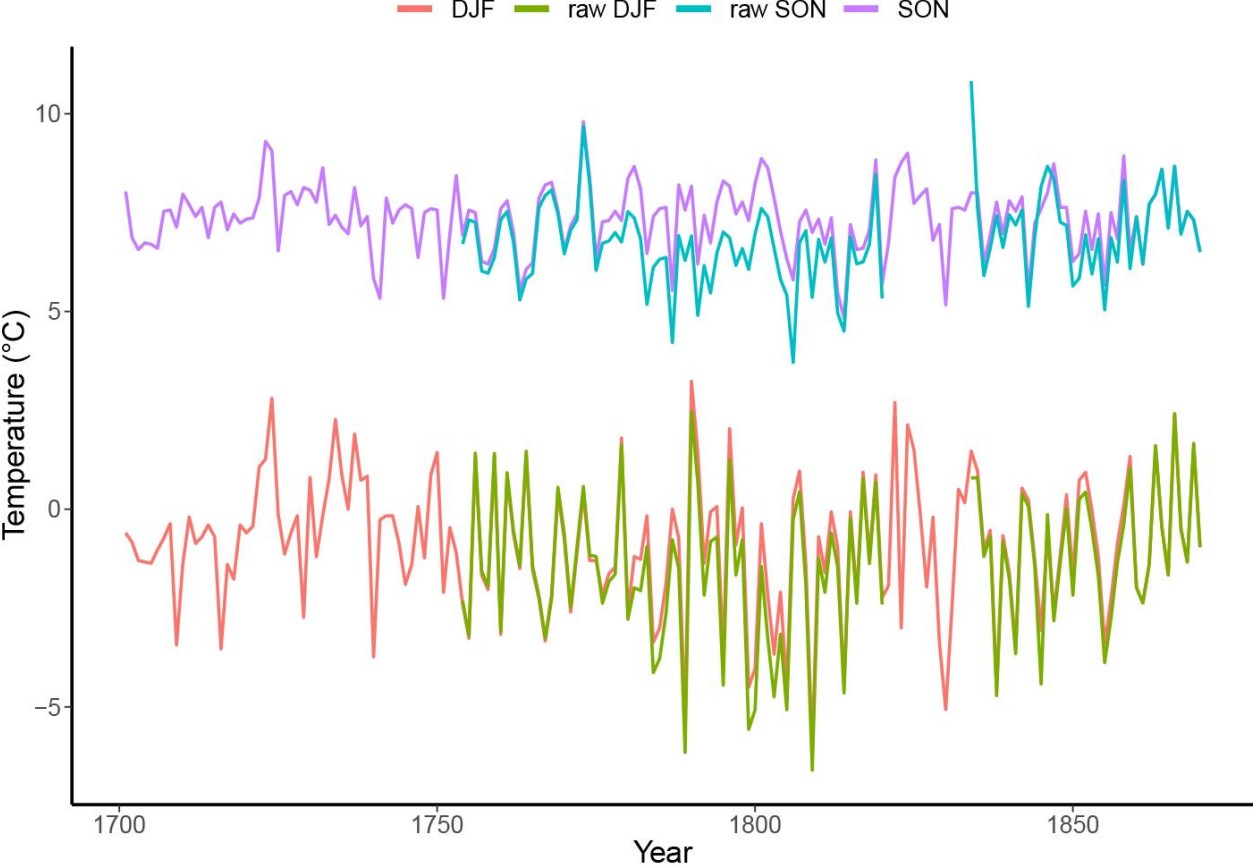

**Figure 8**: Raw and homogenized seasonal DJF and SON mean temperatures at Lund, 1701-1870. Sources: Copenhagen (Cappelen *et al.*, 2019), Berlin-Dahlem (DWD, 2018), De Bilt (Durre *et al.*, 2008; Lawrimore *et al.*, 2011), Lund (Tidblom, 1876), Uppsala (Bergström & Moberg, 2002) and Stockholm (Moberg *et al.*, 2002, Moberg, 2021). Notes: See Appendix A for further discussion of the homogenization process.

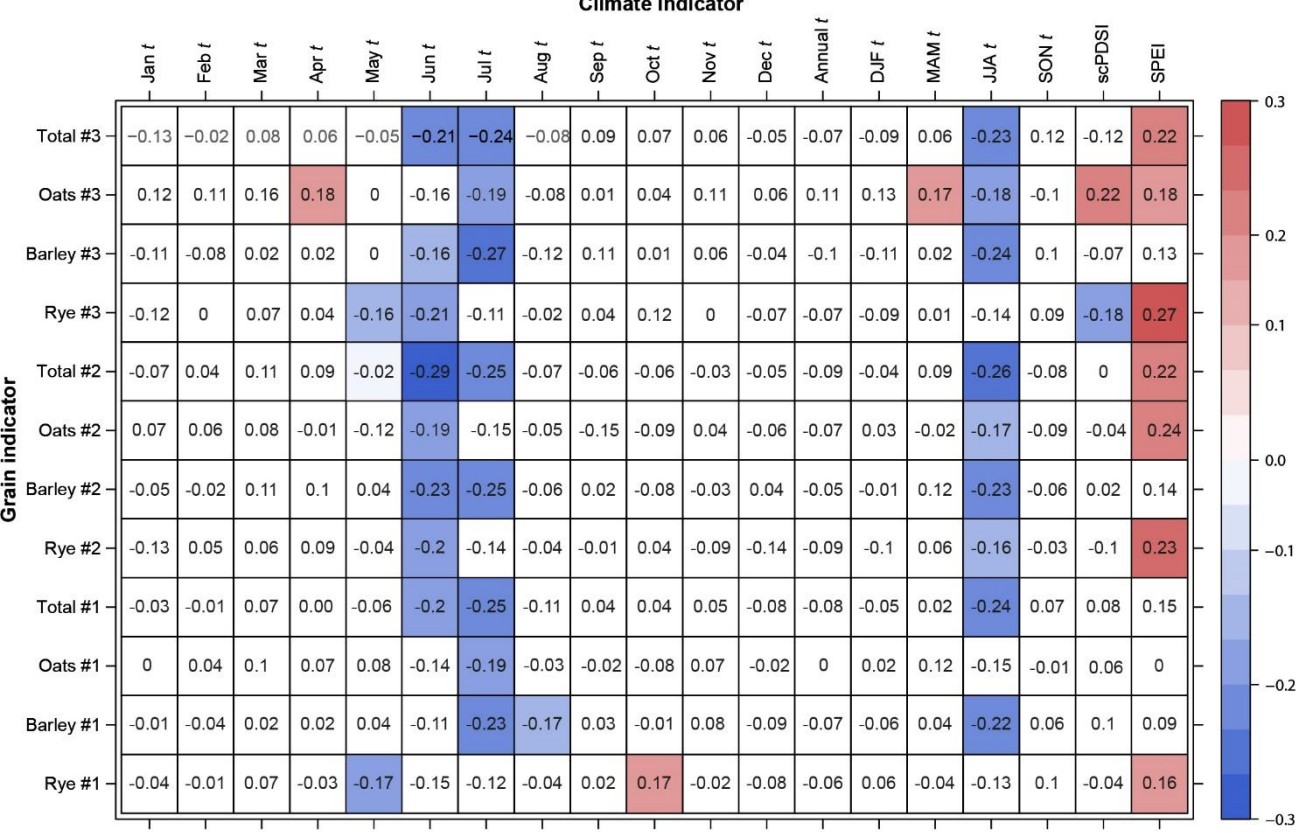

**Figure 9:** Correlations of grain series vs temperature and hydroclimate indicators, c. 1702-1865. Notes: Only statistically significant ($p \leq 0.05$) correlations are colored. Clusters are signified by the # number. Sources: HDSA, Cook *et al.* (2015), Seftigen *et al.* (2017) and the homogenized monthly temperature series (APPENDIX A).

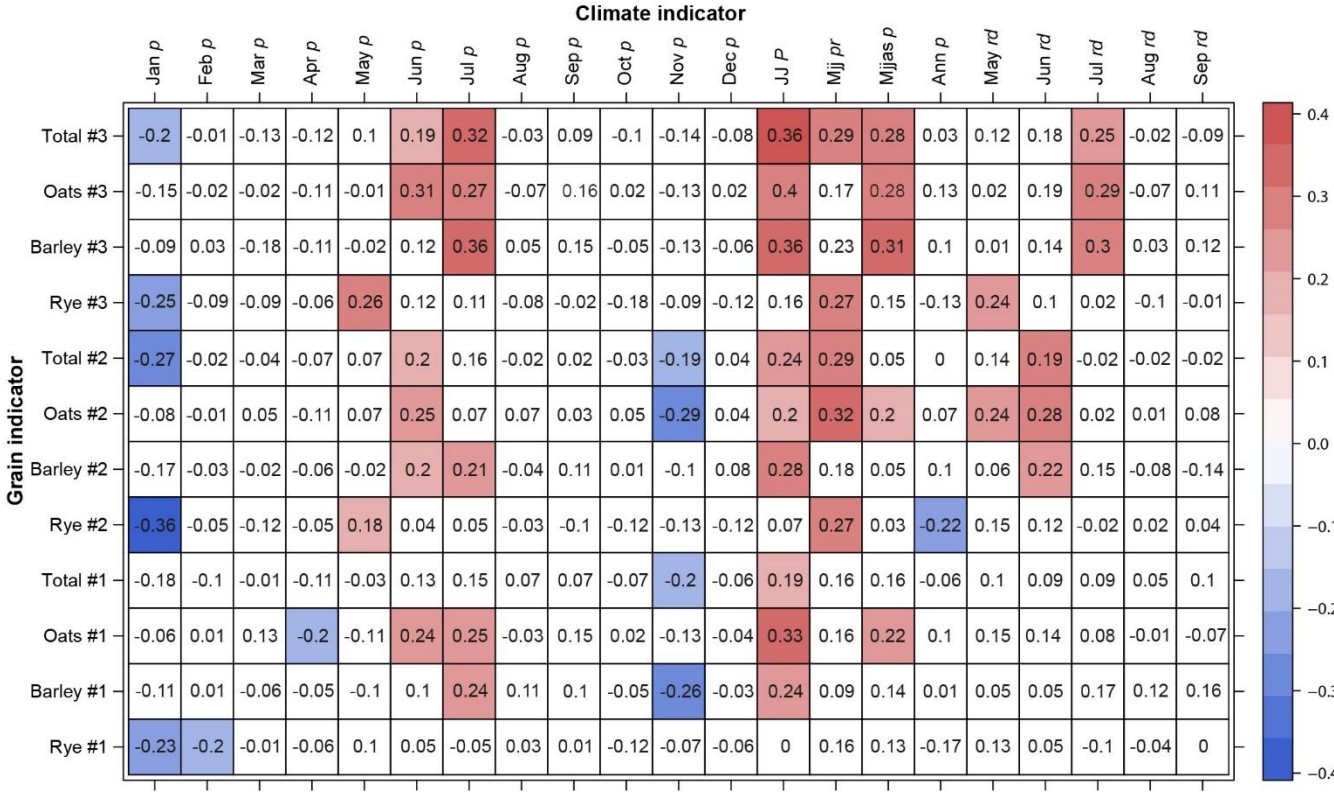

**Figure 10**: Correlations of grain series vs precipitation indicators, c. 1748-1865. Notes: Only statistically significant ($p \leq 0.05$) correlations are colored. Clusters are signified by the # number. Sources: HDSA, Sefitgen *et al.* (2020) and Tidblom (1876).

**Grain indicator**

| Climate indicator | Autumn wheat | Mixed grains | Oats | Barley | Autumn rye |
|---|---|---|---|---|---|
| $T_{min}$ Aug | -0.1 | 0.03 | 0.04 | -0.01 | -0.21 |
| $T_{min}$ Jul | -0.26 | 0.22 | 0.29 | -0.39 | -0.07 |
| $T_{min}$ Jun | 0.33 | -0.36 | -0.33 | -0.3 | 0.11 |
| $T_{min}$ May | -0.03 | -0.13 | -0.12 | -0.16 | 0.11 |
| $T_{min}$ Apr | -0.1 | -0.06 | -0.08 | 0.04 | 0.06 |
| $T_{max}$ Aug | -0.06 | -0.25 | -0.2 | -0.14 | 0.07 |
| $T_{max}$ Jul | -0.19 | -0.32 | -0.28 | -0.19 | 0.1 |
| $T_{max}$ Jun | 0.01 | -0.51 | -0.41 | -0.44 | 0.06 |
| $T_{max}$ May | -0.13 | -0.04 | -0.03 | -0.03 | -0.01 |
| $T_{max}$ Apr | -0.07 | -0.01 | -0.01 | 0.01 | 0.05 |
| Dec p | -0.19 | -0.14 | -0.11 | -0.09 | -0.25 |
| Nov p | -0.07 | -0.14 | -0.08 | -0.04 | -0.41 |
| Oct p | -0.26 | -0.14 | -0.1 | -0.05 | 0.01 |
| Sep p | 0 | -0.19 | -0.21 | -0.24 | 0.05 |
| Aug p | 0.19 | -0.12 | -0.09 | -0.13 | 0.06 |
| SPEI | 0.13 | 0.37 | 0.37 | 0.26 | 0.02 |
| Mjj pr | 0.46 | 0.49 | 0.57 | 0.46 | 0.17 |
| Jul p | -0.01 | -0.01 | 0.02 | -0.04 | -0.16 |
| jun p | -0.13 | -0.37 | 0.43 | 0.48 | 0.16 |
| May p | -0.05 | 0.29 | 0.26 | 0.14 | -0.12 |
| Apr p | 0.03 | 0.11 | 0.07 | -0.04 | -0.17 |
| Mar p | -0.14 | 0.01 | 0.03 | -0.2 | 0 |
| Feb p | -0.1 | -0.18 | -0.1 | -0.16 | -0.12 |
| Jan p | 0.13 | -0.02 | 0 | -0.02 | -0.16 |
| Dec t | 0.18 | -0.01 | 0.03 | 0.06 | -0.27 |
| Nov t | 0.09 | -0.02 | 0.14 | 0.2 | -0.02 |
| Oct t | 0.09 | -0.01 | 0.02 | 0.04 | 0.03 |
| Sep t | 0.04 | 0.11 | 0.18 | 0.13 | 0.04 |
| Aug t | -0.17 | -0.31 | -0.23 | -0.21 | 0.1 |
| Jul t | -0.23 | -0.48 | -0.46 | -0.41 | 0.11 |
| Jun t | 0.13 | -0.52 | -0.49 | -0.5 | 0.16 |
| May t | 0.21 | -0.39 | -0.39 | -0.38 | 0.02 |
| Apr t | -0.03 | -0.03 | -0.06 | -0.03 | 0.03 |
| Mar t | -0.11 | -0.04 | -0.04 | -0.03 | 0.17 |
| Feb t | 0.08 | 0.02 | 0.05 | 0 | -0.05 |
| Jan t | 0.2 | 0.14 | 0.15 | 0.15 | 0 |

**Figure 11**: Correlations of grain series vs climate indicators 1865-1911. Notes: Only statistically significant ($p \leq 0.05$) correlations are colored. Sources: BiSOS, Seftigen *et al.* (2017), Seftigen *et al.* (2020) and SMHI (2021).

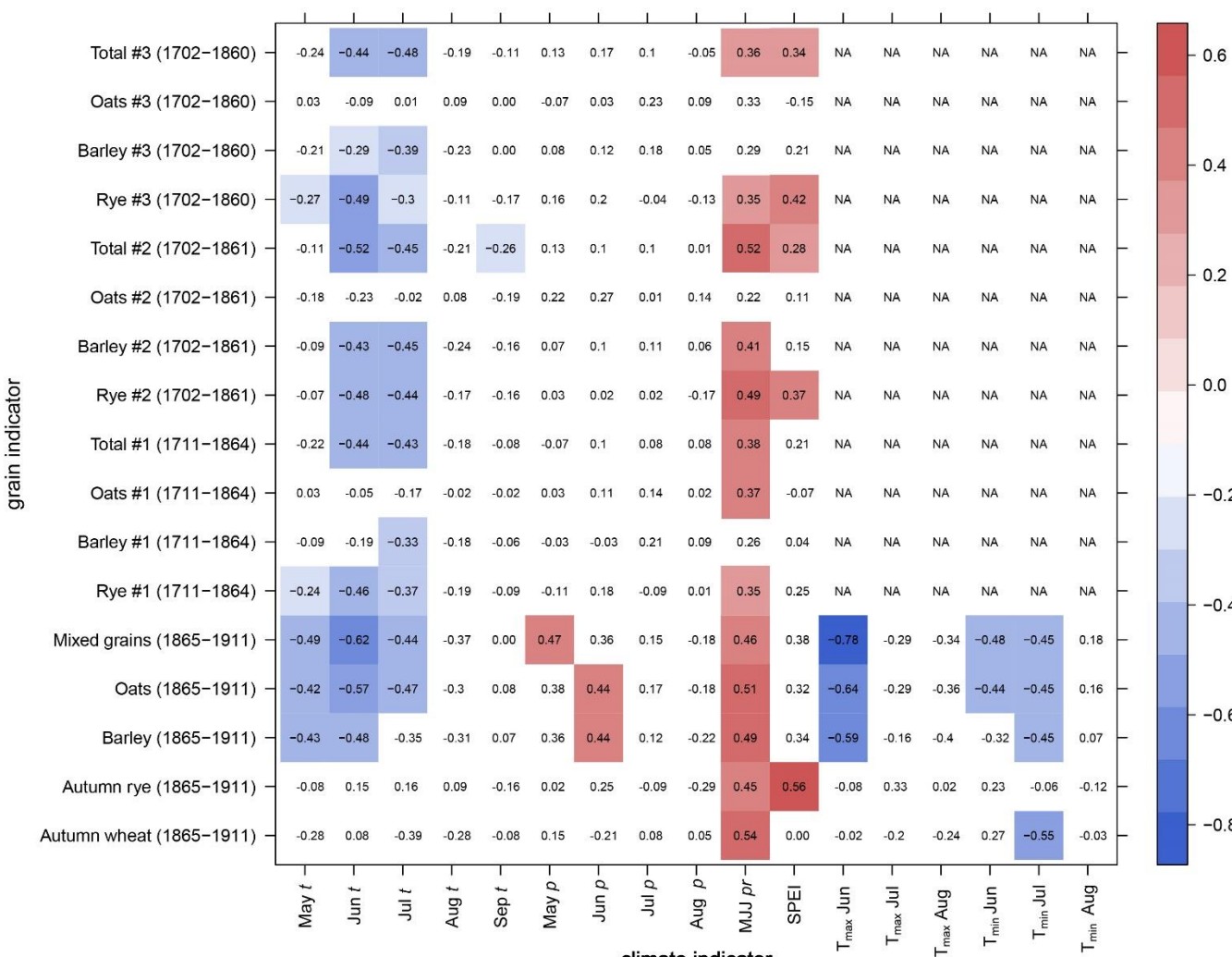

**Figure 12**: Correlations of grain series vs climate indicators c. 1702-1865/1865-1911 during relatively dry years. Notes: Only statistically significant ($p \leq 0.05$) correlations are colored. Clusters are signified by the # number. Sources: HDSA, BiSOS, Tidblom (1876), Seftigen *et al.* (2017), Sefitgen *et al.* (2020), SMHI (2021) and the homogenized monthly temperature series (APPENDIX A).

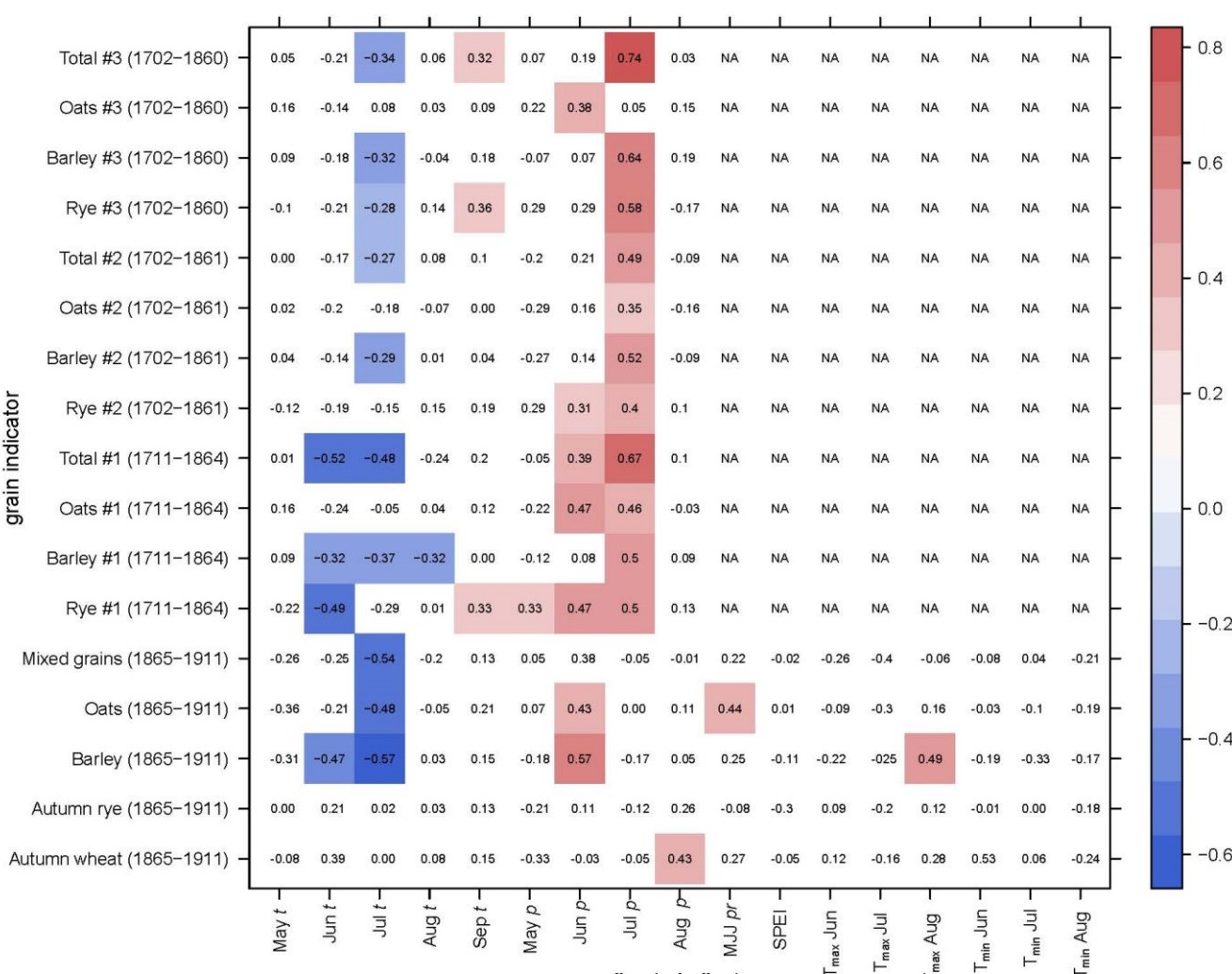

**Figure 13**: Correlations of grain series vs climate indicators c. 1702-1865/1865-1911 during relatively wet years. Notes: Only statistically significant ($p \leq 0.05$) correlations are colored. Clusters are signified by the # number. Sources: HDSA, BiSOS, Tidblom (1876), Seftigen *et al.* (2017), Sefitgen *et al.* (2020), SMHI (2021) and the homogenized monthly temperature series (APPENDIX A).


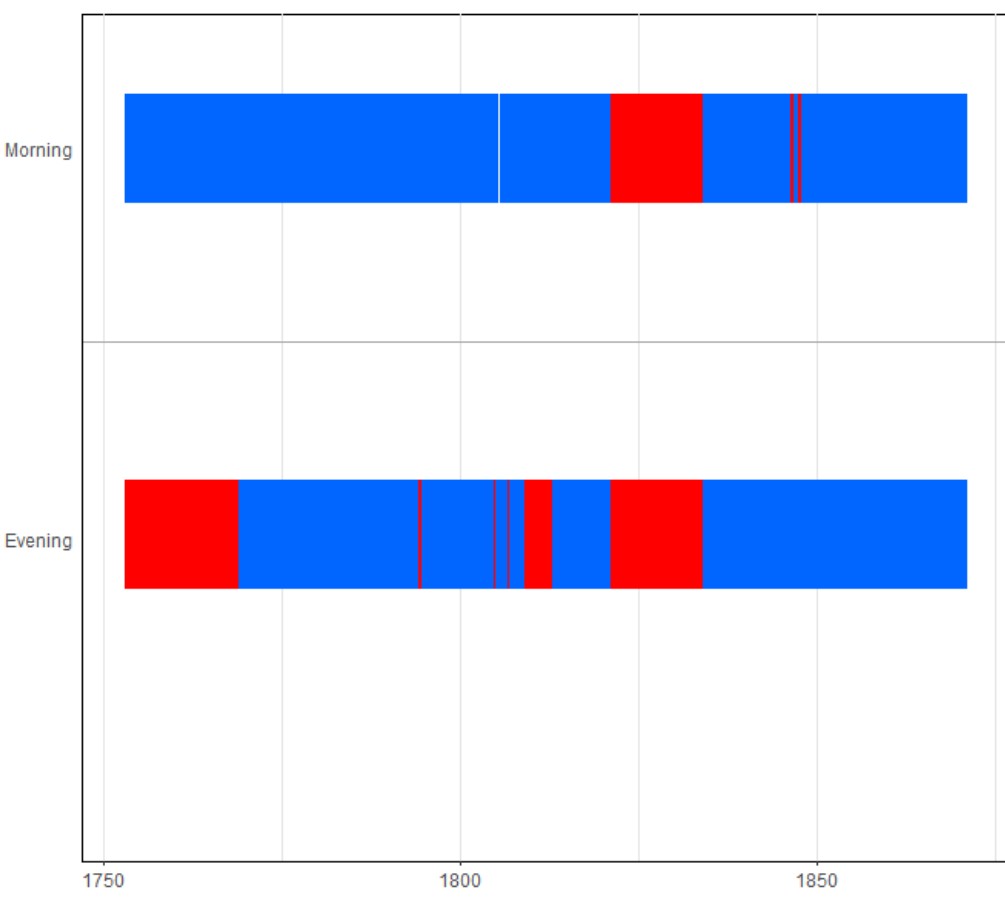


**Figure A1**: Gaps in the Lund temperature series, 1753-1870. Sources: Tidblom (1875). Notes: Blue color signifies available data and red color signifies a gap. Measurement units are in the form of pentad (5-days) averages.



**Table A1**: Correlations between monthly temperatures in Lund and other network series.

| | Copenhagen | Central England | Uppsala | Stockholm | Berlin-Dahlem | De Bilt |
|---|---|---|---|---|---|---|
| **January** | 0.93 | 0.73 | 0.85 | 0.88 | 0.83 | 0.82 |
| **February** | 0.95 | 0.78 | 0.84 | 0.87 | 0.89 | 0.86 |
| **March** | 0.93 | 0.80 | 0.83 | 0.86 | 0.90 | 0.88 |
| **April** | 0.77 | 0.66 | 0.72 | 0.71 | 0.77 | 0.80 |
| **May** | 0.69 | 0.43 | 0.72 | 0.70 | 0.76 | 0.60 |
| **June** | 0.40 | 0.16 | 0.55 | 0.51 | 0.53 | 0.45 |
| **July** | 0.56 | 0.32 | 0.59 | 0.57 | 0.61 | 0.49 |
| **August** | 0.64 | 0.41 | 0.71 | 0.70 | 0.66 | 0.59 |
| **September** | 0.45 | 0.31 | 0.60 | 0.58 | 0.64 | 0.47 |
| **October** | 0.70 | 0.51 | 0.73 | 0.75 | 0.85 | 0.79 |
| **November** | 0.83 | 0.41 | 0.73 | 0.77 | 0.83 | 0.76 |
| **December** | 0.93 | 0.76 | 0.86 | 0.88 | 0.89 | 0.80 |
| **Annual** | 0.51 | 0.67 | 0.72 | 0.72 | 0.77 | 0.80 |
| **Spatial correlation** | 0.94 | 0.62 | 0.83 | 0.84 | 0.84 | 0.75 |

Sources: Copenhagen (Cappelen *et al.*, 2019), Berlin-Dahlem (DWD, 2018), De Bilt (Durre *et al.*, 2008; Lawrimore *et al.*, 2011), Lund (Tidblom, 1876), Uppsala (Bergström & Moberg, 2002) and Stockholm (Moberg *et al.*, 2002, Moberg, 2021). Notes: Spatial correlation coefficients are obtained by ACMANT, where increment series are correlated after monthly climatic means have been removed (Domonokos & Coll, 2017).




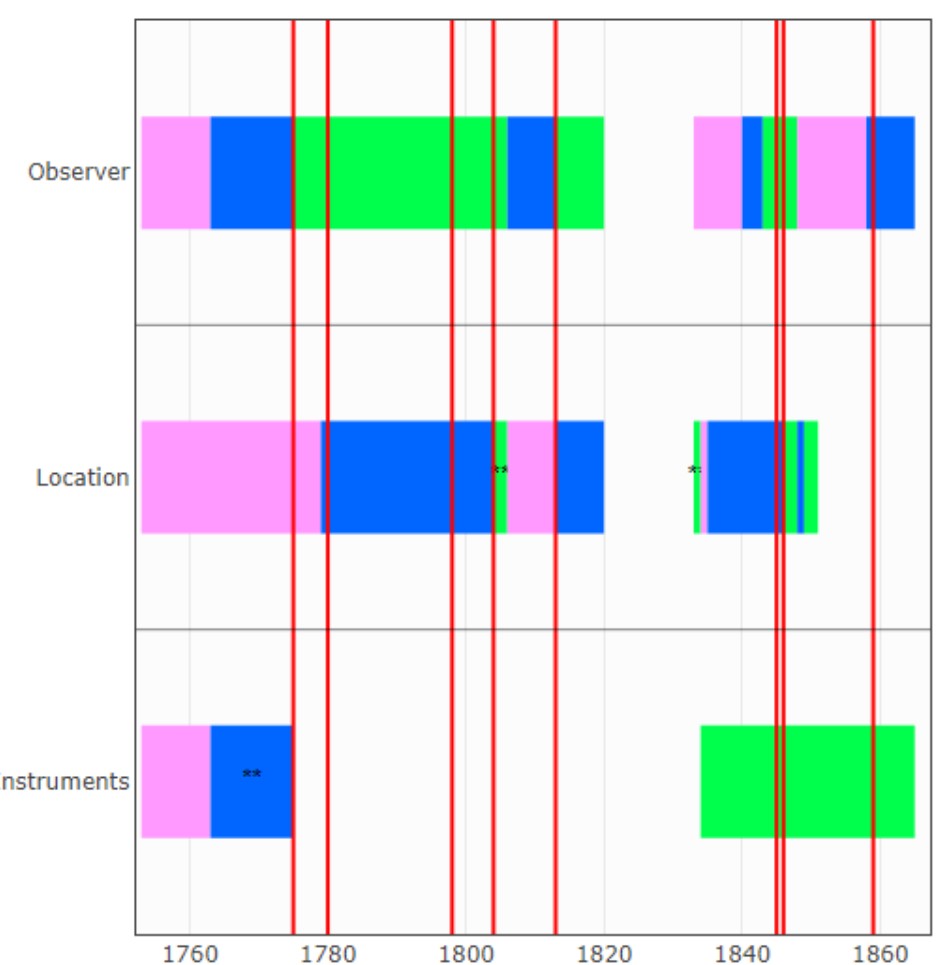

**Figure A2**: Station history and ACMANT-detected breaks, 1753-1870. Sources: Tidblom (1875), Schalén (1968) and Bärring *et al.* (1999). Notes: Each colored area represents a distinct period location, observer or set of instruments. ** denotes multiple changes during the indicated period. Detected breaks are shown by the vertical red lines. Gaps in white signify missing data.



**Table A2**: Average time of day for morning and evening measurements, 1753-1870

| Period (morning, AM) | 1753-1774 | 1775-1806 | 1807-1849 | 1850-1858 | 1859-1870 |
|---|---|---|---|---|---|
| January | 8.4 | 8.0 | 7.9 | 6 | 8 |
| February | 7.6 | 7.1 | 7.5 | 6 | 8 |
| March | 6.8 | 6.0 | 6.9 | 7 | 8 |
| April | 6.7 | 5.2 | 6.3 | 7 | 8 |
| May | 6.6 | 4.6 | 6.2 | 7 | 8 |
| June | 6.7 | 4.5 | 6.2 | 7 | 8 |
| July | 6.5 | 4.4 | 6.1 | 7 | 8 |
| August | 6.6 | 4.7 | 6.2 | 7 | 8 |
| September | 6.7 | 5.5 | 6.3 | 7 | 8 |
| October | 7.1 | 6.6 | 6.6 | 7 | 8 |
| November | 7.9 | 7.6 | 7.2 | 7 | 8 |
| December | 8.5 | 8.2 | 7.6 | 7 | 8 |

| Period (evening, PM) | 1769-1791 | 1792-1820 | 1834-1849 | 1850 JF | 1850 M-1870 |
|---|---|---|---|---|---|
|  | 9.9 | 8.5 | 9.8 | 10 | 9 |

Sources: Tidblom (1875). Notes: Single letters in the "Period"-row denotes months, e.g. JF denotes January-February.




**Table B1**: Years at or below the 33th percentile of SPEI (drier conditions) during the years 1702-1865

| | | | | | | | | | | | | | |
|---|---|---|---|---|---|---|---|---|---|---|---|---|---|
| 1704 | 1709 | 1712 | 1719 | 1726 | 1727 | 1740 | 1741 | 1747 | 1748 | 1749 | 1771 | 1774 | 1775 |
| 1781 | 1782 | 1783 | 1784 | 1785 | 1786 | 1787 | 1788 | 1789 | 1790 | 1798 | 1801 | 1803 | 1807 |
| 1808 | 1810 | 1811 | 1812 | 1816 | 1818 | 1819 | 1820 | 1821 | 1822 | 1823 | 1824 | 1825 | 1826 |
| 1827 | 1828 | 1829 | 1838 | 1843 | 1845 | 1846 | 1847 | 1850 | 1852 | 1853 | 1854 | 1855 | 1857 |
| 1859 | 1861 | | | | | | | | | | | | |


Source: Seftigen *et al.* (2017).


**Table B2**: Years at or above the 67th percentile of SPEI (wetter conditions) during the years 1702-1865

| | | | | | | | | | | | | | |
|------|------|------|------|------|------|------|------|------|------|------|------|------|------|
| 1706 | 1707 | 1708 | 1713 | 1715 | 1717 | 1718 | 1720 | 1722 | 1725 | 1730 | 1731 | 1732 | 1733 |
| 1734 | 1735 | 1736 | 1737 | 1738 | 1742 | 1744 | 1745 | 1750 | 1751 | 1752 | 1753 | 1754 | 1755 |
| 1756 | 1757 | 1759 | 1761 | 1763 | 1764 | 1765 | 1766 | 1767 | 1769 | 1776 | 1777 | 1778 | 1779 |
| 1793 | 1794 | 1797 | 1799 | 1805 | 1815 | 1817 | 1834 | 1835 | 1851 | 1860 | 1862 | 1865 | |

Source: Seftigen *et al.* (2017).




**Table B3**: Years below the median of SPEI (drier conditions) during the years 1865-1911

| 1868 | 1873 | 1874 | 1875 | 1876 | 1877 | 1879 | 1880 | 1881 | 1883 | 1887 | 1888 | 1889 | 1891 |
|------|------|------|------|------|------|------|------|------|------|------|------|------|------|
| 1893 | 1895 | 1896 | 1901 | 1902 | 1904 | 1905 | 1906 | 1911 |      |      |      |      |      |

Source: Seftigen *et al.* (2017).


**Table B4**: Years above or at the median of SPEI (wetter conditions) during the years 1865-1911

| 1865 | 1866 | 1867 | 1869 | 1870 | 1871 | 1872 | 1878 | 1882 | 1884 | 1885 | 1886 | 1890 | 1892 |
|------|------|------|------|------|------|------|------|------|------|------|------|------|------|
| 1894 | 1897 | 1898 | 1899 | 1900 | 1903 | 1907 | 1908 | 1909 | 1910 | | | | |

Source: Seftigen *et al.* (2017).
