# Peer review of "Climate variability and grain production in Scania, c. 1702-1911"

_Climate of the Past, 2021_

## Author Response (AR1)

CP-2021-52
**Climate variability and grain production in Scania, c. 1702–1911**
Martin Karl Skoglund
Special Issue: International methods and comparisons in climate reconstruction and impacts from archives of societies
Handling Editor: Chantal Camenisch, chantal.camenisch@hist.unibe.ch

**Author's response**

In the following, I have divided the RC1 and RC2 into distinct answerable parts, where I answer each part separately. Two reference lists are used, one for each authors response section. Referees are denoted RC1 and RC2 and the Author's response is denoted AC1 and AC2 in line with abbreviations used in the official discussion section for the article at https://cp.copernicus.org/preprints/cp-2021-52/. In the end of the document there is a list where all changes made to the manuscript by the author are summarized.

**Comments from RC1 and answers from AC1**

**RC1 part 1:**

"This study analyses the relationship between climate variability and grain production in southern Sweden for the 18th and 19th centuries. In the long introductory part, the author provides detailed information about the farming system of the study area, historical background, natural conditions, grain crops, and their varieties. He stressed the importance of crop diversity for stable crop production. It is interesting that reversed relationship of the crop production to temperature compared to other parts of Scandinavia, as well as other parts of Europe, was found in this study.

Historical database of the tithe records is used for the period before 1865 while official statistics on county level were utilized after that year. Data on grain production from the historical database were pre-processed in several steps attempting to solve some biases and sources of uncertainty (normalizing, de-trending). However, not all pre-processing steps are sufficiently explained. For instance, what is the role of the "threshing coefficient". While the reasons for aggregating data are well explained, the application of the cluster analysis (Section 2.2) is rather strange. How it was decided that exactly three clusters are optimal? The final number of clusters was decided subjectively or any objective measure was used? While individual villages are clustered, the map in Figure 3 presents administrative units belonging to different clusters. It would be useful to explain more clearly, whether the BISOS data from the 1865–1911 period were standardized in a similar way as the HDSA data."

**Answer**:

Historically, the amount of grain obtained after threshing varied in different localities across Scania (in fact this is a common phenomenon in historical agriculture in general). The tithe in Scania was collected *before* threshing. Olsson & Svensson (2017a, referenced in the article), have thus constructed threshing coefficients for different parishes and type of farming districts in Scania, allowing a more accurate conversion from *harvested volumes* to *threshed volumes*, the latter of course being a more accurate measure of the final output of grain production. These threshing coefficients are based on actual threshing accounts found in various sources, e.g. manorial archives, church archives and probate inventories. The threshing coefficients can be found at https://www.ekh.lu.se/en/research/economic-history-data/HDSA-1702-1881 in the "STATA do-file.txt" (last accessed 2021-06-16). In the article, the threshing coefficients thus influences the relative production levels in the analyzed villages for the period 1702-1865, which for instance can be seen in the average grain production for each cluster shown in Fig. 5.

Regarding the clusters, it is correct that the hierarchical cluster analysis (HCA) sorts individual villages. The administrative units shown in Fig. 3 are parishes, which in general are constituted by a number of villages. In some parishes, different villages were sorted into different clusters in the HCA, leading to some parishes being represented by multiple clusters. This is not an issue, since in this example, the parishes are merely employed as a descriptive tool (like the type of farming district and the institutional status of villages) to describe the more abstract (from an historical point of view) clusters. This is discussed in the article (lines 326-329). Finally, regarding the number of clusters in the HCA. The HCA is often referred to as an explorative approach, given that

the number of "true" clusters is not known. Even though there have been multiple methods proposed for optimizing the number of clusters, it remains difficult to know *a priori* what the best metric to use is. I used one of the more common metrics, e.g. the *gap statistic* (Tibshirani et al. 2001). Specifically, I used the clusGap function in the factoextra package in *r* setting a maximum number of potential clusters at 10, yielding the optimal number of three clusters after 100 bootstrap samples (Kassambra & Mundt, 2020). Following the logic of trying to reduce the interpretative and descriptive issues related to HCA (discussed in lines 326-329), this number can be compared to the most common categorization employed in the historical literature when discussing relative specialization in production, i.e. the type of farming district. In Scania three different types of farming districts have been conceptualized, namely the forest districts (sw. *skogsbygd*), the mixed districts (sw. *risbygd*) and the plain districts (sw. *slättbygd*) (see references in the article, e.g. Campbell, 1928, Dahl, 1989 or Bohman, 2010). As can be seen in Fig. 4 there is some relationship between the clustering results, although it is quite limited.

**RC1 part 2:**

The manuscript is not well structured. While the introductory part is rather long, with numerous details, methods are mixed with results in Sections 2 and 3. Section 2.4 is followed by Section 3.1. Some section titles do not correspond to the following paragraphs (e.g. Section 1.4.1). Results of correlation analysis (Section 3.1) are mixed with discussion (e.g. lines 496–504). Results are presented in the form of several correlation matrices (Figs. 9–13). Correlation coefficients between crop production series and temperature/precipitation/drought characteristics were calculated for two different periods, for different clusters and different crops. And these correlations are repeated for "dry" and "wet" years. The description and interpretation of such results are rather long and not very synoptic. It is very hard to orient in the text and to find any signal in presented correlations. I miss any information which correlations are statistically significant (those in colored boxes?) and on which level (p-values)?

**Answer**:

The motivation or idea for the initial structure of the paper was that the hierarchical cluster analysis and other data homogenization procedures was part of how the data was managed, while the results were more delimited to the statistical correlations between grain production indicators and climate indicators. However, given that parts of the method-section contains actual results this initial structure could be improved in line with the commentators comments by more clearly separating methods and results. I agree with the commentators note that the lines 496-504 more appropriately belongs to the discussion section.

The commentator points out some other errors in the structure of the pre-print. For example Section 2.4 should be followed by a heading titled Section 3 and not Section 3.1. An important error of omission in Figs. 9-13 pointed out by the commentator is that it is not specified what correlations are significant. In the notes to each figure it should be clearly stipulated that only significant correlations are colored (p ≤ 0.05). Adding this will also clarify the signals discussed in the text. The 'signals' corresponding to the patterning of colorations in Figs. 9-13.

**RC1 part 3:**

In spite of a relatively long and detailed introductory part, I miss any direct information on harvest dates (or threshing dates?). This date may indicate the time when the grain production of the given year was determined. Harvest dates are mentioned only indirectly in August and September (lines 199 – 201). In this sense, a correlation of a grain harvest with Oct, Nov, and Dec climate of a given year seems to have no meaning. Contrary, it would be more meaningful to correlate the grain harvest of a given year with Dec, Nov, and Oct climate of a previous year. It would be useful especially for crops sown in autumn.

**Answer**:

Regarding the months of October, November and December, it is described in lines 453-454 that I use lagged (i.e. previous year) values. In the previous lines, 452-453, I briefly mention that harvesting was usually *completed* in late August or during September. However, I would agree with the commentator that the time of harvesting (and sowing) are quite important and some elaboration is probably in order.

There is generally a lack of sources on harvesting (and threshing dates) for Scania before the 20[th] century. Even in the official statistics, BiSOS 1865-1911, there is a lack of harvest dates (even though reporting on harvest and sowing dates were part of the forms that were sent out for the collection of agricultural statistics). Most likely, there was a relatively large variation across parishes and villages when actual harvesting was begun. However, in farmers' diaries and some parish descriptions from the 19[th] century one can find some specific as well as general notations on harvest, sowing and threshing dates. For example, in the parish description of Husie socken in southwestern Malmöhus from 1826, the parish priest Carl Carlsson noted that the climate of Husie *village* (the village and parish had the same names) was mild and relatively humid and undoubtedly beneficial for arable farming. Sowing was usually started between the middle of April until the middle of May (Andersson, 1986). In a local farmers diary from the same village, the farmer Anders Andersson on the farm Skrävlinge nr 1 wrote down sowing and harvesting dates during a some years in the early 1820s (Andersson, 1986). In 1821, sowing of barley and mixed-grains (barley and oats) started on the 18[th] and 15[th] of May, respectively, and lasted roughly a week (ended on the 22th of May). In 1823, peas, oats and vetches were first sown on the 14[th] of April and barley and mixed-grains on the 15[th] of May (finished 27[th] of May). In 1826 barley was sown starting on the 12[th] of May. In most of these years, rye sowing started in the beginning of October. Regarding harvests, they were begun on the 21[st] of August in 1821. In 1826, due to excessive heat and drought, the harvest (of at least rye) was begun on the 25[th] of July.

In a collection of parish descriptions from Malmöhus Län in 1828 there are short summaries of general sowing and harvesting dates for a little more than a dozen parishes. According to these descriptions, sowing of oats started between the middle of April until late May, depending on the parish and current weather conditions. Barley sowing was initated from late April until early June, albeit mostly in the 2[nd] half of May. Sowing of autumn-rye began somewhere between the middle of August until early September, and in some instances in late October, however mostly in the latter part of September. Harvesting generally started during August or sometimes in early September, or in late July for the autumn-rye crop (Bringeus, 2013).

There appears to be some differences between the forest and plains district as well as parishes in different latitudes. However the number of available sources presented here are too few to say anything with certainty in this regard. In relation to the discussion in the article of the potential similarity of barley landarces in southern Sweden and more northerly parts of Scandinavia, it is interesting to note that sowing and harvesting dates of barley in the northerly province of Jämtland are quite similar to those available from Scania, i.e. sowing in the middle of May and harvesting in August. This could possibly be inserted into the discussion. Furthermore, in line with the commentators points, I could add general sowing and harvesting dates for Scania (to the extent they are available), as well as shorten the introduction.

**RC1 part 4:**

While the analysis of the relationship between grain production and climate is based on simple correlation analysis, a relatively extensive discussion mentions a number of aspects that were not analyzed in this study. For instance, lines 584-585: "... the absence of a climate signal in the spring and autumn months, as well as the last summer month of August to some extent. This could be interpreted as spring or autumn frosts not being a systematic threat …" However, correlation does not mean causality and it would be correct to add some info about the frequency of spring/autumn frosts.

Similarly, lines 630-631: "If conditions were relatively wet or dry in the early summer, the effects from subsequent precipitation and temperatures later in June and especially July would theoretically have been amplified." Such claims should be supported by their own results and / or citations from other similarly focused studies.

**Answer:**

The formulation referred to on lines 584-585 could be amended to further clarify that it is merely correlations and not causal relationships being discussed. However, I would emphasize that what I write on the specified lines is that it *could be interpreted*, i.e. it is merely one possible interpretation of the statistical associations. Furthermore, I agree that it would be correct to add information in the article on the frequency of spring/autumn frosts. This can be done using data from the second half of the 19[th] century when daily temperature data is available.

Regarding the comment on lines 630-631, it is a theoretical stipulation but also partly based on the results in the article (see Fig. 12 and Fig. 13) and the literature (see for example Brunt, 2004, on the benefit of rain spread out across the growing season). It is theoretical, or hypothetical in the sense that it is based on my speculations on the possibility that there is a skew towards early summer conditions in the SPEI-reconstruction used in the article. Perhaps an amendment is in order to clarify the hypothetical nature of what is being discussed on the specified lines?

**RC1 specific comments:**

- Map of the study area with outlined geography mentioned in the text would be very useful

Answer: This could possibly be added, however it is a relatively comprehensive task given that there exists to good geographical data relevant to 18[th] and 19[th] century land use. For example, there was large changes (reductions) in forested land and (increases) in arable land.

- Section 2.1 refers to clusters. However, clustering is explained later in Section 2.2.

**Answer**: Should be amended.

- Homogenization of the Lund temperature measurements from 1753 is an important by-product of this study. However, this series was also extended further back to 1701 and there is no information about the validation of this earliest part of the "calculated" series. Looking at Figs. 6 and 7, the calculated series (before 1753) seems to have lower variability compared to part of measured temperatures (after 1748) for all seasons. Was the variability of the calculated series adjusted in any way?

**Answer**: The larger variability seen in Figs. 6 and 7 from 1753 can only be seen in the graphs representing the raw series; looking at only the graphs for the homogenized series no such discernible shift can be seen.

- It would be useful to add some information to Table 2 about the length (N) of the period that was used to calculate correlations between the Lund series and the other temperature series. Are there all correlations statistically significant? At that level?

**Answer**: Common periods were used to calculate correlations, which are all significant at the $\leq 0.05$ level. The spatial correlation coefficient in this context is mainly important for the purposes of the ACMANT procedure, see answer below.

- The problem of homogenization of the early instrumental temperature measurements closely relates to the so called "the early instrumental warm-bias" (see e.g. Böhm al., 2010). It would be useful at least to comment on it. While in the Greater Alpine Region warm bias was found especially in summer, this study found "cold" bias for Lund.

**Answer**: Such a comment could be added.

- Table 2 – It is not clear, how the spatial correlation was calculated and why it is listed on the last row of the table. One would expect that it is a correlation between Lund and neighboring stations (or some spatial temperature field?). In this sense, it would be listed as a separate column, not a row.

**Answer**: The notes for Table 2 shortly describes the method for how the spatial correlation is calculated in ACMANT. Only one spatial correlation coefficient is obtained per station, which is why it is added as a row and not a column in Table 2. The spatial correlation is important because it shows whether a station series is a valid input into the ACMANT homogenization procedure. It is described in more detail in the AMANT manual and ACMANT scientific description referred to in Domonkos & Coll (2017).

- From correlation analysis it follows that explained common variance (r-squared) is mostly negligible. For instance, when the correlation coefficient r=0.3, grain harvest and climate share less than 10% ($r^2$=0.09) of the common variance. It would be useful to explain more (or even quantify) the role of other factors. Some of them are mentioned in the discussion.

**Answer**:

It is not unproblematic to jump from a correlation coefficient obtained from a correlation analysis to an $r^2$ without a properly specified regression model (which carries with it some different assumptions). This article is mainly concerned with estimating the extent of relationship between climate variability and grain production. It is not, as the commentator correctly observed in a previous comment, to establish specific causalities between climate and grain production. Whether a correlation coefficient of 0.3 in this context is mostly negligible or not is difficult to determine, however one can subjectively argue one way or the other. If a correlation coefficient of similar magnitudes is obtained across the same months, in different samples, data-sets, time periods as well as different crops, I would argue that the relationship was probably not *historically* negligible. I would further emphasize that in the context studied here, the margins in production were important. There is also the issue discussed by Beillouin et al (2020), and Edvinsson et al (2009) noted on lines 236-237 in the article, that a lack of detailed climate data usually leads to an underestimation of the impacts of climate variability on grain production. For example, in this article, I do not have access to daily temperature and precipitation data in the pre-1860 period, nor do I have minimum and maximum values at a sub-annual level.

Regarding the direct inclusion of other factors, this is a task that is beyond the scope of this article and would require a study of its own, probably at a much lower level of analysis (e.g. looking at only a smaller subset of villages). One of the more important factors in this context is probably soil. To obtain more detailed soil data would require extensive efforts to rectify soil maps with historical land use maps, and even then it would remain unclear which crop was grown at a particular plot of land in a given year. Nor is historical land use maps available for all villages in the sample, and for those villages that there are available maps, they cover only a few select years. Other possible and largely time-invariant factors include institutional set-up, farming system and other geographical factors like type of farming district. However, besides soil types, there is a theoretical gap in the literature as to why and how such other factors would affect the relationship between climate variability and grain production. This being said, most of these factors are actually indirectly included in the analysis, see the description of the different clusters in Fig. 3 and 4, Table 1 and lines 333-383.

**RC1 technical corrections:**

- Lines 25-25: reference to "Huhtamaa & Helama, 2017b" is mentioned twice in the list

**Answer**: Should be fixed.

- Line 34-35: correct to (Osvald, 1959; Persson, 2015).

**Answer**: Should be fixed.

- Line 78: correct to: … century.

**Answer**: Should be fixed.

- Line 97: "…the suly of winter fodder". Please check. Is it correct?

**Answer**: Should be corrected to 'supply'.

- Line 109: „starting in 1749/1757" – this is not clear

**Answer**. Should be clarified to 1757.

- Footnote 4: „The share of oats WAS quite low"

**Answer**: Should be fixed.

- Line 289: please unify: BiSOS or BISOS?

**Answer**: Should be unified to BiSOS.

- Line 302. Normalized production anomalies is abbreviated asNPAa, but differently in formula (1) and in the text (line 306)

**Answer**: Should change normalized production anomalies on line 306 to NPA.

- Line 347: The data … has subsequently incorporated … Please check

**Answer**: Appears to be in order to the author.

- Line 358: "four clusters" – Should not be "three"?

**Answer**: Correct, should be three.

- Line 382: "…increase in an ascending order …"

**Answer**: Appears to be correct to the author.

- Line 447-8: What is the meaning of „simple" climate variables?

**Answer**: "Simple" climatic variables in this context refers to the mean values at seasonal, monthly or annual levels of temperature, or the sum of precipitation at the same time-scales. An example of adding complexity would be to convert these values into deviations from a long-term or moving average. Please see the discussion and references on lines 443-448.

- Line 459: What is the meaning of "…most consistent coefficients"?

**Answer**: That the direction, magnitude and significance of coefficients are consistent across samples, time-periods and different crops. Could be changed to for example: "producing the clearest signal".

- Please check figure and table captions, correct and complete. For instance, there are no „Descriptive statistics" in Table 2, Figure 5 – "…estimated loess" is not clear. The loess function is used here as a low-pass filter. Figure 7 – correct the caption – figure relates to DJF and SON seasons. Figures 9 – 13 – please add information about the statistical significance of the correlation coefficients.

**Answer**:

Descriptive statistics could be added to Table 2. The extent of the series are described on lines 392-397. Fig. 5 shows the estimated loess (smoothed lines) and the NPAs (point-lines) for each cluster and crop (see legend). Legend could be adjusted to e.g. "barley NPA" and "barley loess"? Fig. 7 should be fixed. Fig. 9-13 should be fixed (see above).

**References**

Andersson, H.: Huse- Lantsocken som blev stadsdel. Några anteckningar kring en sockenbeskrivning från 1828. Elbogen 16 (1-4), 1-24, 1986.

Beillouin, D., Schauberger, B., Bastos, A., Ciais, P. and Makowski, D.: Impact of extreme weather conditions on European crop production in 2018. Phil. Trans. R. Soc. B 375:20190510, doi: 10.1098/rstb.2019.0510, 2020.

Bohman, M.: Bonden, bygden och bördigheten: Produktionsmönster och utvecklingsvägar under jordbruksomvandlingen i Skåne ca 1700-1870. Diss. Lund University: Lund University, oai: DiVA.org:umu-99296, 2010.

Bringeus, N. A.: Sockenbeskrivningar från Malmöhus Län 1828. Arcus Förlag, 2013.

Campbell, Å.: Skånska bygder under förra hälften av 1700-talet : etnografisk studie över den skånska allmogens äldre odlingar, hägnader och byggnader. Uppsala: Lundequistiska bokhandeln, 1928.

Dahl, S.: Studier i äldre skånska odlingssystem. Stockholm: Stockholms universitet, 1989.

Edvinsson, R., Leijonhufvud, L. and Söderberg, J.: Väder, skördar och priser i Sverige. In: Liljewall, B., Flygare, I., Lange, U., Ljunggren, L. and Söderberg, J. (eds) Agrarhistoria på många sätt. 28 studier om människan och jorden. Kungl. skogs- och lantbruksakademien, 2009.

Kassambra, A. & Mundt, F.: Package 'factoextra'. https://rpkgs.datanovia.com/factoextra/index.html last accessed 23 June 2021. R package version 3.1.2, 2020.

Tibshirani, R., Walther, G. & Hastie, T.: Estimating the number of clusters in a data set via the gap statistic. J. R. Statist. Soc. B, 2001.

**Comments from RC2 and answers from AC2**

**RC2 part 1:**

General comments

The article studies the relationship between climate variability and gran production in Scania (southern Sweden), between 1702 and 1911. The study also claims, in the abstract, that it will shed new light on the climate history of the region by homogenizing the Lund instrumental series. The article, therefore, has several goals and is ambitious. As some of my questions, have been dealt with in the discussion already, I try to focus on some other issues that could improve the manuscript. The manuscript has potential, but the manuscript could use a thorough revision when it comes to structure. In its current form, it is difficult to follow the authors line of thought. Sentence structure is often speculative and results and approaches needs to express intent more clearly.

My initial impression was that the manuscript would be improved if it was written as two separate articles and it would make it easier for the author to focus on grains and climate. The first article could present the homogenization of the Lund series, which then could be used as a background for a more in-depth approach to write the second article, which would compare climate variability and grain production. As it reads now, the meteorological series from Lund is not actually mentioned until page 14, which is followed by an analysis, results and discussion (section 2.3). As such, the manuscript provides new perspectives and approaches, but the manuscripts structure and formulations need to be improved.

The reason for my suggestion to write two articles instead of one article is that I am unfamiliar with the Lund series. Does it cover the 1753-2020 period (the end period is not mentioned)? What are the gaps (months, years, days?) mentioned on line 386? This section is extremely condensed, and I would prefer if the series was introduced more thoroughly. As it is, there are no references to the series. Also, it reads somewhat peculiar when it is noted that there are series from 'nearby regions' (line 392), mentioning both Uppsala (c. 500km) and Central England time series (c. 1000km), and I wonder if the distance of 500km between Lund and Uppsala, or 1000km between Lund and England could be considered 'nearby'? This is a matter of semantics, however, I think that the series and the correlation deserves a more in-depth discussion than this. Especially considering the results and the correlations in Table 2. The results are interesting. It is intriguing that the correlations during the months May-October, between Lund and Stockholm, Uppsala and Berlin-Dahlem are stronger than those with neighboring Copenhagen (c. 40km). Why is this? Is it a question on instrumental reliability, series validity or climatic factors? Answering this question is not the purpose of this article, but it seems quite central for the analysis and series in general. Moreover, in Figure 6 and 7, the series is extended to 1701 even though this is not mentioned earlier. In summary, the approach is interesting, but it needs more context. Finally, on page two, it is said that the climate in Scania resembles that of England. Why then is the correlation with the Central England time series the lowest of all? What part of the 'relatively mild' (line 33) Scania climate is comparable to England? Also, what is the correlation period analyzed in Table 2, is it the studied period 1753-1922 or 1753-2020? A general description of the climate in Scania, without comparisons to other European regions, could make it clearer.

**Answer:**

The main purpose of the climatic reconstruction and homogenization is to obtain climate indicators relevant to grain production in Scania during the study period. This is why the reconstructed temperature series is extended back to 1701, which should be explained more clearly in the manuscript. Why I do not attempt a similar approach for precipitation is explained on lines 426-428.

The instrumental temperature measurements in Lund began in 1753 for daytime measurements and in 1768 for measurements in the evening and continues until today, albeit with a few gaps and inhomogeneities, as discussed in the manuscript (see lines 385-388). Efforts to standardize and make the measurement process more transparent and scientific were undertaken in the last decades of the 19th century (see Tidblom, 1876). The series can thus be divided into two parts, the early instrumental measurement period from 1753 until 1860 (measurements until 1870 were published by Tidblom, 1876, see line 387 in beginning of section 2.3) and the latter instrumental period after 1860. The series for the latter period is referenced at lines 442-443. This periodization is mirrored by those in the data sets on grain production, i.e. 1702-1865 and 1865-1911.

Regarding the distance between the areas of temperature measurements, it is quite common for temperatures to be spatially correlated across large areas at a monthly or seasonal time-scale. This can be seen for example in Tab. 2, which is why it is possible to bridge series from stations many hundreds of kilometers distanced from each other. Examples in previous research of this are for example Parry & Carter (1985) who bridged Edinburgh to Central England, roughly 400 km apart, Nordli (2004) who interpolated from Uppsala to Trondheim at 550 km apart, and Dobrovolny et al. (2010) who reconstructed Central European temperatures using instrumental data from Switzerland, Germany, Austria and the Czech Republic (including Vienna and Geneva 800 km apart).

For the method employed in the manuscript, the most important factor for homogenization and filling of gaps is the number of network series, which should be at least four, and spatial correlations should be at a minimum 0.4. With higher spatial correlations and more network series, the results become more robust. As can be seen in Tab.2, Lund and Copenhagen has the largest spatial correlation. Nevertheless, the reviewer brings up the question of why summer temperatures in Copenhagen has a lower correlation coefficient with those in Lund compared to Stockholm, Uppsala and Berlin-Dahlem (while also noting that answering this question is not the purpose this manuscript). Furthermore, the reviewer asks whether the differences are due to instrumental reliability, series validity or climatic factors.

First of all, the results of the ACMANT procedure should not be taken as a questioning of the instrumental reliability and validity of the network series. In the procedure, the network series are assumed to be homogenous (Domonokos & Coll, 2017). As noted on lines 390-394, all the network series have been subjected to homogenization efforts. However, the Copenhagen series is subject to more gaps than the other series. The Copenhagen series starts in 1768 and has gaps between the years 1768-1781, 1789-1797, meaning that from a statistical point of view it is more susceptible to outliers. Regarding the validity and reliability of the Lund-series itself, it is discussed on lines 385-387 and 410-419, and the purpose of the homogenization of the Lund-series in the manuscript is of course to improve its reliability and validity.

The last potential factor the reviewer brings forward to explain the differences in correlation coefficients between the series is climate. As can be seen in Tab. 2, there is a clear reduction in the correlation coefficients between Lund and the network series in the summer months compared to the winter months. In other words, there is a climatic seasonal aspect to the general spatial correlation. Given all this, it could be that the correlation coefficients between Lund and Copenhagen is affected by some outliers, yielding slightly lower coefficients compared to the other network series.

With respects to gaps in the early instrumental period, i.e. 1753-1870, there is one large gap for all measurements between the years 1821 and 1833, as well as some smaller gaps in daytime measurements February 1794 as well as between April and September in 1846 and 1847. For evening measurements there is one gap between 1809-1813 and some smaller gaps in February through July in 1794, December 1801, parts of August and from middle of September until the middle of November in 1804, late August and early September, most of October and a dozen days in November in 1805 as well as September and October in 1806. The series is referenced in the beginning of section 2.3 at line 387, see Tidblom (1876).

The reviewer had the impression that the manuscript could be divided into two parts, because it would make it easier for the author to focus on the relationship between grain production and climate variability. The reviewer also points out that the Lund series is not mentioned until page 14. Currently, sections 1.3, 1.4, 2.4, 3.1, 3.2 and 3.3 are almost entirely devoted to the subject. Furthermore, section 2.1 and 2.2 are entirely devoted to producing series on grain production. Section 2.3 revolves around the climate data, most of which is devoted to the homogenization and gap filling of the Lund temperature series. Hence, I would argue that the manuscript is already very much focused on the relationship between grain production and climate variability, as almost all sections are devoted to it, and the other sections are necessary building-blocks in that venture. That being said, I propose to write an Appendix, where I elaborate and present more descriptive statistics on the temperature series homogenization and gap filling, including discussing the apparent warm-bias found in other series in central Europe, and the apparent cold-bias found in this study for the late 18th century.

Regarding the resemblance of the climate of Scania to England (and northern France) on page 2, the statement will be revised. Climatically, Scania is usually categorized as having a continental climate, more comparable to central and eastern Europe than England and northern France that have an Atlantic maritime climate, the latter type of climate being warmer in the winter and, on average, slightly colder in the summer than the continental climate (Metzger et al, 2005). The intended purpose of the statement metioned above (appears on line 31, see also the whole paragraph on lines 30-40) was to highlight the capacity for intensive grain production (where climate

together with other factors, most notably soil, are the most important determinants), which is comparable to areas of England and northern France, as well as large parts of central and eastern Europe.

I agree that the reviewer is correct to question the comparison to England and France (although I would insert here that two regions can have a similar climate but experience different climate variability, this is especially clear in regards to precipitation). The more apt comparison in this introductory contextualizing part of the manuscript would be to compare Scania to areas characterized by intensive grain production and a similar climate in continental Europe (see Metzger, 2005 for a commonly cited climatic stratification of the environment in Europe).

Regarding the capacity for intensive grain production, there are no robust historical estimations of such to my knowledge, although Olsson & Svensson (2010) compares the development of agricultural production in Scania to other areas in Europe during the 19th century. Slither van Bath (1963) collected grain yield ratios for a wide array of localities across Europe, including Scania. However, these do not control for the sowing intensity per area unit, and thus not useful for this type of comparisons. There are several contemporary estimations of the capacity of grain production or long-term winter-wheat yields (e.g. EEC, 2011; Boogard et al, 2013:135,138 & Schils et al, 2018:114).

Finally, regarding the reviewers question on Tab. 2: it shows the correlations between Lund and the network stations in years during the homogenization period when there are available measurements from Lund (i.e. 1753-1870, excluding the gap years of 1821-1833).

**RC2 part 2:**

In section 1.1. There is quite a lot of focus on climate in the 16th and 17th centuries whereas climate development during the 19th century, receives very little space. Maybe Figure 1 could be employed for a more extensive discussion on climatic development? The section could be strengthened by placing focus on the studied period.

**Answer:**

It should be pointed out here that the period that receives the most attention in section 1.1 is the 18th century (lines 75-85), which forms the first half of the study period. I agree that the 19th century should receive more attention. In line with the comments made by RC 1, I proposed that a description of the occurrence of frost in the growing season in Scania could be added to the manuscript, using 19th century data. In relation to this, I could expand upon the climate in the 19th century.

**RC2 part 3:**

Section 1.2. Farming in Scania. The theme is farming and harvesting, but I would like to see more information on harvest dates and threshing, especially as it is of great relevance for the analysis and the clustering (and because Scania seems to differ from other parts of the Nordic countries?). Some of this is already mentioned in the discussion with RC 1. Adding to this discussion, maybe there could be a sentence or to describing the length of the growing season (line 589)?

**Answer:**

To my knowledge, there is no research that clearly shows that sowing and harvesting dates in Scania differ significantly from other parts of Scandinavia. While I do point out in the manuscript that Scania has a longer growing season than other parts of Scandinavia, this does not imply that sowing and harvesting dates differ correspondingly. As I argue in the manuscript, any such account has to control for the type of crop variety being cultivated. In my own unpublished investigations, I have found little differences in sowing and harvesting dates in Scania and other parts of northerly Sweden or parts of Norway, at least for barley and other spring grains. For autumn-rye, harvesting dates are also similar, whereas sowing dates vary to a greater extent. I argue in the manuscript, in line with archaeobotanical research, that most of the barley and rye varieties cultivated in Scania were similar to those grown in more northerly parts of Scandinavia, at least until the 19th century. See the response to RC1 for more information on harvest and sowing dates in Scania, where I also agreed with the reviewer that they could be added in a summarized version to the manuscript. Regarding threshing dates, they are even more rare than harvest and sowing dates, and presumably they are much more variable than the latter. On line 295 I note the important fact that tithes were collected before threshing.

**RC2 part 4:**

Section 1.4 mostly seems like a presentation of previous research and I wonder if this could be presented earlier in the manuscript as part of 'previous research', especially the presentation of the research conducted by Edvinsson et al. (2009).

**Answer:**

Section 1.4 indeed is a summary of previous research on the theme 'crop diversity and resilience' that is relevant to the manuscript. As it stands, the section appears well situated where it is to the author. However, the research by Edvinsson et al. (2009) and Utterström (1957) could be presented in a summarized form earlier in the manuscript as it has bearing specifically to farming conditions in Scania during the study period, as the reviewer suggests.

**RC2 part 5:**

Section 2 'sources and methods' contain a presentation of methods and sources, but it also contains an analysis, which makes the manuscript structure confusing. The homogenizatino is a result in itself and should (preferably) not be presented in a section called 'sources and methods'.

I think a more classic structural approach of the manuscript (where material, method and analysis are presented in separate sections) could work better and it would improve readability.

**Answer:**

A similar point was made by RC1, and I do agree with the points made by both reviewers on the structure of the manuscript, where sources, methods and analysis each should appear in clearly distinguishable sections.

**RC2 part 6:**

Specific comments

The manuscript has a very speculative language. This affects the overall impact and scientific quality of the manuscript. The term 'relative' is used frequently and situations or climatic conditions are often described as 'relative'. A search indicates that the word 'relative' appears 62 times in the manuscript (four times in the abstract) in various contexts to describe a myriad av situations. For example, (line 65) is explained that 'relative peace' dominated in the 1700s compared to previous centuries. I am not familiar with Swedish history, what does this indicate in terms of wars and skirmishes? Is the frequency or magnitude of skirmishes that defines 'relative' peace? However, I do not think this historical overview of the political history of the region is necessary because the author does not return to this subject or its impact on harvests or threshing.

This use of relative continues throughout the manuscript. On page four, the term 'relative' occurs five times. On line 94, 'relative lack of wood' and on line 105 'in their relative specializations', and then on line 115, 'with relatively much…' and later in the same sentence 'saw a relatively large increase'. Finally, in the footnote on page 4 'relatively abundant'.

Relative is a subjective term and should be avoided as it in a scientific investigation provides no actual perspective of change, magnitude or proportions. For instance, what does a 'relative lack of wood', indicate or describe? Is it an indication of amount of wood, distance to wood (as in a forest), lacked access to firewood? And on line 85 (page 3) it is explained that the 1810s and 1840s were 'relatively cold', relative to what place and period? Relative to the 1600s? Was it colder all year round, was only the summers colder or was the 1810s colder than previous/later periods (decades) or what is just cold in comparison to warmer periods (decades)? Even in the results (line 464) it is said that the analysis shows 'relatively large negative associations' and Figure 13 includes an analysis of 'relatively wet years'. How wet is a 'relatively' wet year, is it possible to quantify and explain this in the text? Is a 'relatively wet year' wetter than 'normal' years, and if so, how much?

Finally, describing things as being 'relative' are vague and it raises a lot of questions. I think that rewriting and rephrasing many of the vague sentences would greatly improve the manuscript and make it easier to evaluate the content. The results would also stand out more clearly.

**Answer:**

The reviewer brings up issues related to using vague terms too much in a scientific context and that especially the word 'relative' occurs too frequently in the paper. I agree with the reviewer on the proposal to rewrite and rephrase many of these lines. Regarding the questions the reviewer raised on the specifics of the wet years shown on Fig. 13, they are described, together with dry years (Fig. 12) on lines 455-458.

**RC2 part 7:**

Technical comments

Line 14. In the abstract it says that new cultivars were being 'imported' at the end of the period from other parts of Europe. This is not discussed in the manuscript. Did the import start at the end of the 19th century or were new cultivars in use all across Scania by the end of the studied period? Why were they imported, was it caused by changes in climate, demand or other issues? As I read the article, there already seemed to be an extremely diversified variation of grains, but the new cultivars are just one variant of autumn-wheat and autumn rye? The questions are rhetoric but stems from how the subject is approached and the changes that occurred in the later part of the investigated period.

**Answer:**

The introduction of the early improved cultivars on a large scale is discussed on lines 116-118 and on lines 500-506. However, the fact that they were mostly imported from other parts of Europe is not mentioned and this information will be added to the manuscript. The introduction on a large-scale of early improved cultivars began in the end of the 19th century, which is implied in the manuscript but this could be clarified (Forsberg, 2015: 11-12). The review on landrace definitions and classifications made by Zeven (1998) also discusses this process.

For example, in 1891, the first share-holding company specializing on the distribution of such cultivars was founded in Svalöv, Scania (Leino, 2017). Smaller imports of early improved cultivars occurred throughout the 19th century, and possibly earlier as well. In parish descriptions from Malmöhus in 1828 there are some occasional mentions of 'Probsteier-rye' and 'Dutch winter-rye' (Bringeus 2013). The process was gradual, and around the turn of the century farmers in Scania were probably growing a mix of old and new varieties (Leino, 2017). This information will also be added to the manuscript.

The question of why they were imported and introduced is a whole research topic in itself. Based on my reading of the literature, it should be seen as a mix of ideology and demand. In the early phases, it was probably more ideological, spearheaded by the promoters of the agricultural reform movement in the 18th and 19th centuries (see for example Jones, 2016). Actual demand picked up in the late 19th century as the early improved cultivars were better suited to the type of agriculture that was increasingly manifesting itself in that period through better plowing, drainage, precision-sowing and fertilization, see Leino, 2017).

The older varieties were certainly more diversified than the early improved cultivars. However, the latter should not be confused with the homogenous varieties obtained through modern plant breeding in the 20th century. Leino (2017) describes how the early improved cultivars were obtained through 'mass selection', whereas modern plant breeding is oriented towards selecting for one particular strain.

**RC2 part 8:**

Line 121. It says that wheat-varieties (plural) will be included in the 1865-1911 period, but there is only one type (Autumn-wheat) mentioned in Figures 11, 12 and 13. In Figure 13 the caption reads 'during relatively wet years', but the figure only shows different periods, not years.

**Answer:**

In line with the discussions of grain varieties in the manuscript and the answer above, autumn-wheat should be understood as plural. Fig. 13 shows the relationship between grain and climate indicators (seasonal, monthly) during wet years. Wet years are defined on line 455-458. However, to clarify and make the manuscript more transparent, I propose to add all the wet and dry years in an Appendix.

**RC2 part 9:**

Line 132-135. I think this comparison of soils is unnecessary.

**Answer:**

Lines 132-135 describes research of rye landraces (distinguishable groups of rye varieties) and is relevant to the findings and discussion of the manuscript see lines 572-591.

**RC2 part 10:**

Line 156. Is this 'however-sentence' suggesting that black oat varieties were grown in Scania or just in neighboring provinces? In the previous sentence it was already stated that it is uncertain (remove 'more' in 'more uncertain' as there are no different levels of uncertainty).

**Answer:**

It is suggesting that black oat varieties were grown in Scania. I propose to remove the word 'more' from the sentence.

**RC2 part 11:**

Line 168 (section 1.4), it is concluded that the diversity of grain varieties "testifies to a relatively flexible farming system in terms of sowing and harvest dates as well as the ability to produce under differing agrometeorological conditions, not least during colder and wetter periods". Could the author please elaborate and explain how a large variety of grains give indications about farming systems and capability of adapting to different agrometeorological conditions? This subject almost seem like an article in itself (I do not think that the existence of different varieties is proof of different systems, we can assume, but it is still just an assumption), but if there is more information, please elaborate. For example, are successfull harvests seen, by the author, as an indication of adaptation?

**Answer:**

Regarding farming systems, it is not stated in the manuscript that the existence of different grain varieties is proof of different systems. I assume that the reviewer means farming systems. As I describe on line 103-105, farmers in Scania practiced a mixed farming system, where livestock husbandry and grain production were integrated and mutually dependent. Under this broad category of mixed farming systems, there were one-field, two-field and three-field systems (defined by the proportion of systematic fallow), either of which was practiced by most villages until the 19th century see lines 95-97, 99-100 or 127-129). Sometimes the terms one-course, two-course and three-course rotations are used. These definitions are in common usage in the Swedish (and European) historiography (Myrdal & Morell, 2011).

There has been a connection made in the literature between the type of farming systems and the composition of grain production, mostly between autumn-rye and two-field and three-field systems, due to the supposed need for a full years fallow after harvest (see the comment made in relation to late-rye on line 127). However, I have not found any such connections looking at data from Scania (unpublished investigations). As farming systems are defined in the manuscript, there is no question in the literature as to there being different systems (variants on the mixed farming system), and I would agree with the reviewer that the cultivations of different grain varieties do not imply different systems. The argument in the manuscript is rather that in the farming systems in Scania farmers could and did grow many different crops, but in particular barley, rye and oats. Within each type of grain, there were multiple varieties that offered further flexibility in sowing and harvesting.

In relation to the first question the reviewer asked, I first refer to lines 175-181 where I discuss the topic of adaptation from a theoretical perspective. The distinct concept of resilience is defined on lines 172-174. Following these definitions, I would like to emphasize that in the lines specified by the reviewer (lines following line 168), I do not explicitly state that the large variety of grains give indications of the farming systems ability to adapt, merely that it offered "the ability to produce under differing agrometeorological conditions", which I would rather put under the "resilience" category. Similar arguments have been made by Michaelowa (2001) for the case of England (which he compares to France). By multiple scholars for the cultivation of both autumn-rye and barley in Finland (see for example Taavitsainen et al, 1998; Holopainen & Helama, 2009; Solantie, 2012; Huhtamaa et al, 2015) and by Berg (2007:11) regarding the use of both spring- and winter-crops, as well as other garden crops in Sweden in the 18th and 19th centuries.

Section 1.4 is called "Crop diversity and resilience" presents the available research on this topic and follows an extensive discussion of the various grain varieties (Section 1.3). The answer to the reviewers question can be found in Section 1.3 and Section 1.4. In essence, the answer boils down to the differences in phenology for different grains. Different grains were sown or harvested during different times during the year, had disparate requirements in terms of accumulated temperature, sunlight and precipitation, at distinct times of the year, and finally, some varieties were more susceptible than others to different weather extremes (see Section 1.3 and references above). In the discussion section of the manuscript, Section 3.2, I note that there was a difference between the grains in their respective relationships to temperature and precipitation, although mainly limited to the summer months (see lines 609-611).

A contributing and complicating factor in this regard was that there was much variety within each grain variety (in contrast with the mass selections of grain varieties describe above, and especially the latter modern plant breeding, the older grain varieties were selected more randomly across a given field, incorporating thousands or millions of roughly similar but distinct grain seeds. I refer to this variety in the discussion lines 611-612, see references therein).

Regarding the final question, successful harvests would be a sign of adaptation, given the definition stated on lines 174-181. However, take heed of the fact that I include exaptation in this definition of adaptation. I would emphasize that this manuscript is mainly concerned with the overall relationship between grain production and climate variability and not to what extent farmers were aware of and exploited this relationship to their advantage. Nonetheless, as I propose on lines 567-573, the results indicate some degree of long-term adaptation towards relatively cool and humid conditions, at least as far as grain production is concerned.

**RC2 part 12:**

Line 333. The figure could be improved. It reads that the figure reveal that all clusters are in Hjärnarp and Toastarp. However, looking at the figure, I cannot see the mentioned places.

**Answer:**

The location of the specified parishes are noted on line 334. I could add parish numbers and a related legend listing all the parishes to the map, although that might be confusing since the analysis is mainly concerned with groups of villages and the county of Malmöhus as a whole. The number of villages are too many to make the legible on a map on this scale (there are 381 villages in the HDSA sample, see Table 1).

**RC2 part 13:**

Line 358. Says that there are four clusters. Should be three? I guess it is the same in Figure 2 caption.

**Answer:**

Should be three clusters.

**RC2 part 14:**

Lines 459. The text has references to clusters in Figure 9. There are, however, no clusters mentioned in the figure caption and it is a bit confusing to understand what is indicated. The figure caption explains that the figure shows

grain and temperature/hydroclimatic indicators, where are the clusters? A similar reference to clusters and Figure 10 is found in lines 467-468.

**Answer:**

The clusters are indicated by the # next to each type of grain. For example Barley #2 is barley production in cluster 2. This should be included in the figure captions or notes.

**RC2 part 15:**

Line 466. The authors says that spring and autumn gives 'almost no statistically significant results', but values are not presented. This was also mentioned by the first RC

**Answer:**

Only statistically significant values have colored cells. This fact should be included into the descriptions to the figures, as suggested in response to RC1. The correlation coefficients between each type of grain and each month or season are included into the respective cells in all figures, where they can be easily observed by the reader.

**RC2 part 16:**

Line 460. Could r-values be included so that the reader could understand the difference between 'slight negative values' and 'relatively weak'? This would help to follow the author's line of thought.

**Answer:**

Regarding the figures, r-values are included as described in the answer above. However, the reviewer is perhaps referring to r-values being included into the actual text, and I propose to do this in the parts of the manuscript where it is relevant.

**RC2 part 17:**

Line 521. This sentence seems to suggest that a correlation of 0.31 is high, as in strong? In the manuscript it states that it is 'quite high' (quite is again a relative term), but is 0.31 high in comparison to other studies, results or years? If correlation is 0.31 then it is low. Please elaborate on this subject and clarify what is meant.

**Answer:**

The points brought up by the reviewer is important, and the language in the manuscript should be revised so instances where the word high (as in strong) is used, they should be replaced with the word "strong" or an equivalent. Similar measures should be taken with the word low (as in weak), which should be replaced with the word "weak" or an equivalent, as to improve readability and clarity of the manuscript.

The evaluation of a specific correlation coefficient should be done in line with statistical principles as well as the context of the study. To take a trivial example, if I found a correlation coefficient of 0.8 when correlating the number of times the letter 'k' appears in my text document when I press the physical key "k" on my laptop keyboard (let us say with a total n of 100), I would consider this weak. In the context of studying the relationship between grain yields and temperature variability on "the margins of agriculture" in Finland, a correlation coefficient of 0.8 would have been seen as very high (see Huhtamaa, 2015, where correlations rarely reach >=0.4).

Returning to the specific example brought up by the reviewer, the correlation of 0.31 is not strong (it is in the bottom of a range of statistically significant correlations between 0.31 and 0.74), and the specified sentenced should be revised accordingly. In other figures, for example Fig. 9, a correlation of 0.29 is high as in strong, considering the time span involved and the lack of detail in the climate variables (compared to figures in Palm, 1997 or Edvinsson et al, 2009, who also studies southern Sweden). It is also strong compared to the other climatic indicators included in the analysis. Overall, it should be expected that lower correlation coefficients be obtained when analyzing longer time periods, given adaptation on the part of farmers and the plant material. Furthermore,

with a lower level of detail in the climate data, one would also expect lower correlation coefficients (both Edvinsson et al, 2009 and ouin et al, 2020, makes similar arguments). It should also be contextualized the historical context of farming. On the very margins of agriculture, one might expect higher correlation coefficients from even seasonal or annual climate indicators. For example, consider northern Scandinavia where late spring and summer temperatures, roughly May through August (MJJA), is more clearly distinguished as the most limiting factor for grain production (Nordli, 2003 & Nordli et al, 2004 used harvest dates from central Norway to reconstruct MJJA temperatures). Implied by these high correlations is also the higher frequency of harvest failure.

In Scania in contrast, grain production was constrained by a more complex combination of agro-meteorological indicators. Precipitation, the nutrient status of the soil (in northern Scandinavia grain farming was mostly done on nutrient-rich soils where soil depletion often led to farm abandonment, see Antonsson, 2004) as well as the humidity of the soil appear to have had a similar importance as temperature (see Utterström, 1957 or Edvinsson et al, 2009). Thus, when Fig. shows a correlation of 0.28 between total grain production in Cluster 2 and June temperatures, this is a strong correlation. Consider: the length of time (ca 150 years), the lack of detail in the climate data (monthly or seasonal averages), that temperature was not the dominant constraining factor and finally the correlations between grain production and other monthly and seasonal climatic indicators. I propose to add a discussion along these lines to the discussion section in the manuscript.

**RC2 part 18:**

Line 533. What does 'this trend' refer too? Was the shift towards wheat reinforced by new variations of autumn rye? Please elaborate or improve sentence.

**Answer:**

This refers to the fact that these new varieties of autumn-rye (the new improved cultivars) were more temperature-sensitive. The sentence should be improved to clarify this.

**RC2 part 19:**

Line 552. The first sentence reads that grain production was 'mainly constrained by precipitation', but the next sentence reads 'However, instead of focusing on just temperature…'. This is a bit confusing. Should it be 'Instead of focusing on precipitation...'?

**Answer:**

Correct, should be revised accordingly.

**RC2 part 20:**

Line 556. What argument is referred too? The dominating role of precipitation? Is this an argument or a result of previous studies?

**Answer:**

Argument made by Utterström (1957) and Edvinsson et al (2009). Utterström (1957) is missing in the reference list of the manuscript and will be added in a revision.

**RC2 part 21:**

Line 558. What period is here referred too? The late 19th century (line 556) or the study-period? This is needed to follow the line of thought without having to read all other studies.

**Answer:**

Refers especially to the 18th century. 'period' should be replaced by 'the 18th century'.

**RC2 part 22:**

Line 606. The sentence after the parenthesis is very speculative. "A relatively… probably… to some extent."

**Answer:**

Should be revised in line with previous comments and answers above.

**RC2 part 23:**

Line 646. What does 'total production' refer to?

**Answer:**

Should be 'total agricultural production'.

**RC2 part 24:**

Line 652. What is considered as the pre-industrial period in Scania? This historical term appears twice in the manuscript, but I do not know when the industrial period started in Southern Sweden or how it relates to the agrarian revolution (Line 60). Is the import of new cultivars (line 14) part of the agrarian revolution or the industrial revolution?

**Answer:**

Roughly the period before the late 19th century. Industrialization in Scania began ca 1850. The agrarian revolution took place between roughly 1750-1870 in Scania, and was characterized first and foremost of dramatic increase in agricultural production relative to population. This increase in agricultural production has been associated with institutional changes (enclosures), new crop rotations and crops, increased market integration, new technological implements, increased use of iron in agricultural tools. The subsequent industrial period brought about, among other things, chemical fertilizers, increased mechanization of labor, the new improved cultivars and the use of steel in agricultural tools.

**RC2 part 25:**

Caption in Figure 6 and 7. A technicality, but I would rather see the sources for this figure included in the captions instead of referring to the footnote in Table 2.

**Answer:**

Sources could be added to Fig. 6 and Fig. 7 captions.

**References**

EEC: DLO-Alterra Wageningen UR, DLO-Plant research International Wageningen UR, NEIKER, Derio, Spain, Institute of Technology and Life Sciences (ITP), Warsaw, Poland & Swedish Institute of Agricultural and Environmental Engineering (JTI), Uppsala: Recommendations for establishing Action Programmes under Directive 91/676/EEC concerning the protection of waters against pollution caused by nitrates from agricultural sources. Review and further differentiation of pedo-climatic zones in Europe, http://publications.europa.eu/resource/cellar/e1d06bc3-58c4- 43a3-b2bc-6ad6d53d7953.0001.01/DOC_1, last accessed 1 July 2021, 2011.

Nordli, P. Ø.: Spring and summer temperatures in Trøndelag 1701-2003. Met.no report 5, 2004.

Nordli, P. Ø., Lie, Ø., Nesje, A. & Dahl, S. O.: Spring–summer temperature reconstruction in western Norway 1734–2003: a data-synthesis approach. International Journal of Climatology, 23(15), doi: 10.1002/joc.980, 2003.

Parry, M.L. and Carter, T.R.: The effect of climatic variations on agricultural risk. Climatic change, 7, 95-110, doi: 10.1007/BF00139443, 1985.

Schils, R. et al: Cereal yield gaps across Europe. European Journal of Agronomy, 101, 109-120, doi: 10.1016/j.eja.2018.09.003, 2018.

Dobrovolný, P., Moberg, A., Brázdil, R., Pfister, C., Glaser, R., Wilson, R., van Engelen, A., Limanówka, D., Kiss, A., Halícková, M., Macková, J., Riemann, D., Luterbacher, J., Böhm, R.: Monthly, seasonal and annual temperature reconstructions for Central Europe derived from documentary evidence and instrumental records since AD 1500. Climatic Change 101, 69-107, doi: 10.1007/s10584-009-9724-x, 2010.

Olsson, M. and Svensson, P.:Agricultural growth and institutions: Sweden 1700-1860. European Review of Economic History, 14, 275-304, doi: 10.1017/S1361491610000067, 2010.

Boogard, H, Wolf, J., Supit, I., Niemeyer, S. & van Ittersum, M: A regional implementation of WOFOST for calculating yield gaps of autumn-sown wheat across the European Union, Field Crop Research, 143, 130-142, doi: 10.1016/j.fcr.2012.11.005, 2012.

Metzger, M. J, Bounce, R. G. H., Jongman, R. H. G, Mücher, C. A., & Watkins, J. W.: A climatic stratification of the environment of Europe. Global Ecology and Biogegraphy, 14, 549-563, DOI: 10.1111/j.1466-822x.2005.00190.x, 2005.

Tidblom, A.V.: Einige Resultate aus den meteorologischen Beobachtungen angestellt auf der Sternwarte zu Lund in den Jahren 1741 - 1870. Lund Universitets årsskrift 12, 1876.

Domonokos, P. and Coll, J.: Homogenisation of temperature and precipitation time series with ACMANT3: Method description and efficiency tests. Int. J. Climatol., 37, 1910-1921, doi: 10.1002/joc.4822, 2017.

Slicher Van Bath, B. H.: Yield ratios, 810-1820. A.A.G. Bijdragen 10, Wageningen, 1963.

Utterström, G.: Jordbrukets arbetare - Levnadsvillkor och arbetsliv på landsbygden från frihetstiden till mitten av 1800-talet. Diss. Faculty of Humanities, Stockholm University, 1957.

Antonsson, H.: Landskap och Ödesbölen: Jämtland före, under och efter den senmedeltida agrarkrisen. Diss. Division of Agrarian History, Swedish University of Agricultural Sciences, 2004.

Nordli, P. Ø.: Spring and summer temperatures in Trøndelag 1701-2003. Met.no report 5, 2004.

Edvinsson, R., Leijonhufvud, L. and Söderberg, J.: Väder, skördar och priser i Sverige. In: Liljewall, B., Flygare, I., Lange, U., Ljunggren, L. and Söderberg, J. (eds) Agrarhistoria på många sätt. 28 studier om människan och jorden. Kungl. skogs- och lantbruksakademien, 2009.

Beillouin, D., Schauberger, B., Bastos, A., Ciais, P. and Makowski, D.: Impact of extreme weather conditions on European crop production in 2018. Phil. Trans. R. Soc. B 375:20190510, doi: 10.1098/rstb.2019.0510, 2020.

Nordli, P. Ø., Lie, Ø., Nesje, A. & Dahl, S. O.: Spring–summer temperature reconstruction in western Norway 1734–2003: a data-synthesis approach. International Journal of Climatology, 23(15), doi: 10.1002/joc.980, 2003.

Huhtamaa H., Helama S., Holopainen J., Rethorn C. & Rohr C.: Crop yield responses to temperature fluctuations in 19th century Finland: provincial variation in relation to climate and tree-rings. Boreal Env. Res. 20: 707–723, 2015.

Holopainen J. & Helama S.: Little Ice Age farming in Finland: Preindustrial agriculture on the edge of the grim reaper's scythe. Hum. Ecol. 37: 213–25, 2009.

Michaelowa, A.: The Impact of Short-Term Climate Change on British and French Agriculture and Population in the First Half of the 18th century. In: Jones, P.D., Ogilvie, A., Davies, T.D. and Briffa, K.R. (eds) History and Climate. Memories of the Future? Springer books, 2001.

Myrdal, J. and Morell, M.:The agrarian history of Sweden: from 4000 BC to AD 2000. Lund: Nordic Academic Press, doi: 10.3098/ah.2012.86.3.130, 2013

Solantie R. 2012. Ilmasto ja sen määräämät luonnonolot Suomen asutuksen ja maatalouden historiassa [The role of the climate and related nature conditions in the history of the Finnish settlement and agriculture]. Jyväskylä studies in humanities 196, University of Jyväskylä, Finland. [In Finnish with English abstract], 2012.

Taavitsainen J.P., Simola H. & Grönlund E.: Cultivation history beyond the periphery: Early agriculture in the North European boreal forest. Journal of World Prehistory 12(2): 199–253, 1998.

Berg, B. Å.: Volatility, Integration and Grain Banks. Studies in Harvests, Rye Prices and Institutional Development of the Parish Magasins in Sweden in the 18th and 19th Centuries. Diss. Stockholm School of Economics, 2007.

Bringeus, N. A.: Sockenbeskrivningar från Malmöhus Län 1828. Arcus Förlag, 2013.

Leino, W.M.: Spannmål - Svenska Lantsorter. Stockholm: Nordiska Museets Förlag, 2017.

Edvinsson, R., Leijonhufvud, L. and Söderberg, J.: Väder, skördar och priser i Sverige. In: Liljewall, B., Flygare, I., Lange, U., Ljunggren, L. and Söderberg, J. (eds) Agrarhistoria på många sätt. 28 studier om människan och jorden. Kungl. skogs- och lantbruksakademien, 2009.

Jones, P. M.: Agricultural Enlightenment – Knowledge, Technology and Nature, 1750-1840, Oxford University Press, 2016.

Zeven, A. C.: Landraces: A review of definitions and classifications. Euphytica 104(2), 127-139, doi: 10.1023/A:1018683119237, 1998.

Forsberg, N. E. G.: Spatial and Temporal Genetic Structure in Landrace Cereals. Diss. Norwegian University of Science and Technology, Department of Biology, 2015.

**List of changes to the manuscript**

The list below describes the most important changes made to the manuscript. For specifics regarding all the rewritten paragraphs, minor technical and language corrections, please see the Author's track-change file.

- Restructuring of the sections of the article, including the removal of section 1.4.1, rewriting parts and renaming section 2.3, 2.4 and adding section 2.5. Added a proper start to Section 3, rewrote and reshuffled section 3.1, adding sections 3.1.1, 3.1.2, 3.2, 3.2.1, 3.2.2 and 3.2.3, removing section 3.3. Added a new section for the discussion, section 4, including subsections 4.1, 4.2, 4.3 and 4.4. Also added a separate section number for the conclusions, section 5.
- Added explicit terminology for the "early study period" (1702-1865) and "later study period" (1865-1911).
- Notable structural changes to the manuscript were changes to the methods and sources section as well as the results section where results and methods are more clearly separated. Specifically, this involved the removal of large parts of the previous section 2.3 where I presented the homogenization of the Lund monthly instrumental temperature series, which is now more comprehensively detailed and discussed in Appendix A. In the revised manuscript, the homogenization method is briefly summarized in Section 2.4, homogenization results are briefly presented in section 3.1.1 and discussed in section 4.3. Section 2.2 has been partly rewritten and divided into section 2.2 and 2.3.
- "Cleaning" results section, section 3, of discussions deemed to belong in the discussion section, section 4.
- Added estimations of the average occurrences of the first autumn frost and last spring frosts during the period 1863-1911 (roughly the later study period), see sections 2.4 and 3.1.2 and 4.1.
- Inclusion general sowing and harvest dates for each main grain in the studied region in section 1.3.
- Inclusion more details regarding the climate in Scania in the 19$^{th}$ century in section 1.1.
- Minor language improvements throughout the manuscript, including the removal of the excessive use of the word "relatively" and similar terms, an issue that was pointed out by RC2.
- Rewriting or removal of unclear or otherwise inappropriate sentences and paragraphs, for example the first paragraph in section 1.2.
- Added descriptions of how p-values were employed for figures 10-14 as well as the use of coloring of cells and the denominations used for each cluster.
- Added two appendices to the manuscript, Appendix A, where the homogenization procedure of the Lund monthly instrumental temperature series is more comprehensively detailed and discussed and Appendix B where tables with dry and wet years are presented.
- Corrected all figure and table captions.
- Corrected all minor technical details pointed out by RC1 and RC2, except in a few cases when there was no actual error as understood by the author.
- Added new and missing references.

---

## Referee Report (RR1)

I liked this paper! It is interesting and I could only find minor flaws.

If I can understand Figure 9 and 10 (correlations for the earlier period until 1865) and figure 11, correlations for the later period (1865-1911), climate sensitivity INCREASED. It is possible that this is an effect of the "enclosure" movement in Scania during the first ½ of 19th century.

In my opinion, something of that kind is hinted at on p. 20, where Cluster 3 (peasant-farmers on freehold land) dominated. It is expected from theory that private ownership will generate greater risk-taking (= more sensitivity to weather conditions) than tenant farming. It is quite possible that Cluster 1 and 2 reflect inflexible leaseholds where tenants were encouraged NOT to experiment, but rather deliver a fixed - or as close to fixed as possible - amount of lease to the landowner.

Minor corrections and thoughts (as they occur):

Line 6 &7: I don't understand why a low share of temperature-sensitive proxy-variable (wheat) is a good thing if you want to study climate variability.

Line 30-31: An admirable ambition to provide an understanding of phenology of historical grain varieties -when this ambition is presented later in the article (p 5), it is rather thin. E.g. the different rye varieties, Larsmässoråg, Svedjeråg etc, is not shown to have different phenology/being of different races. I've always understood these "varieties" as being harvested at Lars mässa or grown on slash-and-burn land.

Line 44: "early study period (1702-1911) and the late study period (1865-1911)".

Ought to be? "early study period (1702-1864) and the late study period (1865-1911)".

Line 45: "conceptualized neither in a simplistic or deterministic"

Should be: "conceptualized neither in a simplistic nor deterministic"

page 3, line 77-86: I get the impression that cold periods in 1740s and 1780s were associated with sand drift etc. But soil erosion was not a problem in the 1694-1698, when it was really cold. I got an impression of inconsistency in argument.

Line 89: the great transformation of agriculture during the period makes it difficult to identify climate signal. True. So why did you choose the period? (=maybe a few lines about sources etc).

Line 158-159: "selection of barley seed a long-term adaptation process"…. Hm? Wasn't the most common way that peasants took some of their harvest as seed for next year? Also, seed grain was not so "pure", if I remember correctly Maths Isacsson and Täpp Peterson (both in Dalecaria) have shown that the grain seed could be so mixed that a farmer THOUGHT he sowed barley but it was so mixed with oats that "the barley turned to oats" (cos of the rainy weather).

Line 181-185: "a flexible farming system", check out Ronny Peterson "Ett reformverk under omprövning" where he discusses the problems with falling production in the late 18[th] century as a driving force for the "enclosure" movement. (Also, be careful with that concept since the connotation in English is different to Swedish conditions prevailing.)

Line 201: "If such adaptations were took place…" = "If such adaptations took place…"

Line 524: "Practically no /-/ correlation /-/autumn wheat /-/ 0.46)": this sentence indicate no, or low, correlation for autumn wheat. But on line 521, the same correlation of 0.46 is regarded as a good result. (I agree – it is not bad. But it has to be equally good (or bad).

Line 602: "not only precipitation but rather the combination of precipitation and precipitation during the summer…" I don't understand.

Line 649-654: I found this rather an ad hoc argument. Why should the "trade deficit" between Scania and Sweden proper result in more northerly grain varieties? As before (and prior to Monsanto™), farmers took part of their harvest and used for seed the next year. I think you might just delete those rows.

Figure 12 & Figure 13 and Table B1, B2, B3 & B4 are really good! Keep at all cost!

---

## Author Response (AR2)

CP-2021-52
**Climate variability and grain production in Scania, c. 1702–1911**
Martin Karl Skoglund
Special Issue: International methods and comparisons in climate reconstruction and impacts from archives of societies
Handling Editor: Chantal Camenisch, chantal.camenisch@hist.unibe.ch

**Author's response**

In the following, I have divided the RC3 (comments from referee #3) into distinct and answerable parts. Author's comments are denoted AC3. RC3 and AC3 are followed by a reference list. In the end of the document there is a list where all changes made to the manuscript are summarized.

**Comments and responses (RC3 and AC3)**

**RC3 part 1:**

"I liked this paper! It is interesting and I could only find minor flaws.
If I can understand Figure 9 and 10 (correlations for the earlier period until 1865) and figure 11, correlations for the later period (1865-1911), climate sensitivity INCREASED. It is possible that this is an effect of the "enclosure" movement in Scania during the first ½ of 19th century.
In my opinion, something of that kind is hinted at on p. 20, where Cluster 3 (peasant-farmers on freehold land) dominated. It is expected from theory that private ownership will generate greater risk-taking (= more sensitivity to weather conditions) than tenant farming. It is quite possible that Cluster 1 and 2 reflect inflexible leaseholds where tenants were encouraged NOT to experiment, but rather deliver a fixed - or as close to fixed as possible - amount of lease to the landowner."

**AC3 part 1:**

Fig. 11 shows higher correlation coefficients than those in Fig. 9 and Fig. 10, and I agree that this might be interpreted as climate sensitivity increasing, at least for the spring-crops and in the face of summer droughts, such as those occurring in the years 1868, 1870 and 1899 (see lines 90-92). However, an important caveat here is the smaller number of years; therefore, I would be careful to make a direct comparison based on the magnitude of the correlation coefficients alone.

Several authors have argued that the Swedish enclosures increase incentives for long-term investments, for example in land improving investments such as diking and other types of draining of lands. Such land improvements increased the share of high-yielding cultivated soils, while at the same time increasing the risk of drought (see the first two paragraphs in Section 4.1). Nyström (2018) found that enclosed farms in Scania did experience slightly increased risks in agricultural production compared to non-enclosed farms in the period 1750-1850. The results obtained here are in line with those results. I have added a few lines in the discussion Section 4.1 where I highlight the institutional difference as well as the differences in soil qualities between Cluster 3 and the other clusters in order to further emphasize this point raised by the referee.

**RC3 part 2:**

"Minor corrections and thoughts (as they occur):

Line 6 &7: I don't understand why a low share of temperature-sensitive proxy-variable (wheat) is a good thing if you want to study climate variability."

**AC3 part 2:**

Previous research on this subject has often been focused on temperature-sensitive grains like wheat or alternatively in marginal areas where temperature was clearly the most important agro-meteorological constraint. However, Scania is a case where farmers was largely cultivating a mix of grains not as sensitive to temperature as wheat while at the same time Scania was far from a marginal grain-producing region (see lines 55-58). These are conditions similar to those pointed out by Michaelowa (2001) as making English grain production more resilient than French grain production during cold periods in the 18[th] century (see lines 195-198). In the Swedish-language historiography, Utterström (1957) and later Edvinsson et al (2009) argued that grain production in southern Sweden was mainly limited by precipitation rather than temperature (as in northern Sweden). A study of the relationship between grain production and climate variability in the 18[th] and 19[th] centuries offers a possibility to further explore these arguments.

**RC3 part 3:**

"Line 30-31: An admirable ambition to provide an understanding of phenology of historical grain varieties -when this ambition is presented later in the article (p 5), it is rather thin. E.g. the different rye varieties, Larsmässoråg, Svedjeråg etc, is not shown to have different phenology/being of different races. I've always understood these "varieties" as being harvested at Lars mässa or grown on slash-and-burn land."

**AC3 part 3:**
I agree that based on the available (to my knowledge) historical source material any possible understanding on the characteristics and extent of different grain varieties is necessarily thin, evident for example on page 5 as suggested by the reviewer. Larsmässoråg (roughly translated as *St Laurentius Day-rye*) was sown around the 10[th] of August, however actual sowing and harvesting dates varied by village and by year, according to local conditions (see lines 137-142). Furthermore, while St Laurentius Day-rye did appear in some instances in Scania, Carl von Linné (1751) points out that it was mostly found on manors in the middle of the 18[th] century. In the parish descriptions cited in Section 1.3, no St Laurentius Day-rye is mentioned and the earliest general sowing date mentioned for autumn-rye is in the middle of August (Bringéus, 2013). Regarding whether St Laurentius Day-rye was a variety with a distinct phenology, the sources do seem to indicate that it did (for example Linné, 1751, Leino, 2017). In the manuscript, Larsmässoråg was incorrectly translated as autumn-rye, this has been changed to *St Laurentius Day-rye.*

In relation to rye varieties, there are two points I would highlight. The first is that the available rye varieties offered in the early study period offered a broad range of possible sowing and harvesting dates (see Section 1.3). The second point I would highlight is the shift to new and more temperature-sensitive autumn-grain varieties in the late 19[th] century (see lines 614-620).

The implications of different grain varieties in terms of the overall relationship between grain production and climate variability is discussed in Section 4.2.

**RC3 part 4:**

"Line 44: "early study period (1702-1911) and the late study period (1865-1911)".
Ought to be? "early study period (1702-1864) and the late study period (1865-1911)"."

**AC3 part 4:**
Corrected.

**RC3 part 5:**
"Line 45: "conceptualized neither in a simplistic or deterministic"
Should be: "conceptualized neither in a simplistic nor deterministic"."

**AC3 part 5:**
Corrected.

**RC3 part 6:**
"page 3, line 77-86: I get the impression that cold periods in 1740s and 1780s were associated with sand drift etc. But soil erosion was not a problem in the 1694-1698, when it was really cold. I got an impression of inconsistency in argument."

**AC3 part 5:**
Increasing sand drift and soil erosion was associated by Mattson (1987) to colder temperatures *as well as* intensified land use practices *and* an increase in heavy winds and storms, particularly easterlies (see lines 79-86). Presumably, these other factors in addition to colder temperature were absent in the cold period in the late 17[th] century. However, there is much less data on wind patterns from the 17[th] century compared to latter centuries. Regarding land use, it has been established that an intensification took place during the 18[th] century and this factor can therefore explain the different results for the latter periods as compared to the 1694-98 period (Bohman, 2010).

**RC3 part 6:**
"Line 89: the great transformation of agriculture during the period makes it difficult to identify climate signal. True. So why did you choose the period? (=maybe a few lines about sources etc)."

**AC3 part 6:**
The statement on line 89 refers mainly to the 19[th] century, constituting roughly half of the total study period. While the annually resolved climate and grain production data is available from the 18[th] century, in the 19[th] century there is even more data, making it feasible to conduct a study of the relationship between grain production and climate variability, despite potential difficulties in easily identifying detrimental or beneficial climatic periods for agriculture (except those years of summer droughts discussed on lines 90-95). Annually resolved data allows for detrending and controlling for the expansion of agricultural expansion during the period.

**RC3 part 7:**
"Line 158-159: "selection of barley seed a long-term adaptation process".... Hm? Wasn't the most common way that peasants took some of their harvest as seed for next year? Also, seed grain was not so "pure", if I remember correctly Maths Isacsson and Täpp Peterson (both in Dalecaria) have shown that the grain seed could be so mixed that a farmer THOUGHT he sowed barley but it was so mixed with oats that "the barley turned to oats" (cos of the rainy weather)."

**AC3 part 7:**
The sentence on lines 158-159 refers to a suggestion made by Cockram *et al*, 2007 who discusses the increasing divergence found in genetic markers of barley seed across northern and southern Europe over the very long term (i.e. in the last 7000-9000 years after the introduction of domesticized grains in Europe), and propose that the mechanism underlying this divergence was adaptation of farming to local natural conditions.

I would agree with the reviewer that the most common way farmers obtained their seed during the period was from the previous years harvest. In the framework of Cockram *et al* (2007), this would have led to adaptation over the very long term. This type of adaptation process can be both "passive" or "active" (see lines 188-194).

To my knowledge, the grains cultivated in Scania and subsequently paid in tithes would have been mostly pure categorizes of grain. Note that there was a distinct category for mixed-grains, which in the context of Scania was mainly a mix between barley and oats, similar to the reviewers example from Dalecarlia. Distinctions between barley and other grains would have been quite important due to the use of barley of brewing beer. Rye was importantly used for yeast bread and was mostly grown as an autumn-crop, distinguishing it from the other spring-crops of barley, oats and mixed-grains.

Furthermore, barley and rye was generally seen as more qualitative grains in terms of nutrition for humans, whereas oats was generally seen as a lower quality grain and often used as fodder for cattle (Dahl, 1942). Finally, each type of grain was priced and valued differently, including mixed-grains, implying that there was an incentive and interest in making sure grains were categorized accurately.

**RC3 part 8**
"Line 181-185: "a flexible farming system", check out Ronny Peterson "Ett reformverk under omprövning" where he discusses the problems with falling production in the late 18th century as a driving force for the "enclosure" movement. (Also, be careful with that concept since the connotation in English is different to Swedish conditions prevailing.)"

**AC3 part 8:**
The sentence in the mentioned lines mainly refers to sowing and harvesting dates. However, as the referee indicates there was a debate at the time (which in some ways is still ongoing) whether the traditional farming systems of *tegskifte* (Swedish variant of open-fields) was inefficient and inflexible. Farming in *tegskifte* could be done in a myriad of ways. Dahl (1989) lists no less than 62 different types of crop-fallow rotations in Scania during the 18th century.

It should be mentioned that in Scania, grain production was increasing throughout most of the latter part of the 18th century, as well in the following century, i.e. before and after the Swedish enclosure reforms. Studies on the effects of enclosure in Scania have found that farms that underwent enclosure experienced a greater increase in production, compared to those that did not (Olsson & Svensson, 2010). Eventually, pretty much all farms and villages underwent enclosure. The Swedish enclosure reforms (*storskifte, enskifte* and *laga skifte*) are briefly discussed on lines 112-118, where I also discuss the farming systems of the early study period.

**RC3 part 9:**
"Line 201: "If such adaptations were took place…" = "If such adaptations took place…"."

**AC3 part 9:**
Corrected.

**RC3 part 10:**
"Line 524: "Practically no /-/ correlation /-/autumn wheat /-/ 0.46)": this sentence indicate no, or low, correlation for autumn wheat. But on line 521, the same correlation of 0.46 is regarded as a good result. (I agree – it is not bad. But it has to be equally good (or bad)."

**AC3 part 10:**
The mentioned sentence has been rephrased.

**RC3 part 11:**
"Line 602: "not only precipitation but rather the combination of precipitation and precipitation during the summer…" I don't understand."

**AC3 part 11:**
Should be **temperature** and precipitation and has been duly corrected.

**RC3 part 12:**
"Line 649-654: I found this rather an ad hoc argument. Why should the "trade deficit" between Scania and Sweden proper result in more northerly grain varieties? As before (and prior to Monsanto™), farmers took part of their harvest and used for seed the next year. I think you might just delete those rows."

**AC3 part 12:**

Farmers did indeed take part of their harvest for seed for the next year. The vast majority of seed was most likely obtained in this way. However, there was also trade in seeds across Sweden, even though its extent is not known. Seed was traded within villages as well as over broader regions. Most known here is the import of rye seed from Finland or the imports of barley varieties like Bråkorn from northernmost Sweden to Bergslagen, the Mälaren valley and as far south as Östergötland (Leino, 2017).

The argument being made on line 649-654 is that such trade across the Sound with the Danish provinces would have reduced after 1658, and conversely that trade in the northwards direction would have increased. Of course, trade in grain for consumption and trade in grain seeds should be considered as two separate phenomena. (Note the example above were barley seed was imported from grain deficit regions like Lappland to grain surplus regions in the South.) While the new political and administrative reality in Scania after 1658 had an effect on trade and possibly on lateral seed exchange with the Danish provinces, the sources are extremely sparse of the latter type of exchange and thus the argument on lines 649-654 is in the end purely speculative. Furthermore, they are only indirectly related to the results presented in the manuscript. Hence, after due consideration of the referee comment in the matter, I have removed the lines as suggested.

**RC3 part 13:**
"Figure 12 & Figure 13 and Table B1, B2, B3 & B4 are really good! Keep at all cost!"

**AC3 part 13:**

No changes made.

**References**

Bohman, M.: Bonden, bygden och bördigheten: Produktionsmönster och utvecklingsvägar under jordbruksomvandlingen i Skåne ca 1700-1870. Diss. Lund University, oai: DiVA.org:umu-99296, 2010.

Bringeus, N. A.: Sockenbeskrivningar från Malmöhus Län 1828. Arcus Förlag, 2013.

Cockram, J., Jones, H., Leigh, F.J., O'Sullivan, D., Powell., W., Laurie, A.D. and Greenland, A.J.: Control of flowering time in temperate cereals: genes, domestication, and sustainable productivity. Journal of Experimental Botany, 58(6), 1231-1244, doi: 10.1093/jxb/erm042, 2007.

Dahl, S.: Studier i äldre skånska odlingssystem. Stockholm: Stockholms universitet, 1989.

Dahl, S.: Torna och Bara. Studier i Skånes bebyggelse-och näringsgeografi före 1860: Carl Bloms boktryckeri, 1942.

Edvinsson, R., Leijonhufvud, L. and Söderberg, J.: Väder, skördar och priser i Sverige. In: Liljewall, B., Flygare, I., Lange, U., Ljunggren, L. and Söderberg, J. (eds) Agrarhistoria på många sätt. 28 studier om människan och jorden. Kungl. skogs- och lantbruksakademien, 2009.

Mattsson, J.O.: Vinderosion och klimatändringar. Kommentarer till 1700-talets ekologiska kris i Skåne. Svensk Geografisk Årsbok, 1987.

Olsson, M. and Svensson, P.:Agricultural growth and institutions: Sweden 1700-1860. European Review of Economic History, 14, 275-304, doi: 10.1017/S1361491610000067, 2010.

Utterström, G.: Jordbrukets arbetare: levnadsvillkor och arbetsliv på landsbygden från frihetstiden till mitten av 1800-talet. Diss. Stockholm University, 1957.

von Linné, C.: Carl von Linnés skånska resa 1749. Wahlström & Widstrand, 1751.

**List of changes to the manuscript**

- Slight changes in the text in line with RC3 part 4, 5, 9, 10, 11, 12
- Added a few lines to Section 4.1 in line with RC3 part 1.
- Removed all footnotes in line with request from the review file validation. Footnotes judged superfluous were completly removed while the rest were incorporated into the text. Footnotes 1, 2, 5 and 6 were removed. Footnotes 3, 4, 7, 8, 9, 10 and 11 were incorporated into the text in a slightly revised form.
- Added a missing reference to the reference list, Bringéus (2013).
- Removed superfluous reference, i.e. Jones *et al* (2005).
- Larsmässoråg was incorrectly translated as autumn-rye, this has been changed to *St Laurentius Day-rye.*